# Cocaine-context memories are transcriptionally encoded in nucleus accumbens *Arc* ensembles

Marine Salery[1,2,7] ✉, Arthur Godino [1,2,7], Yu Qing Xu[1,2], John F. Fullard [2,3,4,5], Romain Durand-de Cuttoli [1,2], Alexa R. LaBanca [1,2], Leanne M. Holt [1,2], Scott J. Russo [1,2], Panos Roussos [2,3,4,5,6] & Eric J. Nestler [1,2,3] ✉

Learned associations between the rewarding effects of drugs and the context in which they are experienced are critical for context-induced relapse. While context re-exposure triggers the recall of such drug-related associative memories it is unclear whether this relies on the reactivation of and plasticity in neuronal populations previously engaged in their acquisition. Here, using the immediate early gene *Arc*, we captured a discrete population of nucleus accumbens (NAc) cells activated during the encoding of cocaine-context memory in mice and showed that this neuronal ensemble is later reactivated upon context-induced recall. Furthermore, we show that ensembles recruited at early vs. late stages of memory encoding are largely distinct and contribute differentially to memory retrieval. Single nuclei RNA-sequencing of these ensembles identified plasticity-related transcriptional programs that segregate cocaine-recruited NAc engram-like cells beyond cell-type composition and revealed molecular features unique to distinct stages of memory processing. These findings suggest that activity-dependent transcription upon initial engram allocation further stamps cells for persistent plasticity programs and thereby supports memory traces at the single-cell level. This study also provides insights into the mechanisms supporting pathological memory formation associated with cocaine exposure.

The lasting retention of associative memories formed between the rewarding effects of drugs and the context in which they are experienced contributes critically to the persistence of high risks of context-induced relapse[1–5]. Associative memories have been proposed to be encoded in the brain in sparse and highly discriminative populations of concomitantly activated neurons that represent the memory trace of a given experience[6–10]. These ensembles of neurons, also referred to as engram cells, are recruited during memory formation and their reactivation supports memory retrieval[11–17]. Emerging evidence indicates that cocaine-recruited neuronal ensembles could similarly encode the pathological memories that underlie some aspects of addiction[18–23].

Engram cells can be segregated based on the initiation of activity-dependent mechanisms such as the induction of immediate early genes (IEGs). These genes—whose induction is efficiently coupled to

[1]Nash Family Department of Neuroscience, Icahn School of Medicine at Mount Sinai, New York, NY 10029, USA. [2]Friedman Brain Institute, Icahn School of Medicine at Mount Sinai, New York, NY 10029, USA. [3]Department of Psychiatry, Icahn School of Medicine at Mount Sinai, New York, NY 10029, USA. [4]Department of Genetics and Genomic Sciences, Icahn Genomics Institute, Icahn School of Medicine at Mount Sinai, New York, NY 10029, USA. [5]Center for Disease Neurogenomics, Icahn School of Medicine at Mount Sinai, New York, NY 10029, USA. [6]Mental Illness Research Education and Clinical Center (VISN 2 South), James J. Peters VA Medical Center, Bronx 10468 NY, USA. [7]These authors contributed equally: Marine Salery, Arthur Godino. ✉e-mail: marine.salery@mssm.edu; eric.nestler@mssm.edu

synaptic activation—have emerged as reliable proxies to define the spatiotemporal organization of neuronal ensembles[24–28]. In the nucleus accumbens (NAc), sparsely distributed IEG-inducing cells provide a robust map of cocaine-recruited ensembles[29–32], with FOS-expressing cells contributing to relapse to cocaine seeking[33–35]. However, it remains unclear whether cells recruited in the engram during cocaine exposure could later contribute to the expression of context-induced retrieval of drug-related memories, similar to what has been proposed in other Pavlovian learning paradigms[36–39]. Further, while IEGs have been extensively used as markers of cellular activation to identify neuronal populations recruited by a specific stimulus, less attention has been paid to the lasting molecular consequences of their induction within ensemble cells. Given that IEGs are postulated to be critical links between synaptic activity and downstream neuronal plasticity (at both the transcriptional and synaptic levels)[40], one important question is to understand how plasticity mechanisms engaged in engram cells at early stages of learning can further support their contribution to memory storage and retrieval.

Here, we address these crucial questions by leveraging one particular IEG, *Arc*. *Arc* encodes activity-regulated cytoskeleton-associated (ARC) protein, which has been shown to functionally regulate several aspects of cocaine action within the NAc[41–46], making it an ideal IEG for identifying cocaine-elicited ensembles. Among other immediate early genes, *Arc* stands out because of its: (1) unique role as a direct effector at the synapse[47,48], (2) still unclear role in the nucleus[42,49], (3) distinctive mechanisms of transcriptional regulation[50], (4) hypothesized role as a homeostatic regulator where it might be induced to promote

"negative" plasticity responses[51,52], and (5) unusual viral-like ability to transfer from cell to cell at least in vitro[53]. Altogether this argues for a singular role for *Arc* in mediating long term synaptic plasticity and thus supporting learning and memory. Using transgenic ArcCreER[T2] mice[36], we captured ensemble cells recruited at different stages of cocaine-context memory encoding and assessed their further reactivation during memory retrieval. We also used single nucleus RNA-sequencing (snRNAseq) to not only determine the cellular profile of cocaine-recruited ensembles, but also assess the transcriptional adaptations that occur in ensemble cells over time.

## Results

### Cocaine-recruited ARC ensembles in NAc are reshaped by repeated cocaine exposure

As a first step in exploring ARC ensemble recruitment in the NAc, we measured endogenous ARC induction in response to acute or repeated cocaine exposure. Mice were habituated to a new context for 3 days, and then administered cocaine either acutely or repeatedly in this same context (Fig. 1a). *Arc* mRNA induction, measured 1 h after the last injection, was significantly lower following repeated cocaine than acute cocaine (Fig. 1b). We next assessed the correlation between *Arc* mRNA levels and cocaine-induced hyperlocomotion for each individual mouse. We found a positive correlation in the acutely exposed group and showed that it was lost in the repeatedly exposed group in which *Arc* mRNA induction was lower (Fig. 1c). Other IEGs such as *Fos*, *Fosb*, *Zif268* and *Npas4* (all of which have been implicated in cocaine action in NAc[32]) also exhibited similar desensitization at the bulk mRNA

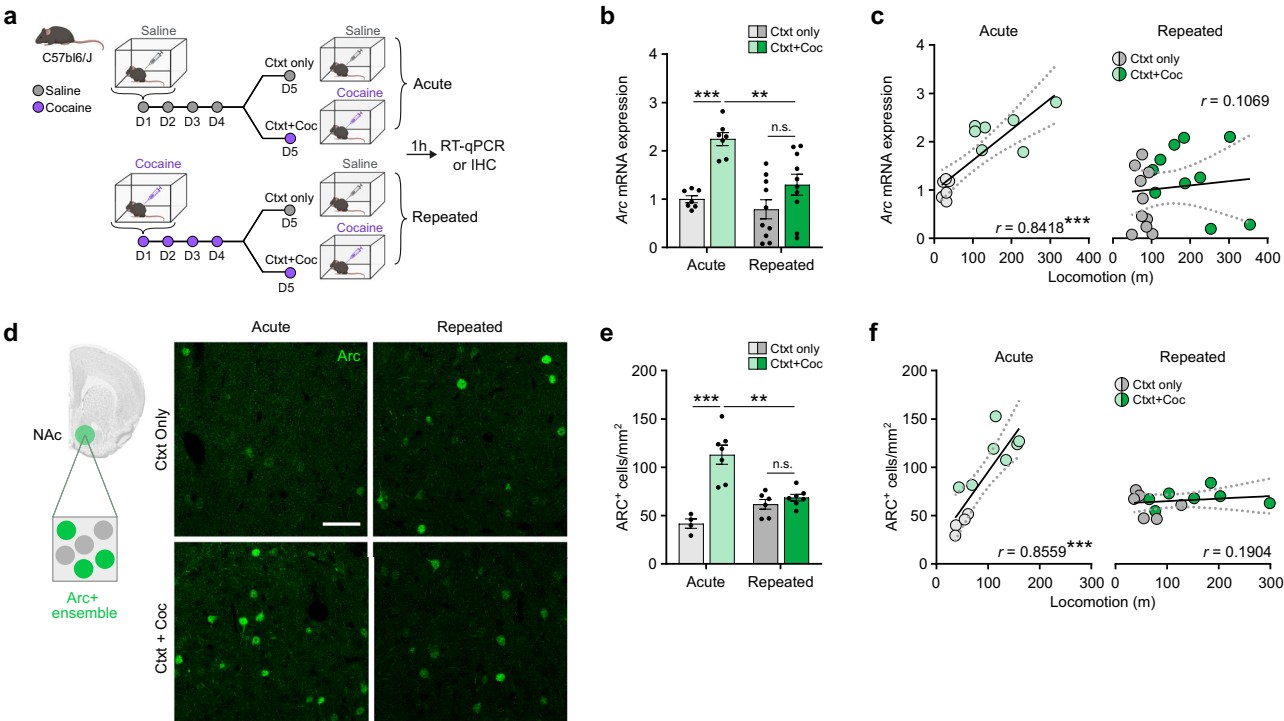

**Fig. 1 | A smaller ensemble is recruited after repeated cocaine-context associations. a** Experimental design for acute vs. repeated experimenter-administered cocaine regimens in male C57BL/6 J mice. IHC Immunohistochemistry. Ctxt context, Coc cocaine. **b** *Arc* mRNA expression in NAc. $n = 7$, acute; $n = 10$, repeated. Two-way ANOVA: interaction regimen x drug, $F_{1,30} = 3.897$, $p = 0.058$; main effect of regimen, $F_{1,30} = 9.577$, **$p = 0.0042$; main effect of drug, $F_{1,30} = 22.04$, ***$p < 0.0001$; followed by a Šidák post-hoc tests. **c** Correlation between *Arc* mRNA levels and cocaine-induced locomotion. Acute (left): $n = 14$, Pearson's $r = 0.8418$, ***$p = 0.0002$. Repeated (right): $n = 20$; Pearson's $r = 0.1069$, $p = 0.6536$. **d** Representative images of ARC+ cells (green, ARC+ ensemble) in NAc. Scale bar,

50 µM. **e** Number of ARC+ cells in NAc. $n = 4$, acute/ctxt only; $n = 7$, acute/ctxt + coc; $n = 6$, repeated/ctxt only; $n = 7$, repeated/ctxt+coc. Two-way ANOVA: interaction regimen x drug, $F_{1,20} = 21.04$, ***$p = 0.0002$; main effect of regimen, $F_{1,20} = 3.050$, $p = 0.0961$; main effect of drug, $F_{1,20} = 31.07$, ***$p < 0.0001$; followed by Šidák post-hoc tests. **f** Correlation between the number of ARC+ cells and cocaine-induced locomotion. Acute (left): $n = 11$; Pearson's $r = 0.8559$, ***$p = 0.0008$. Repeated (right): $n = 13$; Pearson's $r = 0.1904$, $p = 0.5333$. Bar graphs are expressed as mean ± SEM. Correlation graphs show the regression line with a 95% confidence interval. Source data are provided as a Source Data File. Created in BioRender. Parise, E. (2025) https://BioRender.com/voi7l54.

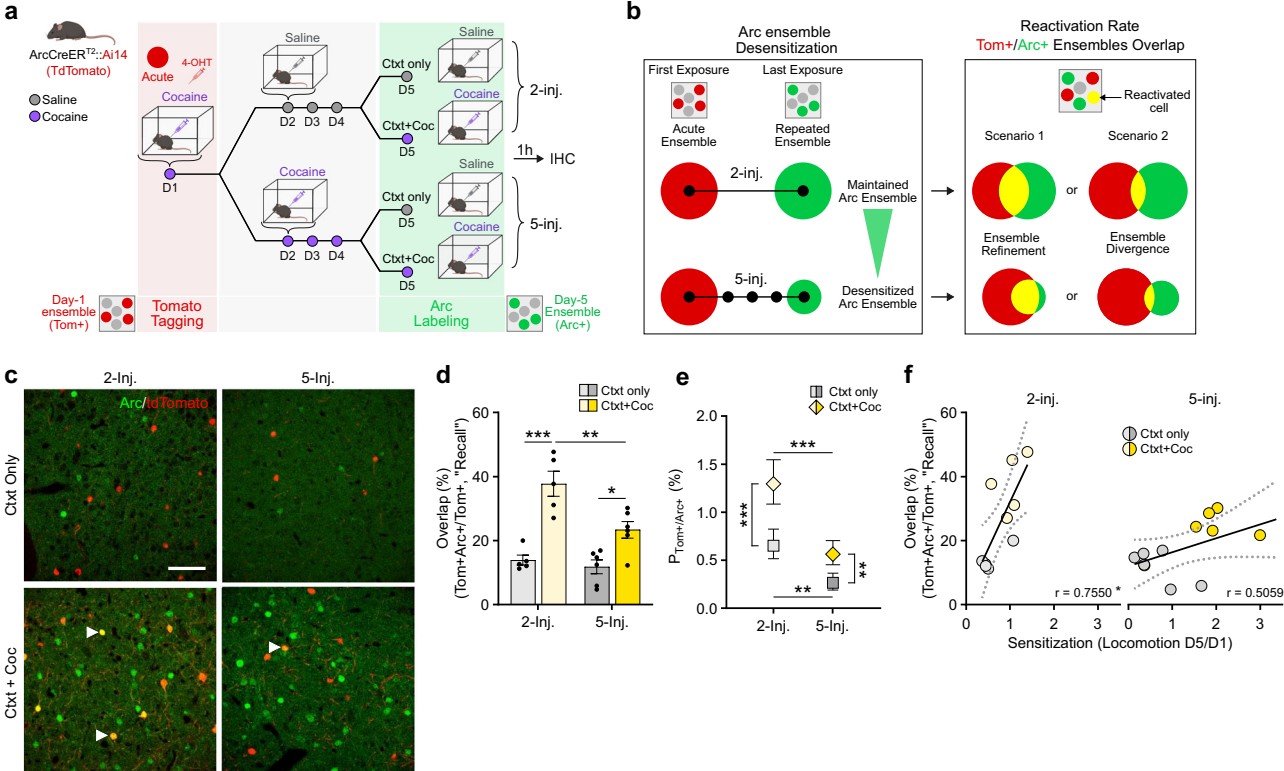

**Fig. 2 | A mostly distinct ensemble is recruited after repeated cocaine-context associations. a** Experimental design for ensemble tagging in a 2-injections (2-inj.) vs. 5-injections (5-inj.) paradigm in Arc-CreER[T2]::Ai14 mice. 4-OHT 4-Hydroxytamoxifen, IHC Immunohistochemistry. Ctxt context, Coc cocaine. **b** Schematic representation of ARC ensemble desensitization after 5 cocaine exposures vs. 2 cocaine exposures (left) and of two possible scenarios for how the 2- and 5-injection (inj.) ensembles compare in cell identity (right). **c** Representative images of cells activated on day 1 (Tom+, red), day 5 (ARC+, green). The arrows indicate representative overlapping neurons that were activated both on day 1 and day 5. Scale bar, 50 μM. **d** Overlap ratio (as "Recall", over total number of Tom+ cells). $n = 5$, 2-inj. groups; $n = 6$, 5-inj. groups. Two-way ANOVA: interaction regimen x challenge, $F_{1,18} = 5.296$, $*p = 0.0335$; main effect of regimen, $F_{1,18} = 9.591$, $**p = 0.0062$; main effect of challenge, $F_{1,18} = 44.09$, $***p < 0.0001$; followed by Šidák post-hoc tests.

**e** Chance-adjusted probabilities of cell activation on both day 1(Tom+, red) and day 5 (ARC+, green). $n = 11906$ cells in $n = 5$ mice, 2-inj.group/Ctxt only; $n = 11042$ cells in $n = 5$ mice, 2-inj./Ctxt+Coc; $n = 14746$ in $n = 6$ mice; 5-inj./Ctxt Only; $n = 15275$ cells in $n = 6$ mice, 5-inj./Ctxt+Coc. Baseline-Category Logit Mixed Model (BCLogMM) equation for Tom+/ARC+ counts: interaction regimen x drug, $z = 0.267$, $p = 0.79$; main effect of regimen, $z = −4.411$, $***p < 0.0001$; main effect of drug, $z = 4.630$, $***p < 0.0001$; followed by Šidák post-hoc tests. **f** Correlation between the overlap ratio and locomotor sensitization. 2-inj. (left): $n = 10$, acute; Pearson's $r = 0.7550$, $*p = 0.0116$. 5-inj. (right): $n = 12$; Pearson's $r = 0.5059$, $p = 0.0933$. Bar graphs are expressed as mean ± SEM. Regression lines and probability estimates are shown with their 95% confidence interval. Source data are provided as a Source Data File. Created in BioRender. Parise, E. (2025) https://BioRender.com/voi7l54.

level, although of varying amplitude and effect size (Supplementary Fig. 1).

We then evaluated the size of ARC+ ensembles in the NAc in the same paradigm (Fig. 1d). The number of cocaine-activated ARC+ cells was significantly lower in the repeated group, indicating the recruitment of a smaller ARC+ ensemble as compared to the one recruited by an acute cocaine injection (Fig. 1e), consistent with the desensitization of *Arc* observed at the mRNA level (Fig. 1b) and recently published work[46]. Moreover, and similar to what we observed with *Arc* mRNA levels, we found a positive correlation between the size of the recruited ensemble and cocaine-induced hyperlocomotion in the acute group that was lost in the repeated group (Fig. 1f).

Together, our data show that the weakening of *Arc* mRNA induction over repeated cocaine exposure is associated with a diminution in the number of NAc cells recruited in the cocaine ensemble. We conclude that repeated sessions of cocaine exposure in a new context reshape ARC expression patterns and lead to a shift in ARC ensemble size. This shift resulted from repeated exposures to cocaine rather than as a function of time since a single re-exposure, 4 days after the initial one, did not alter ARC ensemble size, or the correlation of ensemble size with cocaine-induced hyperlocomotion (Supplementary Fig. 2).

## ARC ensembles recruited at initial vs. later stages of cocaine-related learning are mostly distinct

Because ARC induction patterns shift during repeated cocaine exposure, with acute vs. repeated ARC ensembles exhibiting different sizes and differently correlating with cocaine-induced hyperlocomotion, we hypothesized that the two ensembles might present distinct features and encode distinct experiences associated with different (initial vs. later) stages of context-reward associative learning. At the cellular level, this would be represented in the degree of overlap between these two ensembles: in other words, do acute and repeated ensembles comprise the same or distinct NAc cells? To measure this experimentally, we capitalized on ArcCreER[T2] mice (which induce Cre recombinase in an activity-dependent manner) crossed to Ai14 reporter mice (which express td-Tomato [Tom] in a Cre-dependent manner) to label cocaine-activated cells with the indelible expression of Tom upon 4-hydroxytamoxifen (4-OHT) injection (Supplementary Fig. 3a). Importantly, Tom expression was absent without 4-OHT treatment, demonstrating negligible levels of leak expression (Supplementary Fig. 3b–d). We leveraged this approach to quantify whether cells from the acute ensemble (recruited on the first day of training by the first injection of cocaine) are recruited again in the repeated ensemble (on the last day by a fifth injection of cocaine,

Fig. 2a). ArcCreER[T2]::Ai14 mice were treated with acute or repeated cocaine in a novel context as above, following a context-dependent locomotor sensitization protocol that heavily relies on NAc circuits and plasticity[42]. Cells activated on day 1 (acute ensemble) were permanently tagged with Tom via the concomitant injection of 4-OHT and cocaine, whereas cells recruited on day 5 (repeated ensemble) were visualized via the induction of endogenous ARC expression 1 h after the last cocaine injection (Fig. 2a).

We considered two possible outcomes (Fig. 2b): (1) The acute ensemble could be largely recruited again by the last cocaine exposure (scenario 1, high overlap), which—because the repeated ensemble is smaller in size compared to the acute one—we refer to as a "refinement" of the ensemble. (2) Alternatively, the last cocaine exposure after repeated treatment might mostly not re-recruit the acute ensemble (scenario 2, low overlap), but rather incorporate new cells reflecting a "divergence" between the NAc ensembles encoding early vs. later stages of context-reward associative learning. The overlap between acute (red, Tom+) and repeated (green, ARC+) ensembles was visualized in NAc (Fig. 2c) with reactivated cells (yellow, Tom+/ARC+) corresponding to cells recruited at both day 1 and day 5. The level of overlap was measured as the number of double-positive (Tom+/ARC+) cells normalized to the total number of Tom+ cells, over the total number of ARC+ or as an $F_1$-score (Supplementary Fig. 4).

We compared the extent of overlap between the acute ensemble and the one activated in response to a single cocaine dose four days later (2-injection condition, which does not cause ARC desensitization, Supplementary Fig. 2) vs. four subsequent consecutive days of cocaine injections (5-injection condition, which does cause ARC desensitization and a smaller ensemble size), with control mice receiving saline injections on day 5 to examine the effect of context re-exposure alone. We found significant overlap of ensemble cells activated by cocaine, as compared to the context only, in both the 2- and 5-injection conditions (Fig. 2d). However, the cocaine-induced overlap was significantly lower in the 5- vs. 2-injection groups (Fig. 2d). Because of ARC desensitization (Fig. 1), the re-exposure ARC+ ensemble in the 5-injection group was smaller than that of the 2-injection group. It is thus critical to verify that reduced reactivation does not simply result from random sampling with overall fewer ARC+ cells. In other words, one important consideration was to assess whether these effects were maintained when accounting for individual "chance" overlap levels, which depend on varying numbers of Tom+ cells, of ARC+ cells and of all cells analyzed in each animal. To that end, we first quantified overlap using symmetrical "Recall" and "Precision" calculations, and computed their harmonic mean as an $F_1$-score (see Methods for details). These analyses confirmed lower overlap for the 5-injection condition (Supplementary Fig. 4). To go further, we then appropriately[54] modeled this type of categorical, multinomial and nested count data largely using logit models, which perform in the log-odds space and allow for individual-wise chance levels (see Methods for details). The resulting chance-adjusted probabilities of overlap for individual Tom+/ARC+ cells belonging to both the initial and reactivated ensembles were significantly higher in both cocaine re-exposed groups as compared to context re-exposure only, but significantly decreased in the 5- vs. 2-injection condition (Fig. 2e). This rules out the possibility that decreased overlap in the repeated group results solely from decreased ARC+ ensemble size or from different Tom+ ensemble size. These results indicate instead that the repeated ensemble recruited after five consecutive injections of cocaine overlaps less with the acute ensemble and incorporates more newly activated cells compared to the ensemble recruited by cocaine after just one prior injection, suggesting that ARC desensitization is accompanied by a shift in ARC-inducing cells towards a new, largely separate ensemble (scenario 2, Fig. 2b). To evaluate the behavioral relevance of ensemble re-recruitment, we correlated the degree of overlap to context-dependent locomotor sensitization in individual

mice, and found a strongly positive correlation in the 2-injection group but not in the 5-injection group (Fig. 2f).

Together, our data show that a subset (20–40%) of the ensemble recruited by the first exposure to cocaine becomes recruited again upon re-exposure to the drug 4 days later, and that the extent of this re-recruitment drops substantially with repeated drug exposures, when a greater fraction of new cells constitutes the ensemble. These findings indicate that mostly distinct ensembles are recruited at early vs. later stages of cocaine-context associative memory encoding and suggest that acute and repeated ensembles might support distinct phases of this learning.

### Acute vs. repeated ARC ensembles are differentially recruited during the retrieval of cocaine-context associative memories

We next studied whether, akin to other types of associative learning[43,44], the retrieval of cocaine-context associative memories triggers the reactivation of ensembles previously activated during encoding, and if such retrieval-associated reactivation might differ between acute and repeated ensembles given their divergence during memory encoding (Fig. 1). To this end, we used a cocaine place preference (CPP) paradigm and evaluated whether NAc cells recruited in ArcCreER[T2]::Ai14 mice during cocaine-context conditioning are recruited again during subsequent memory recall. Encoding cells were permanently tagged (Tom+, red) via the concomitant injection of 4-OHT and cocaine during conditioning, whereas recall-activated cells were visualized using induction of endogenous ARC (ARC+, green) during the test session (Fig. 3a).

We first examined whether acute vs. repeated ensembles are more prone to reactivation during memory recall. We defined the level of recall-induced ensemble reactivation as the ratio of double positive Tom+/ARC+ cells to total Tom+ cells. The acute ensemble was captured upon an acute cocaine injection in a 1-pairing paradigm (1 P, weak conditioning), while the repeated ensemble was captured on the last day of repeated cocaine injections in a 5-pairing paradigm (5 P, strong conditioning, Fig. 3a). As expected, repeated pairings induced a much stronger preference for the cocaine-paired side than an acute pairing, indicating more robust memory formation and a strong vs. weak conditioning with repeated vs. acute treatment, respectively (Fig. 3b). Consistent with the relative desensitization of ARC induction after 5 daily doses of cocaine (Fig. 1e), the repeated/strong-conditioning ensemble was smaller than the acute/weak-conditioning one (Supplementary Fig. 5), confirming the ability our of tagging approach to efficiently capture these two ensembles in a CPP paradigm.

Both acute and repeated ensembles exhibited significantly higher reactivation after the test session as compared to homecage controls, yet with the repeated ensemble being less reactivated than the acute one (Fig. 3c, d). As in Fig. 2, we calculated "Precision", "Recall" ratio and $F_1$-score which confirmed this observation (Supplementary Fig. 5). As in Fig. 2, we further performed logit models to most appropriately correct for individual-wise "chance" levels due to random sampling across ensembles of varying sizes. While the chance-adjusted probability for a cell to be reactivated during the test session was likewise increased for both ensembles, this probability was still significantly lower for cells belonging to the repeated/strong-conditioning ensemble (Fig. 3e). Importantly, locomotor activity during CPP testing—while sensitized by cocaine conditioning—was not different between acute and repeated groups (Supplementary Fig. 5a), indicating that differences in locomotion per se do not underlie the observed differences in reactivation. We then asked whether the extent of an ensemble's reactivation correlates with memory strength during the test session. We found opposite patterns: the acute ensemble's reactivation was strongly positively correlated with CPP scores, whereas the repeated ensemble's reactivation was strongly negatively correlated (Fig. 3f). These data indicate that the recall of cocaine-context associative memories triggers the reactivation of a subset of cells that were

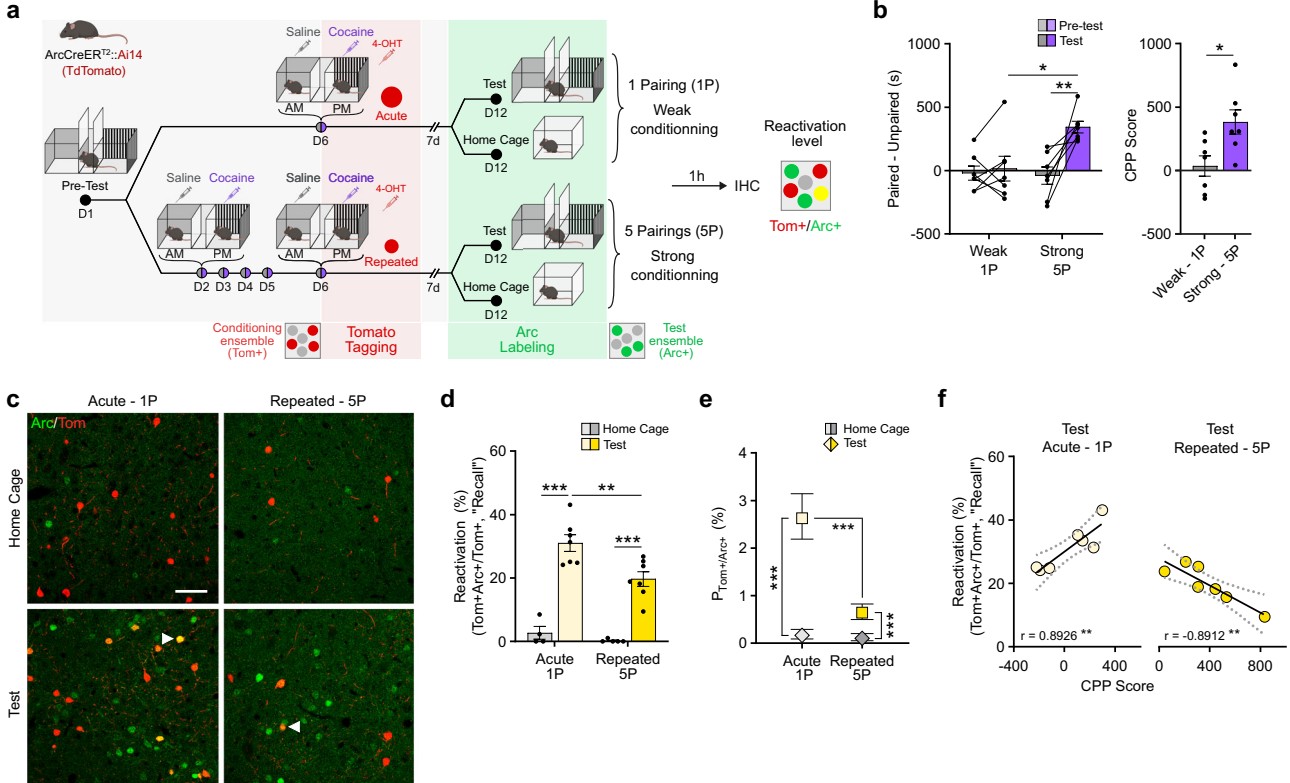

**Fig. 3 | Encoding-recruited acute and repeated cocaine ensembles are reactivated during the retrieval of cocaine-context associative memories.**
**a** Experimental design for CPP ensemble tagging in Arc-CreER[T2]::Ai14 mice. 4-OHT 4-Hydroxytamoxifen, 1 P 1 Pairing, 5 P 5 Pairings. **b** Time spent in the paired vs unpaired chamber (left) and CPP score (right). $n = 7$, weak; $n = 6$, strong. Left: Two-way RM-ANOVA: interaction conditioning x session $F_{1,24} = 6.23$, $*p = 0.0199$; main effect of conditioning $F_{1,24} = 4.89$, $*p = 0.0368$; main effect of session $F_{1,24} = 9.05$, $**p = 0.0061$; followed by Šidák post-hoc tests. Right: Two-tailed unpaired $t$-test: $t_{12} = 2.791$ $*p = 0.0163$. **c** Representative images of cells activated during conditioning (Tom+, red), test (ARC+, green) or both (arrows indicated). Scalebar, 50 μM. **d** Reactivation ratio (as "Recall", over total number of Tom+ cells). $n = 4$, 1 P/homecage; $n = 5$, 5 P/homecage; $n = 8$, 1 P/test; $n = 10$, 5 P/test. Two-way ANOVA: interaction conditioning x test, $F_{1,19} = 3.559$, $p = 0.0746$; main effect of conditioning, $F_{1,19} = 8.783$, $**p = 0.008$; main effect of test, $F_{1,19} = 104.9$, $***p < 0.0001$;

followed by Šidák post-hoc tests. **e** Chance-adjusted probabilities of cell activation during both conditioning (Tom+, red) and test (ARC+, green). $n = 8595$ cells in $n = 4$ mice, 1 P/homecage; $n = 16538$ cells in $n = 7$ mice, 1 P/test; $n = 9951$ cells in $n = 5$ mice, 5 P/homecage; $n = 16130$ cells in $n = 7$ mice, 5 P/test. Baseline-Category Logit Mixed Model (BCLogMM), equation for Tom+/ARC+ counts: interaction conditioning x test, $z = -2.093$, $*p = 0.0363$; main effect of conditioning, $z = -1.002$, $p = 0.3163$; main effect of test, $z = 8.984$, $***p < 0.0001$; followed by Šidák post-hoc tests. **f** Correlation between the reactivation ratio and CPP score. 1-pairing (left): $n = 7$; Pearson's $r = 0.8926$, $**p = 0.0068$. 5-pairing (right): $n = 7$, Pearson's $r = -0.8912$, $**p = 0.0071$. Bar graphs are expressed as mean ± SEM. Regression lines and probability estimates are shown with their 95% confidence interval. Source data are provided as a Source Data File. Created in BioRender. Parise, E. (2025) https://BioRender.com/voi7l54.

previously recruited during the encoding of these memories, suggesting engram-like properties for NAc ARC ensembles in this context. Moreover, and somewhat counter-intuitively, the ensemble supporting a weak memory was more likely to be reactivated during recall, with the extent of its reactivation correlating positively with memory performance. Conversely, the comparatively lower reactivation of the strong memory-encoding ensemble was correlated negatively with memory strength. These results suggest opposite contributions of these two ensembles in the retrieval of cocaine-context associative memories.

To causally test this hypothesis, we used optogenetic-mediated neuronal activation to artificially promote reactivation of acute vs. repeated NAc ensembles during the retrieval of cocaine-context associative memories. Using the same cocaine CPP paradigm, acute and repeated ensembles were permanently tagged in ArcCreER[T2]::Ai32 mice, which express channelrhodopsin (ChR2-EYFP) in a Cre-dependent manner (Supplementary Fig. 6), thus allowing the permanent expression of ChR2 in conditioning-recruited cells and their subsequent light-induced reactivation during the test (Fig. 4a). Experimental light-mediated reactivation of the acute, but not the repeated, ensemble elicited a higher CPP score during the test session

as compared to a no-light, within-subject control condition (Fig. 4b). As another control, reactivation of a cocaine ensemble captured in the homecage (where mice received equivalent exposures to cocaine without context conditioning) did not significantly affect CPP performance (Fig. 4c). These findings establish context-selective, yet differential roles for the acute and repeated ensembles and suggest that functionally distinct and mostly non-overlapping NAc cell populations cooperate to support both the encoding and the later retrieval of cocaine-context associative memories.

## Ensembles recruited during acute vs. repeated encoding of cocaine-context associative memories have distinct plasticity-related transcriptional signatures

Activity-dependent transcription is critical for the initiation of plasticity mechanisms within neuronal circuits supporting learning and memory[40]. In the NAc, a wide range of transcriptional adaptations have been implicated in drug-related memories[55]. Here, we aim to assess the level of confinement of such bulk changes to ensemble cells recruited during the encoding and retrieval of cocaine-context associative memories. We hypothesized that the initiation of activity-dependent transcriptional programs at the individual cell level could support the

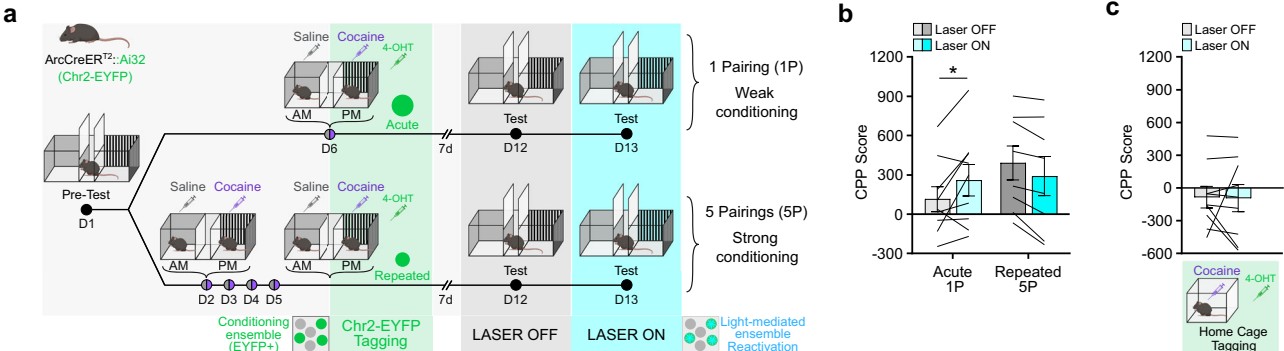

**Fig. 4 | Artificial reactivation of a cocaine-recruited ensemble. a** Experimental design for CPP ensemble tagging and optogenetic-mediated reactivation Arc-CreER[T2]::Ai32 mice. 4-OHT 4-Hydroxytamoxifen, 1 P 1 Pairing, 5 P 5 Pairings. **b** Light stimulation of the tagged ensembles during CPP test. $n = 9$, 1-pairing; $n = 8$, 5-pairing. Two-way RM-ANOVA: interaction conditioning x light, $F_{1,15} = 10.25$, **$p = 0.0059$; main effect of conditioning, $F_{1,15} = 0.8076$, $p = 0.3830$; main effect of light, $F_{1,15} = 0.3439$, $p = 0.5663$. **c** ArcCreER[T2]::Ai32 mice were injected i.p. with 4-OHT and cocaine in their homecage and the tagged ensemble was optogenetically reactivated during the test session. Light stimulation of the homecage ensemble during CPP test. $n = 8$. Two-tailed paired $t$-test, $t_{16} = 0.054$, $p = 0.9576$. Bar and line graphs are expressed as means ± SEM. Source data are provided as a Source Data File. Created in BioRender. Parise, E. (2025) https://BioRender.com/voi7l54.

recruitment of a discrete cell population into memory-encoding ensembles. Additionally, the differential contribution of the acute vs. repeated ensembles to memory retrieval could rely on divergent cellular properties, whether intrinsic or acquired through learning, coincident with or causal to *Arc* induction within ensemble cells. To more deeply phenotype the cells that constitute these ensembles, we combined snRNAseq with CPP testing to transcriptionally profile the individual NAc cells recruited in the ARC ensembles during the encoding and retrieval of cocaine-context associations by comparison to non-ensemble and to non-reactivated neighboring cells.

Similar to Tom tagging (Figs. 2, 3) and to ChR2 tagging (Fig. 4), acute and repeated conditioning ensembles were permanently tagged in ArcCreER[T2]::Sun1 mice, in which the permanent expression of a SUN1-GFP fusion protein allows for nuclei isolation via fluorescence-activated nuclei sorting (FANS, Supplementary Fig. 7). Tissue was collected 30 min after the beginning of the test session for the detection of recall-induced transcriptional programs in ensemble nuclei (Fig. 5a). By combining each nucleus's FANS status (GFP+/GFP-) and *Arc* mRNA expression (*Arc*+/*Arc*-), we segregated NAc nuclei and their transcriptional profiles according to their recruitment during, respectively, memory encoding (remote ARC induction, 7 days before retrieval) and memory recall (recent ARC induction, 30 min before sacrifice), or during both (reactivated, engram-like cells, GFP+/*Arc*+) (Fig. 5a, b).

We first assessed whether certain NAc cell types would be preferentially recruited in the cocaine ensemble and evaluated potential divergences between the acute vs. repeated ensembles. From a merged dataset containing transcriptomes from all captured GFP+ and GFP- nuclei, unsupervised dimensionality identified 5 cell type clusters that were further annotated by comparison to publicly available snRNAseq databases[56,57] and that recapitulated well-characterized, canonical NAc cell types (Supplementary Fig. 8 and Supplementary Data 2). When quantifying the proportion of ensemble cells (GFP+) within each cell type, both acute and repeated ensembles showed a striking depletion in glial cells to the benefit of neuronal enrichment (Fig. 5c, d). The NAc neuronal population is composed of two mostly non-overlapping populations of medium spiny projection neurons (MSNs) that segregate between expression of D1 vs. D2 dopamine receptors and that have been shown to oppositely modulate CPP behavior[58–60]. We considered whether biased recruitment of D1 or D2-MSNs into the acute and repeated ensembles could reflect a differential recruitment of this population at different stages of cocaine-context memory encoding. We found

that D1-MSNs accounted for the large majority of neurons (70%) recruited in the cocaine ensemble without any differences between the acute and repeated groups (Fig. 5c, d). This bias towards D1-MSNs is consistent with the large literature supporting the recruitment of this NAc cell type upon cocaine exposure[60–62]. We confirmed this result using fluorescent in situ RNA hybridization where we quantified the proportion of tagged cells that express D1 receptors (D1+) vs. those that do not (D1-) (Supplementary Fig. 9a–c). Acute and repeated ensembles exhibited a similar proportion (around 70%) of D1-MSNs, suggesting that the portion of recruited D1-MSNs does not contribute to between-group functional differences. However, within-group ratios of D1-MSNs differentially correlated with CPP scores and with ensemble reactivation levels only for the repeated ensemble. A higher proportion of D1-MSNs in the repeated ensemble was associated with both a lower CPP score (Supplementary Fig. 9d) and higher reactivation level upon recall of cocaine-context memories (Supplementary Fig. 9e). Furthermore, subclustering of D1-MSNs recapitulated recently published, transcriptionally-defined D1-MSN subclusters[57] (Supplementary Fig. 10a–c) and, consistent with those published observations, cocaine-activated (GFP+) cells were enriched in the same D1-MSN subcluster (Drd1-MSN-1) enriched for Ca[2+]-related transcripts (Supplementary Fig. 10d, e), which provided a robust cross-validation of both datasets. Together, these data suggest that more subtle cell-specific features (within subpopulations of ARC+ D1- and D2-MSNs) might account for the differences observed between acute and repeated ensembles, rather than a distinct cell-type composition.

Beyond cell-type composition, we hypothesized that specific activity-dependent transcriptional programs could be selectively induced in ensemble cells as a result of remote ARC induction. We thus investigated whether NAc cells allocated to the conditioning ensemble (GFP+) during memory encoding would exhibit persistent transcriptional features that distinguish them from cells that are not recruited into the ensemble (GFP-), and whether such features could further segregate acute vs. repeated ensembles at the molecular level. This first differential expression analysis was restricted to home-cage control samples to circumvent potential confounding effects of recall-induced transcription. Whether recruited in the acute or repeated ensembles, GFP+ cells separated from non-ensemble cells (GFP-) via the differential expression of numerous genes (Supplementary Data 3) associated with plasticity-related signaling pathways (e.g., dopamine cAMP signaling, synaptogenesis, Fig. 5e, f) – enrichment for which was significantly stronger in repeated ensemble cells, suggesting larger or

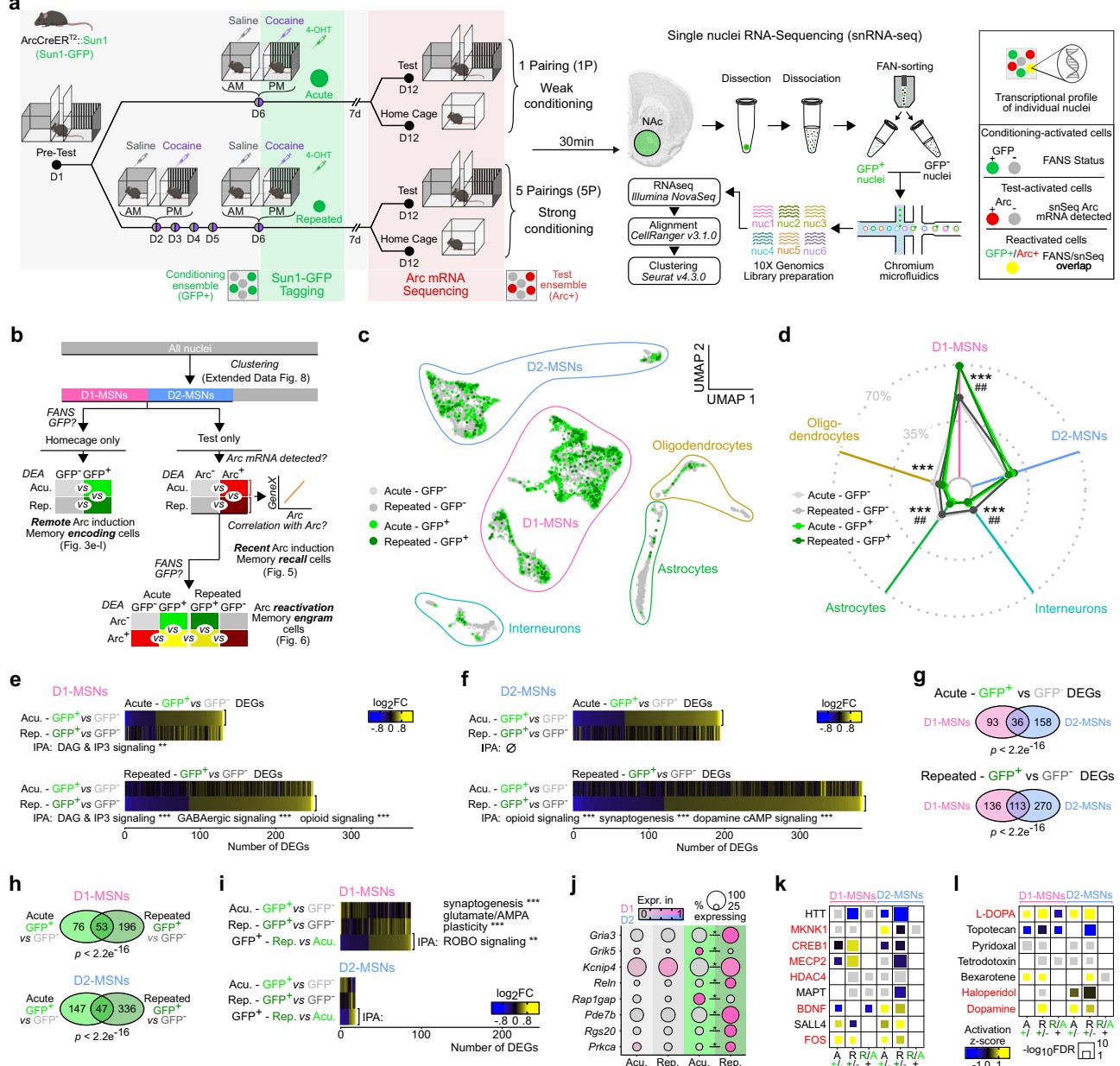

**Fig. 5 | Transcriptional profiles of cells recruited in the acute and repeated ARC ensembles during the encoding of cocaine-context associative memories.**
**a** Experimental workflow for CPP ensemble tagging, FANS and snRNAseq in Arc-CreER[T2]::Sun1 mice. 4-OHT 4-Hydroxytamoxifen, 1 P 1 Pairing, 5 P 5 Pairings. FANS Fluorescence-activated nuclei sorting. **b** snRNAseq analysis workflow for pairwise analyses of differentially expressed genes (DEGs). Acu. Acute, Rep. Repeated. **c** UMAP reduction and cell type annotation of all collected nuclei (n = 11,539). **d** Proportion of nuclei from each treatment/FANS status combination in each cell type cluster. Exact numbers are available in Supplementary Fig. 8c. $\chi^2$ tests: Acute: $\chi^2$ = 284.23, df = 4, p < 0.001. Repeated: $\chi^2$ = 74.26, df = 4, p < 0.001. FDR-adjusted p-values correspond to standardized Pearson's residuals. ***p < 0.001 GFP+ vs GFP-, Acute; ##p < 0.01 GFP+ vs GFP-, Repeated. **e** Expression heatmaps of GFP+ vs. GFP- DEGs in D1-MSNs in acute (top) and repeated (bottom) conditioning groups, along with 3 most significantly enriched IPA canonical pathways (** FDR < 0.01, *** FDR < 0.001). **f** Expression heatmaps of GFP+ vs. GFP- DEGs in D2-MSNs in acute

(top) and repeated (bottom) conditioning groups, along with 3 most significantly enriched IPA canonical pathways (** FDR < 0.01, *** FDR < 0.001). **g** Overlap of GFP+ vs. GFP- DEGs between D1- and D2-MSNs in either acute (top) or repeated (bottom) conditioning groups (Fisher's exact test p-values). **h** Overlap of GFP+ vs. GFP- DEGs between acute and repeated conditioning groups in D1-MSNs (top) and D2-MSNs (bottom) (Fisher's exact test p-values). **i** Expression heatmaps of DEGs in GFP+ nuclei between repeated and acute groups in D1-MSNs (top) and D2-MSNs (bottom), along with corresponding 3 most significantly enriched IPA canonical pathways (** FDR < 0.01, *** FDR < 0.001). **j** Expression dot plots of selected D1-MSN DEGs involved in neuronal physiology and signaling (*FDR < 0.05). **k** Most significant IPA predicted upstream regulator proteins across comparisons in D1- and D2-MSNs. **l** Most significant IPA predicted upstream regulator chemical substances across comparisons in D1- and D2-MSNs. Created in BioRender. Parise, E. (2025) https://BioRender.com/voi7l54.

more specific plasticity changes in those ensembles. Importantly, GFP+ nuclei exhibited similar patterns of differentially expressed genes (DEGs) across ensembles (acute and repeated, Fig. 5g) and cell types (D1- and D2-MSNs, Fig. 5h), as shown by similar gene expression

heatmap patterns and significantly overlapping gene lists. We then formally tested for statistically significant DEGs between acute and repeated GFP+ ensemble cells and found a higher number of DEGs in D1 than in D2 cells (Fig. 5i). Furthermore, these repeated vs. acute DEGs

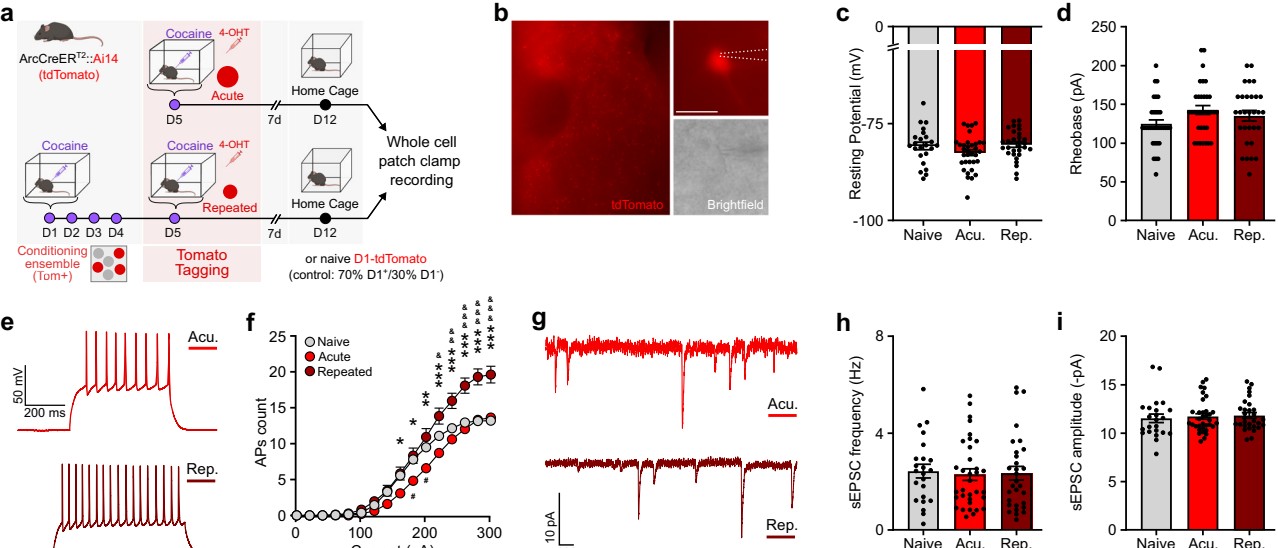

**Fig. 6 | Cells recruited in the acute vs. repeated ensembles exhibit different intrinsic excitability. a** Experimental design for ensemble tagging and ex vivo electrophysiology study of Arc-CreER^T2^::Ai14 mice. 4-OHT 4-Hydroxytamoxifen. Whole cell patch clamp recordings of Tom+ cells (n = 34 neurons, acute; n = 30 neurons, repeated) were performed in ArcCreER^T2^::Ai14 male mice (n = 8, acute; n = 7, repeated). For controls, recordings from naive D1-tdTomato male mice (n = 11) allowed for patching neurons (n = 33) in fixed relative proportions (70% D1+ / 30% D1-) matching those of Arc ensembles (Supplementary Fig. 9) **b** Representative wide-field (top) and close-up (bottom) views of patched MSNs. Scale bars 20 μm. **c** Resting membrane potential. LMM-ANOVA: $F_{2,13.668} = 1.88$, $p = 0.1897$. Acu. Acute, Rep. Repeated. **d** Rheobase. LMM-ANOVA: $F_{2,20.96} = 1.7497$, $p = 0.1983$. **e** Representative membrane responses from a Tom+ neuron in response

to a 280 pA current injection from an acutely (top) or chronically (bottom) treated mouse. **f** Number of evoked action potentials (APs) in response to increasing depolarizing current steps. LMM-ANOVA: interaction group x current, $F_{30,1410.4} = 9.72$, ***$p < 0.001$; main effect of group, $F_{2,20.3} = 4.85$, *$p = 0.0190$; main effect of current, $F_{15,1410.4} = 525.56$, ***$p < 0.001$; followed by Šidák post hoc tests (repeated vs acute: *$p < 0.05$, **$p < 0.01$, ***$p < 0.001$; repeated vs naive: ^&^$p < 0.05$, ^&&^$p < 0.01$, ^&&&^$p > 0.001$; acute vs naive: ^#^$p < 0.05$). **g** Representative voltage-clamp recordings of Tom+ neurons held at -70 mV. **h** Spontaneous excitatory synaptic currents (sEPSC) frequency. LMM-ANOVA: $F_{2,14.12} = 0.026$, $p = 0.9749$. **i** sEPSC amplitude. LMM-ANOVA: $F_{2,12.96} = 0.17$, $p = 0.8453$. Bar and line graphs are expressed as means ± SEM. Source data are provided as a Source Data File. Created in BioRender. Parise, E. (2025) https://BioRender.com/voi7l54.

were largely involved in synaptic function, signaling, and plasticity; a few select examples—glutamate receptors (*Gria3, Grik5*), ion channels (*Kcnip4*), and signal transducing proteins (*Reln, Rap1gap, Pde7b, Rgs20*)—are highlighted in Fig. 5j. Such larger differences in D1-MSNs compared to D2-MSNs might relate to a stronger role for D1-MSNs in drug-induced plasticity at the synaptic and circuit level[60]. These cluster-specific differential expression patterns—obtained via logistic regression modeling—were consistent with those obtained using other cell-level statistical frameworks commonly used for mouse brain snRNAseq such as Wilcoxon Rank-Sum testing[63–68], negative binomial testing[69], and MAST[63,70,71], as well as with those obtained using random pseudo-sampling followed by pseudo-bulk differential expression testing[72] (Supplementary Data 6). These observations confirmed that the acute and repeated ensembles, while sharing some common transcriptional properties (i.e., a shared transcriptional profile unique to all cocaine-activated cells and distinct from non-ensemble cells), also exhibit finer neurophysiology-related differences that could underlie their distinct functional features. Finally, upstream regulator predictions confirmed that both acute and repeated ensembles engage transcriptional programs that are likely mediated by canonical striatal plasticity pathways (e.g., CREB, MAPK, BDNF signaling), and likely in a dopamine-dependent manner, in both D1- and D2-MSNs (Fig. 5k, l).

Lastly, we probed whether transcriptomic make-up, in the form of combinatorial DEG expression, could be sufficient to efficiently discriminate between acute vs. repeated ensemble cells. We trained support vector machine (SVM) classifiers to do so using levels of DEGs as features and individual GFP+ nuclei as observations, and they performed well (Supplementary Fig. 11a, b). This not only confirms the robustness of our DEG calls, but also suggests that such subtle, multi-

genic transcriptomic information might allow the capture of distinct neuronal ensembles at the functional level.

This initial snRNAseq analysis demonstrated that the activation of canonical striatal plasticity programs during cocaine-associated learning is largely restricted to cells that previously induced ARC expression, underscoring at the single cell level that ARC induction represents not only a marker of recent activity, but also a predictor of later neuronal plasticity. This dataset also dissects the plasticity-related, activity- and dopamine-dependent transcriptional signature unique to cocaine-activated NAc cells. Beyond these shared features, we show as well that this molecular signature can discriminate between ensemble cells according to the stage of encoding at which they were recruited (early/acute ensemble vs. late/repeated ensemble), and that these transcriptional differences are associated with distinct electrophysiological properties in the two ensembles.

## Acute and repeated ensembles exhibit lasting differences in their electrophysiological properties

The physiology-related transcriptional differences revealed by our snRNAseq data between acute and repeated ensemble cells suggested potential differences at the physiological level. We thus compared the electrophysiological properties of cells recruited in the acute vs. repeated ensemble in ArcCreERT2::Ai14 mice with ex vivo whole cell patch clamp recordings (Fig. 6a, b). Mice were habituated to a new context for 3 days, and then administered cocaine either acutely or repeatedly in this same context. Cocaine-recruited ensemble cells were permanently tagged (Tom+, red) via the concomitant injection of 4-OHT and cocaine during either the first and single (acute) or the fifth and last (repeated) cocaine exposure (Fig. 6a). Recordings from ensemble cells (Tom+) were performed on NAc slices at 7 days post

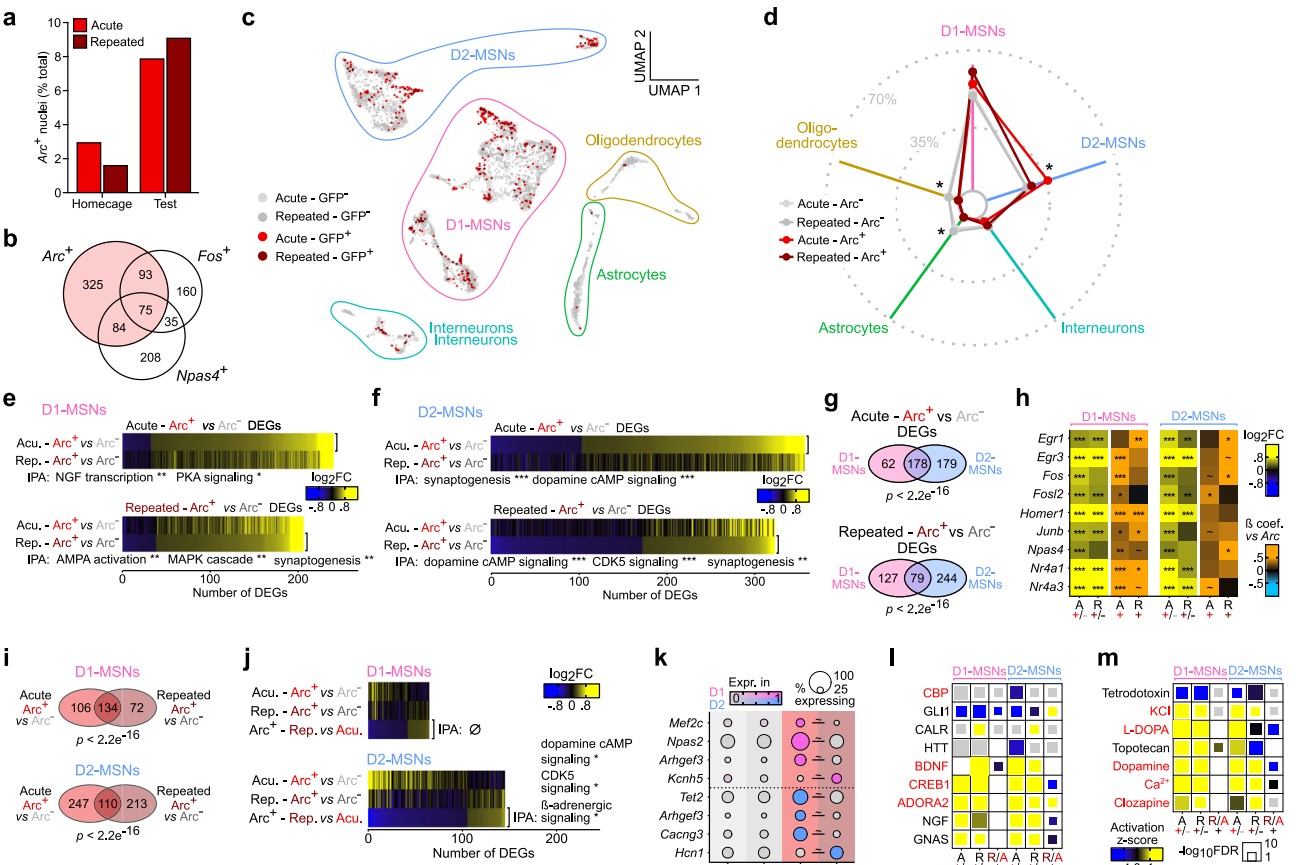

**Fig. 7 | Transcriptional correlates of recent *Arc* induction in acute and repeated ensembles upon cocaine-context memory retrieval. a** Number of nuclei with detectable *Arc* mRNA expression. **b** Overlap between *Arc*+ nuclei with *Fos*+ (Fischer exact test \*\*\**p* < 0.0001) and *Npas4*+ (Fischer exact test \*\*\**p* < 0.0001) nuclei. **c** UMAP reduction and cell type annotation of nuclei from CPP test samples only (*n* = 5,448). **d** Proportion of nuclei from each treatment/*Arc* status combination in each cell type cluster. Exact numbers are available in Supplementary Fig. 8i. $\chi^2$ tests: Acute: $\chi^2$ = 6².155, df = 4, \*\*\**p* < 0.001. Repeated: $\chi^2$ = 25.224, df = 4, \*\*\**p* < 0.001. FDR-adjusted *p*-values correspond to standardized Pearson's residuals. \*\*\**p* < 0.001 Arc+ vs Arc-, Acute; ##*p* < 0.01 Arc+ vs Arc-, Repeated. **e** Expression heatmaps of *Arc*+ vs. *Arc*- DEGs in D1-MSNs in acute (top) and repeated (bottom) conditioning groups, along with 2 or 3 most significantly enriched IPA canonical pathways (\*\*\* FDR < 0.001). Acu. Acute, Rep. Repeated. **f** Expression heatmaps of *Arc*+ vs. *Arc*- DEGs in D2-MSNs in acute (top) and repeated (bottom) conditioning groups, along with 2 or

3 most significantly enriched IPA canonical pathways (\*\*\* FDR < 0.001). **g** Overlap of *Arc*+ vs. *Arc*- DEGs between D1- and D2-MSNs in either acute (top) or repeated (bottom) conditioning groups (Fisher's exact test *p*-values). **h** Heatmaps of expression and correlation with Arc expression for selected IEGs in D1- (left) and D2-MSNs (right). **i** Overlap of *Arc*+ vs. *Arc*- DEGs between acute and repeated conditioning groups in D1- (top) and D2-MSNs (bottom) (Fisher's exact test *p*-values). **j** Expression heatmaps of DEGs in *Arc*+ nuclei between repeated and acute groups in D1- (top) and D2-MSNs (bottom), along with 2 most significantly enriched IPA canonical pathways (\* FDR < 0.05, \*\*\* FDR < 0.001). **k** Expression dot plots of selected DEGs in D1- and D2-MSNs (~ FDR < 0.1). **l** Most significant IPA predicted upstream regulator proteins across comparisons in D1- and D2-MSNs. **m** Most significant IPA predicted upstream regulator chemical substances across comparisons in D1- and D2-MSNs.

---

tagging to assess lasting changes in ensemble-specific electrophysiological properties.

ARC+ cells recruited in the acute and repeated ensembles exhibited similar intrinsic baseline membrane properties along with similar resting membrane potential and rheobase (Fig. 6c). We found that increasing steps of current evoked significantly more action potentials in neurons from the repeated ensemble as compared to neurons recruited in the acute ensemble (Fig. 6e, f), indicating an overall increased intrinsic excitability for repeated ensemble cells compared to acute ensemble cells. At the synaptic level, both the frequency and amplitude of spontaneous excitatory post-synaptic currents (sEPSCs) were unchanged between cells belonging to the acute vs. repeated ensemble (Fig. 6g–i).

These data—altogether consistent with recent similar work[46]—show that cells recruited in the cocaine ensemble exhibit divergent excitability profiles depending on their recruitment at early (acute) vs. later (repeated) stages of drug exposure. Such electrophysiological differences are likely to be supported at least in part by ensemble-

specific regulation of genes involved in homeostatic plasticity (e.g., voltage-gated ion channels, intracellular molecular cascades, etc.).

## Recent *Arc* induction denotes recall-evoked transcriptional regulation

As described above, the 1- vs. 5-pairing paradigms were associated with the expression of, respectively, weak or strong memory. We hypothesized that there are molecular correlates for such differences in memory strength, and asked whether the transcriptional programs engaged in recall-recruited cells would differ according to past cocaine history (1 vs. 5 pairing).

Cells recruited during the retrieval of cocaine-context associative memories were identified based on the detection of *Arc* mRNA within individual ensemble cells from CPP test samples only. Consistent with our previous findings (Supplementary Fig. 5), the number of *Arc*+ expressing nuclei was increased by memory recall (test session, Fig. 7a). These cells significantly overlap with ensembles defined by the expression of other IEGs like *Fos* and *Npas4* (Fig. 7b). With regard to

cell types, a majority of recall-activated ensemble cells are D1-MSNs, although in slightly lower proportions than the encoding-recruited ensembles (Fig. 7c, d). At the transcriptional level, cells recruited during recall (*Arc+*) were distinguished from non-recruited ones (*Arc-*) through the differential expression of numerous genes (Supplementary Data 4), largely involved in synaptic signaling and plasticity (e.g., synaptogenesis, dopamine cAMP signaling, MAPK cascade, Fig. 7e, f). A significant proportion of these DEGs were regulated in both D1- and D2-MSNs (Fig. 7g), illustrating that the transcriptional programs of recent *Arc* induction might exemplify a neuronal activation signature that goes beyond MSN subtype identity per se. Consistently, within each of these MSN populations, *Arc+* cells also expressed higher levels of numerous other IEGs compared to *Arc-* cells, and furthermore *Arc* expression levels within individual nuclei were strongly correlated with those of the other IEGs (Fig. 7h). Predicted upstream regulators of these gene sets implicated well-established activity-dependent transcriptional regulators, including CBP, as well as their likely dependence on dopamine signaling (Fig. 7l, m). This finding both confirmed the ability of our snRNAseq approach to robustly identify nuclei with recent transcriptional activation, as well as the fact that those IEGs are induced together, in largely—but not completely—overlapping sets of NAc MSNs, upon CPP memory retrieval—which up to now has remained a recurring question in the field. Furthermore, subclustering D1-MSNs revealed that, in contrast to cocaine-activated nuclei (Supplementary Fig. 10d, e), recall-activated transcription of IEGs was not selective to Drd1-MSN-1 cluster (although trends were observed, Supplementary Fig. 10f), suggesting that recall-induced transcriptional activation, including of *Arc*, depends on molecular and circuit processes likely distinct from those of cocaine-activated transcription during memory encoding.

Despite significant similarities (Fig. 7b, e, f, and g), these recall-induced (*Arc+*) cells displayed significant gene expression differences depending on ensemble membership (acute, 1-pairing vs. repeated, 5-pairing groups, Fig. 7j). This last analysis points towards gene regulation mechanisms like "priming" and "desensitization" at the level of individual genes within individual cells: for instance, we highlighted genes that were only induced in activated cells with (e.g., *Kcnh5*, *Hcn1*) or without (e.g., *Npas2*, *Arhgef3*, *Tet2*, *Cacng3*) a history of repeated cocaine exposure (Fig. 7k). Together, these data strongly support the idea that *Arc*-expressing ensemble cells support memory processes—including both memory encoding and retrieval—through activity-dependent plasticity- and memory-related transcriptional programs, and that these programs are modulated by past cocaine history (1 vs. 5 pairing).

### Recall-induced ARC ensemble reactivation is transcriptionally mediated

One remaining question was to investigate whether these ensemble transcriptional features might play a part in governing reactivation at the single-cell level, especially given that we have demonstrated distinct reactivation properties for ensembles encoding distinct phases of learning (acute/1-pairing vs. repeated/5-pairing, Fig. 3). We hypothesized that NAc cells recruited in the acute vs. repeated ensembles during the encoding of cocaine-context associative memories acquire some divergent cellular properties that modulate their subsequent reactivation upon recall and consequently their contribution to memory retrieval. We thus examined whether cells reactivated during recall (*Arc+/GFP+*) exhibit different transcriptional responses depending on their former recruitment status, and whether such responses differ between acute vs. repeated ensemble cells.

Again, reactivated cells (*Arc+/GFP+*) are predominantly D1-MSNs (Fig. 8a). Cells recruited during recall (*Arc+*) were significantly more likely to have been previously recruited in the encoding ensemble (GFP+), meaning that the recall ensemble preferentially recruited cells that were already recruited during encoding, except for D2-MSNs of the

repeated group (Fig. 8b). Such preferential re-recruitment of memory-encoding cells is consistent with the immunohistological observations above (Fig. 3d–f) and with properties inferred for engram-like cells.

We then examined genes that were differentially regulated selectively in reactivated cells—by comparing cells activated during both encoding and recall (GFP+/*Arc+*) to cells only activated once during either encoding (GFP+/*Arc-*) or recall (GFP-/*Arc+*), and subtracting from the resulting DEG lists any genes that can be considered as activity-regulated (Fig. 8c). Only a few genes survived this highly stringent selection process, for either D1- or D2-MSNs from either the acute or repeated ensemble (Fig. 8d), yet with the potential to influence neuronal function at the levels of both synaptic signaling and transcriptional regulation. We further computationally tested whether the expression of those few "reactivation" genes could efficiently predict the observed reactivation status with SVM classifiers (Supplementary Fig. 11a). These classifiers performed well (Supplementary Fig. 11c), suggesting that the subtle transcriptional differences that are highlighted here are likely sufficient to dictate reactivation at the ensemble level, and therefore represent a prime avenue for future research.

Finally, we formally examined gene expression differences between reactivated (GFP+/*Arc+*) cells from the repeated vs. acute ensembles (Fig. 8e and Supplementary Data 5) and found that "reactivation-predicting" genes appeared to also be DEGs in that last comparison (Fig. 8f). This analysis suggests subtly distinct, transcriptionally-modulated reactivation mechanisms in the acute and repeated ensembles, which could account for their distinct reactivation probabilities (Fig. 3), and in turn could modulate their contribution at different stages of memory encoding.

## Discussion

The results of this study demonstrate that expression of a cocaine-context memory by re-exposure to the context triggers the reactivation of a subset of NAc neurons that had been previously activated during exposure to the drug itself and memory encoding, that the extent of this reactivation correlates with memory strength, that their artificial reactivation can support memory retrieval, and that these neurons exhibit long-lasting, plasticity-related transcriptional adaptations. These subsets of neurons thus meet canonical, well-accepted criteria to be defined as functional memory "engram" cells[8,10,17], and are therefore likely to represent a cellular memory trace and substrate for cocaine-associated memories in NAc. This is similar to what has been described for glutamatergic pyramidal neurons in hippocampus-related learning[6,14,39], only we are showing it here for GABAergic MSNs in NAc. It is striking to also note that the reactivation rates of these NAc cocaine engram-like cells appear similar if not higher than what tends to be observed for hippocampal engrams in Pavlovian learning paradigms[6,39]. Yet, and similar to the aforementioned hippocampal studies, the respective memory encoding/retrieval ensembles contain additional, "non-engram" cells (i.e., activated but not reactivated Tom+/ARC- cells) that are less likely to directly support memory per se. Whether these result from non-specific tagging (from non-behaviorally relevant neuronal activity) or encode distinct behavioral sub-parameters (e.g., locomotion, motivation, behavioral state, etc.), and, in our case, if/how these differ between acute vs. repeated drug conditioning, remain to be determined. In particular, more finely distinguishing between neurons that encode memory recall ("engram" cells) and those that mediate extinction learning will be crucial, especially in a task like CPP testing where memory recall is experimentally not dissociable from the first instance of memory extinction.

We go on to show that, during a course of repeated cocaine exposures, the recruited NAc ensemble gets smaller but incorporates newly activated cells that were not recruited upon the first exposure. These findings establish that mostly distinct ensembles are recruited at later vs. early stages of cocaine-context learning. The differential

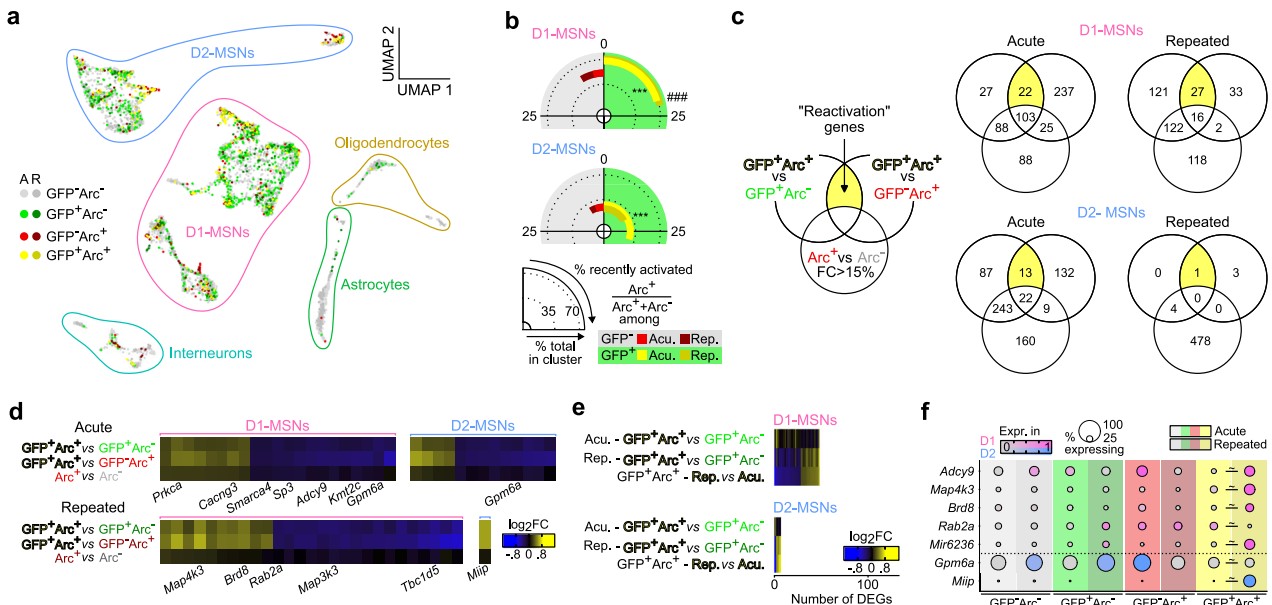

**Fig. 8 | Transcriptional correlates of Arc reactivation in cells recruited during both the encoding and retrieval of cocaine-context associative memories.**
**a** UMAP reduction and cell type annotation of nuclei from CPP test samples only ($n = 5,448$), colored according to treatment group, GFP FANS status, and *Arc* expression status. **b** Proportion of nuclei from each treatment/GFP/*Arc* status combination in D1-MSNs (top) and D2-MSNs (bottom), highlighting preferential recent *Arc* induction (*Arc+*) in remote *Arc*-activated (GFP+) nuclei (D1, acute: $\chi^2_{(1)} = 148.43$, $p < 2.2e^{-16}$; D1, repeated: $\chi^2_{(1)} = 23.067$, $p = 1.56e^{-6}$; D2, acute: $\chi^2_{(1)} = 102.77$, $p < 2.2e^{-16}$; D2, repeated: $\chi^2_{(1)} = 2.57$, $p = 0.1083$). ***$p < 0.001$ Arc+ vs Arc-, Acute; ##$p < 0.01$ Arc+ vs Arc-, Repeated. Exact numbers are available in Supplementary Fig. 8j. Acu. Acute, Rep. Repeated. **c** Schematic intersectional strategy to identify putative reactivation-related genes, i.e., genes that are DEGs between GFP+/*Arc+* and GFP+/*Arc-* nuclei as well as between GFP+/*Arc+* and GFP-/*Arc+* nuclei but are not activity-dependent, i.e., are not changed overall in *Arc+* vs. *Arc-* nuclei

(for added stringency, all genes with an *Arc+* vs. *Arc-* fold change >15% were removed, independent of FDR-based significance. Venn diagram representing the number of genes identified as DEGs with this intersectional strategy for each cell type (D1 vs. D2-MSNn) and each cocaine ensemble (acute vs. repeated). **d** Expression heatmaps of putative "reactivation" genes in acute (top) and repeated (bottom) ensembles in both D1-MSNs (left) and D2-MSNs (right). Select genes involved in neuronal signaling and transcription are annotated. **e** Expression heatmaps of DEGs in GFP+/*Arc+* reactivated nuclei between repeated and acute groups in D1-MSNs (top) and D2-MSNs (bottom), along with corresponding most significantly enriched IPA canonical pathways (* FDR < 0.05). **f** Expression dot plots of selected D1-MSNs and D2-MSNs repeated vs. acute DEGs involved in neuronal physiology, signaling, and transcriptional regulation ( - FDR < 0.1), that are also "reactivation" genes.

contribution of these distinct neuronal populations to memory recall, which we demonstrate by use of optogenetic approaches, suggests that they could encode distinct aspects of these associative memories. The repeated ensemble—smaller, more excitable, and the reactivation of which during memory recall correlates negatively with memory strength—could exert a homeostatic effect on NAc circuits during later stages of learning. Conceptually, such role for ARC in homeostatic processes at the ensemble level is consistent with the counter-intuitive idea that, at the cellular level, ARC is in fact induced in cells and synapses in part to downscale their contribution to synaptic plasticity (e.g., by promoting AMPA endocytosis), especially during later phases of long-term plasticity[49,51,52]. Similar interpretations have been proposed at the chromatin level, where ARC induction in the nucleus is also thought to act as a brake on transcriptional plasticity to support homeostatic regulation[42,49]. By contrast, our evidence shows that the acute ensemble drives memory formation and recall. Acute and repeated ensembles exist in brain coincidentally in time in response to a range of overlapping stimuli, and further work is needed to understand how their concomitant recruitment is integrated to determine behavioral responses.

Another intriguing question resides in the extent to which these Arc ensembles are overlapping or distinct from those formed by other IEGs in similar conditions, most of which have been implicated in cocaine responses[32] and plasticity in general[24,40]. While we show that other IEGs exhibit similar desensitization with repeated cocaine (Supplementary Fig. 1) at the bulk mRNA level, and significantly correlate with *Arc* induction upon CPP recall at the single-cell level (Fig. 7h), it remains to be determined how *Arc*, *Fos*, *Npas4*, etc.

ensembles emerge, coexist and cooperate at distinct stages of drug memory encoding and recall—especially in light of the existence of IEG-defined "superensembles"[31] in the striatum, and of the fact that CPP retrieval appears to induce *Arc*, *Fos* and *Npas4* in significantly but only partially overlapping ensembles (Fig. 7b). It is established that *Arc* is induced in many of the same cells as *Fos*, for instance, in simple stimulus-response situations (e.g., an acute injection of cocaine[42,46,56]). However, because of its paradoxical relationship with homeostatic plasticity (see above), *Arc* might stand as a unique marker for "homeostatic ensembles" and might thereby label functionally distinct ensembles or sub-ensembles in more complex memory-related paradigms—an hypothesis that remains to be tested experimentally.

Our snRNAseq dataset provides unprecedented insight into the transcriptional profiles of ensemble cells. We demonstrate that plasticity programs in NAc are largely restricted to cells that are functionally engaged in memory encoding and retrieval across cell types, and associate with *Arc* and other IEGs induction. By establishing the transcriptional correlates of ensemble allocation, we show that, beyond their activation at key phases of memory processes, ARC-defined ensemble cells also exhibit molecular features likely to support both their recruitment to the ensemble and their further contribution to memory storage through traditional plasticity mechanisms. The causal roles of individual genes, such as those identified in Fig. 8, alone or in combination, to engram allocation during drug memory encoding or engram reactivation during drug memory retrieval, thus stand as attractive avenues for follow-up work, which will complement emerging data on the molecular substrates of these processes in other memory circuits, especially in terms of transcriptional control[16,73].

Additionally, these data confirm that ARC expression efficiently stamps cell populations harboring transcriptional programs relevant to learning and memory. Future work is needed to establish which of these cellular adaptations are causally mediated by ARC induction. Nevertheless, our data accentuate at single-cell resolution that ARC represents an effective marker of both recent neuronal activation but also of the induction of larger, longer-lasting, plasticity programs.

Our snRNAseq dataset is limited by several, technically unavoidable caveats. Due to the scarcity of ARC⁺ GFP-tagged cells, it was necessary to pool multiple dissection samples before sorting and sequencing. This resulted in the loss of individual, sample-level behavioral data, which could have been used as a predictor for transcriptomics data, as well as prevented the use of more robust pseudo-bulking analytical methods. However, our conclusions were consistent with those derived from multiple other cell-level statistical frameworks, and we have also used an alternative sample-level random pseudo-sampling pseudo-bulking strategy[72] with overall consistent findings (Supplementary Data 6), together suggesting that the gene expression patterns we describe here are overall robust with respect to analytical and statistical power differences. Our approach also made it impossible to segregate core vs. shell NAc subregions, which play distinct circuit roles in reward-related behaviors[5,74]. Very recent technical advances, largely relying on the use of antibody-based multiplexing—which had been previously described[75] but had yet to be more widely adopted—have the potential of reliably enabling the capture of rare activity-defined target populations while retaining individual sample identity[76]. This updated methodology, abbreviated as XPoSE-seq, thus stands as an attractive and exciting platform for future transcriptional studies of activated ensembles with improved resolution and statistical power.

In conclusion, we here identify a shared molecular profile that distinguishes all ensemble-allocated cells—whether D1 or D2-MSNs in the acute as well as repeated ensembles. We also provide evidence for more subtle transcriptional regulation that is specific to cells recruited at different stages of learning (i.e., acute vs. repeated ensembles). This finding indicates that transcriptional regulation tracks memory formation and storage at the cellular level. This work thereby provides more general insight into the mechanisms by which neuronal ensembles are shaped at the molecular level to control memory, and underscores the relevance of studying drug addiction as a form of pathological memory from both a fundamental and translational standpoint.

## Methods

### Animals
C57BL/6 J mice were purchased from The Jackson Laboratory at 7 weeks of age and experiments performed at 8–10 weeks of age. ArcCreER^T2 mice (IMSR_JAX:022357) were a generous gift from Christine Denny[6] at Columbia University and bred in-house on a C57BL/6 J background. ArcCreER^T2 mice were crossed in-house to the following transgenic reporter mouse lines obtained from the Jackson Laboratory: Ai14 (IMSR_JAX:007914), Ai32 (IMSR_JAX:012569), and Sun1 (IMSR_JAX:030952). D1-tdTomato (MGI:4360387) were bred in-house. All mice were group-housed, maintained on a 12:12 h dark/light cycle (08:00 lights off; 20:00 lights on) and were provided with food and water *ad libitum*. All behavioral testing was conducted in the dark phase at 8–18 weeks of age. All mice were maintained according to the National Institutes of Health guidelines for Association for Assessment and Accreditation of Laboratory Animal Care accredited facilities, and all experimental protocols were approved by the Institutional Animal Care and Use Committee at Mount Sinai.

### Drug treatments
Cocaine HCl (from the National Institute on Drug Abuse) was diluted in 0.9% NaCl saline solution (ICU Medical) and injected intraperitoneally at 20 mg/kg. An aqueous formulation was used for 4-hydroxytamoxifen (4-OHT) delivery[49]. Briefly, 4-OHT (Sigma, H7904) was first dissolved in DMSO at 20 mg/mL, and then diluted to 1 mg/mL in sterile saline containing 2% Tween-80. Mice were injected intraperitoneally with this solution at 10 mg/kg. For controls, the vehicle consisted of a saline solution only containing 2% Tween-80 and 5% DMSO, and was injected at a similar volume than 4-OHT (0.1 mL/10 g body weight). All cage mates always received the same treatment.

### Activity-dependent tagging of ARC ensembles
Tagging of cocaine-activated cell populations was achieved with concomitant injection of cocaine and 4-OHT right before mice were placed into either the locomotor chamber (for locomotor sensitization) or the cocaine-paired chamber (for CPP). In both paradigms, mice were taken out of the chamber 1 h after the beginning of the behavioral task, put back into their homecages, and left undisturbed for 5 h to avoid any non-specific recombination and tagging.

### Context-dependent locomotor sensitization
Locomotor activity was measured in a clear plexiglass open field arena (40 × 40 × 30 cm) as the number of laser beam-breaks collected over the duration of the test. Mice were placed in the open-field arena for 30 min after injection of a saline solution during three consecutive days for habituation before the actual experiment was performed. The protocol of context-dependent locomotor sensitization consisted of 5 daily sessions of 30 min in which locomotor activity was recorded for 30 min after an injection of saline or cocaine. Locomotor sensitization was calculated as the ratio of locomotion on day 5 over the locomotion on day 1.

**For the study of Arc ensemble desensitization in C57BL/6 J mice.** After the three days of habituation, C57BL/6 J male mice were injected i.p. with either cocaine (20 mg/kg) or saline in the open-field arena. On days 1–4, mice received daily injections of saline ('Acute') or cocaine ('Repeated'). On day 5, mice from each group ('Acute' and 'Repeated') received either a saline injection ('Context only') or a cocaine injection ('Context + Cocaine') and tissue was collected 1 h after the last injection.

**For the study of Arc ensemble reactivation in Arc-CreER^T2 mice.** After the three days of habituation, Arc-CreERT2::Ai14 male mice were injected with either cocaine (20 mg/kg) or saline in the open-field arena. On day 1 mice were injected i.p. with 4-hydroxytamoxifen (4-OHT, 10 mg/kg) concomitantly with cocaine (20 mg/kg) in the open-field arena to achieve the tagging of cocaine-activated Arc ensembles. On days 2–4, animals received daily injections of saline (2-injections group) or cocaine (5-injection group). On day 5, mice were injected with either saline ('Context only') or cocaine ('Context + Cocaine') and tissue was collected 1 h after the last injection.

**For ex vivo electrophysiology study of Arc ensembles in Arc-CreER^T2 mice.** ArcCreERT2::Ai14 male mice were injected i.p. with cocaine (20 mg/kg) or saline in the open-field arena on one day or on five consecutive days and injected i.p. with 4-hydroxytamoxifen (4-OHT, 10 mg/kg) at the beginning of the last open-field session to achieve the tagging of cocaine-activated Arc ensembles. Tissue was collected at 7 days post tagging and acute ex vivo NAc slices were prepared before whole cell patch-clamp recordings of cocaine-activated Arc ensembles.

### Cocaine conditioned place preference
Unbiased cocaine conditioned place preference paradigm (CPP) was performed using a three-chambered CPP set-up[50]. Apparatus consisted of two chambers distinguished by distinct visual (gray vs. stripes walls) and tactile cues (small vs. large grid floors) separated by a small neutral

area. Locomotion and time spent in each chamber was measured using an overhead video-tracking system (Ethovision XT 11) set to localize the mouse center point at each time of the trial. On the pre-conditioning phase (pretest day), mice were placed in the neutral area and allowed to freely explore the three chambers for 20 min. During the conditioning phase, drug-context learning was achieved by pairing an injection of saline with one chamber in the morning and a second injection of cocaine with the other chamber in the afternoon for either a single day (1-Pairing, weak conditioning group) or five consecutive days (5-Pairing, strong conditioning group). After injections, mice were confined to a given chamber for a period of 45 min. Groups and pairing sides were assigned after pretesting to balance any pre-existing chamber bias. On the post-conditioning phase, CPP testing was carried out with each mouse allowed again to freely explore all the chambers for 20 min. The CPP score was calculated as the difference in time spent on the cocaine-paired chamber vs. the saline-paired chamber during post-conditioning vs. pre-conditioning.

**For the study of Arc ensemble reactivation in Arc-CreER$^{T2}$ mice.** Arc-CreER$^{T2}$::Ai14 male mice were conditioned with cocaine (20 mg/kg i.p.) on one day or on five consecutive days and injected i.p. with 4-hydroxytamoxifen (4-OHT, 10 mg/kg) at the beginning of the last conditioning session to achieve the tagging of cocaine-activated Arc ensembles. Mice were then tested 7 days later for their preference for the cocaine-paired chamber. NAc tissue was collected 1 h after the beginning of the test session.

**For the artificial reactivation of Arc ensembles in Arc-CreER$^{T2}$ mice.** Arc-CreER$^{T2}$::Ai32 male mice were conditioned with cocaine (20 mg/kg i.p.) on one day or on five consecutive days and injected i.p. with 4-hydroxytamoxifen (4-OHT, 10 mg/kg) at the beginning of the last conditioning session to achieve the tagging of cocaine-activated Arc ensembles. Mice were then tested 7 days later for their preference for the cocaine-paired chamber with the laser off and then on. NAc tissue was collected 1 h after the beginning of the test session.

**For the transcriptional profiling of Arc ensembles in Arc-CreER$^{T2}$ mice.** Arc-CreER$^{T2}$::Sun1 male and female mice were conditioned with cocaine (20 mg/kg i.p.) on one day or on five consecutive days and injected i.p. with 4-hydroxytamoxifen (4-OHT, 10 mg/kg) at the beginning of the last conditioning session to achieve the tagging of cocaine-activated Arc ensembles. Mice were then tested 7 days later for their preference for the cocaine-paired chamber. NAc tissue was collected 30 min after the beginning of the test session. Bilateral dissections from 16–20 mice were pooled together group-wise into one single sample and processed for snRNAseq.

### RNA extraction and quantitative real-time PCR
Mouse brains were collected after cervical dislocation and followed by rapid bilateral NAc punch dissections from 1 mm-thick coronal brain sections using a 14 G needle and frozen on dry ice. RNA extraction was performed using the RNeasy Micro Kit (Qiagen) following manufacturer instructions. RNA 260/280 ratios of 2 were confirmed using spectroscopy, and reverse transcription was achieved using the iScript cDNA Synthesis 385 Kit (BioRad). Quantitative PCR using PowerUp SYBR Green (Applied Biosystems) was used to quantify cDNA using an Applied Biosystems QuantStudio 5 system. Each reaction was performed in triplicate and relative expression was calculated relative to the geometric average of 3 control genes (*Ppia*, *Tbp*, *Rpl38*) according to published methods[50,51]. Sequences of primers are available in Supplementary Data 1.

### Immunohistochemistry (IHC)
At the indicated times after drug treatment or behavioral task (see figure legends), mice were rapidly anesthetized with an intraperitoneal

injection of pentobarbital (50 mg/kg, Fatal Plus, Vortex Pharmaceutical) and euthanized via transcardiac perfusion with a fixative solution containing 4% paraformaldehyde (PFA) (v/v) in 0.1 M phosphate buffer saline (PBS) delivered with a peristaltic pump at 6 mL/min for 5 min. Brains were removed from the skull and post-fixed for 24 h in a 4% PFA solution at 4 °C. Sections of 30 μM thickness were cut in the frontal plane with a vibratome (Leica) and kept at -20 °C in a cryoprotectant solution containing 30% ethylene glycol (v/v), 30% glycerol (v/v), and 0.1 M phosphate buffer (PB), Sections were then processed for IHC. Briefly, on day 1, sections were washed three times in 0.1 M phosphate buffer saline (PBS) and permeabilized for 15 min in a solution containing 0.2% Triton X-100 (Sigma Aldrich) in PBS. After three rinses in PBS, sections were incubated 1 h at RT in a blocking solution containing 10% Donkey Serum (v/v) in 0.1 M PBS. Sections were then washed three times in PBS and incubated overnight at 4 °C with the primary antibodies diluted in the blocking solution. The following primary antibodies were used either individually or in combination: ARC was detected using a rabbit polyclonal antibody raised against ARC (1/500, Synaptic System, 156-003), td-Tomato was detected using a goat polyclonal antibody raised against RFP (1/1000, Rockland, 200-101-379) and EYFP was detected using a polyclonal goat raised against GFP (1/1000, Aves Lab, GFP-1020). Following primary antibodies incubation, sections were rinsed three times in PBS and incubated 90 min at RT with adequate combination of the following secondary antibodies: donkey anti-rabbit Alexa-488-conjugated (1/500, Jackson Immunoresearch), donkey anti-goat Rhodamine Red (Jackson Immunoresearch) and donkey anti-chicken Alexa-488-conjugated (1/500, Jackson Immunoresearch). After three rinses in PBS, sections were incubated 10 min in DAPI for nuclei counterstaining before being rinsed three times in PBS, three times in PB and mounted in ProLong Diamond Antifade mounting medium (ThermoFisher Scientific).

The following antibodies were used for immunohistochemistry: ARC was detected using a rabbit polyclonal antibody raised against ARC (1/500, Synaptic System, 156-003), td-Tomato was detected using a goat polyclonal antibody raised against RFP (1/1000, Rockland, 200-101-379) and EYFP was detected using a polyclonal goat raised against GFP (1/1000, Aves Lab, GFP-1020).

### RNA fluorescent in situ hybridization (FISH)
At the indicated times after drug treatment or behavioral task (see figure legends), mice were rapidly anesthetized with an intraperitoneal injection of pentobarbital (50 mg/kg, Fatal Plus, Vortex Pharmaceutical) and euthanized via transcardiac perfusion using a fixative solution containing 4% paraformaldehyde (PFA) (v/v) in 0.1 M phosphate Buffer Saline (PBS) delivered with a peristaltic pump at 6 mL/min for 5 min. The brain was removed from the skull and tissue was post-fixed overnight in a 4% PFA solution and stored at 4 °C. Sections of 30 μM thickness were cut in the frontal plane with a vibratome (Leica, Nussloch, Germany) and kept at -20 °C in a solution containing 30% ethylene glycol (v/v), 30% glycerol (v/v), and 0.1 M phosphate buffer. NAc slices were mounted on Superfrost Plus microscope slides (Fisher Scientific) and processed for RNA FISH using RNAscope Multiplex Fluorescent Reagent Kit v2 (ACD Bio) according to manufacturer instructions using mouse probes for *tdTomato* (tdTomato, #317041) and *Drd1a* (Mm-Drd1a-C2, #406491-C2) transcripts. Sections were counterstained with DAPI and mounted in ProLong Diamond Antifade mounting medium (ThermoFisher Scientific).

### Image acquisition
For both IHC and FISH, images were acquired in a 16-bits range using a laser scanning confocal microscope (SP8, Leica) with a 40X oil immersion objective (Zoom 0.9, pixel size: $x = 0.36$ μm, $y = 0.36$ μm). Images were acquired in frame mode with a frame size of 1024 × 1024 pixels. Acquisition settings (laser intensities and gain) were kept identical across samples. Signal was quantified in NAc core and medial

shell along the rostro-caudal axis between AP + 1.7 mm and AP + 1.1 mm for total of 12-18 different images (363.64 × 363.64 μm) per animal (i.e., 2–3 images per side, on three adjacent sections, for a total of ~2 mm² of tissue imaged per animal). Images were acquired by an experimenter blind to the group, removing any potential bias from region-of-interest selection.

## Image quantification

For IHC, the number of ARC-, tdTomato-, and/or EYFP-positive cells was manually counted in their respective channels using the "cell counter" plugin in Fiji. For each image, the total number of cells was quantified in the DAPI channel using a published custom pipeline[42]. For single-channel quantifications, absolute numbers of ARC-, tdTomato-, and/or GFP-positive cells were normalized to the combined area imaged for each animal, and thus expressed as cells/mm². For dual-channel quantifications (i.e., analyses of overlap or reactivation), the number of double positive cells was assessed by overlapping cell counter markers from each individual channel, and overlap/reactivation rates were calculated using the formulas below following recent methods[77]:

$$\text{Overlap/Reactivation("Recall")} = \frac{\text{ARC} + /\text{tdTomato} + \text{cells}}{\text{tdTomato} + \text{cells}} \times 100 \quad (1)$$

$$\text{Overlap/Reactivation("Precision")} = \frac{\text{ARC} + /\text{tdTomato} + \text{cells}}{\text{Arc} + \text{cells}} \times 100 \quad (2)$$

Finally, as a first step to correct for varying sizes of each respective ensembles, we computed F1-score as the harmonic mean of Precision and Recall:

$$F_1 - \text{score} = 2 \times \frac{\text{Precision} \times \text{Recall}}{\text{Precision} + \text{Recall}} \quad (3)$$

Further, we introduced another statistical framework to account for the individual (animal-wise) chance levels that originate in such nested designs. We calculated reactivation levels over "chance" reactivation (i.e., the likelihood of double-positive cells given the respecive number of single-positive cells for each channel among the total number of cells simply explained by random re-sampling), while accounting for the fact that these "chance" reactivation levels were unique to each animal (i.e., a nested design where individual cells are considered within individual animals). Inspired by reference work on similar experimental designs[54], animal-wise count data were treated as categorical with 4 outcomes (ARC-/tdTomato-, ARC-/tdTomato+ , ARC+/tdTomato-, or ARC+/tdTomato+), and baseline-category logit models were fitted for these multinomial counts to estimate chance-adjusted log-odds for each outcome for each combination of group membership (2 × 2 design) using the *mclogit::mblogit* function and including a random effect for nesting. Estimated probabilities were then back-calculated from log-odds estimates. 95% confidence intervals for these estimated probabilities were computed by Monte-Carlo simulation, simulating 10,000 repeated draws from a multivariate-normal sampling distribution whose parameters are given by the logit model (logistic regression coefficients and variance-covariance matrix) with *MASS::mvrnorm*. Finally, group-wise probability estimates were compared with Sidak's *post hoc* tests using *emmeans::emmeans*. These analyses were run in R v4.2.2. Custom R code and full model statistics are available upon request.

For FISH, the number of *tdTomato*- and/or *Drd1*-positive cells was manually counted in their respective channels using the "cell counter" plugin in Fiji[52]. The number of double positive cells was assessed by overlapping cell counter markers from each individual channel, and D1

enrichment was calculated as:

$$\%\text{D1} = \frac{\text{Drd1} + /\text{tdTomato} + \text{cells}}{\text{tdTomato} + \text{cells}} \times 100 \quad (4)$$

## Nuclei purification and fluorescence-activated nuclei sorting (FANS)

Mouse brains were collected after cervical dislocation and bilateral NAc dissection was rapidly performed from 1 mm-thick coronal brain sections using a 14 G needle and tissue frozen on dry ice. Bilateral dissections from 16–20 mice were pooled together group-wise into one single sample, which did not allow for the use of pseudobulking analysis strategies (see below). Nuclei suspension was obtained by homogenization of frozen pooled NAc samples in 4 mL of low-sucrose lysis buffer (0.32 M sucrose, 5 mM CaCl₂, 3 mM Mg(Ace)₂, 0.1 mM EDTA, 10 mM Tris-HCl) using a large clearance then a small clearance pestle of a glass Dounce tissue grinder (Kimble Kontes). The homogenates were filtered in an ultracentrifuge tubes (Beckman Coulter) through a 40 μm cell strainer (Pluriselect) into ultracentrifuge tubes (Beckman Coulter), underlaid with 5 mL of high sucrose solution (1.8 M sucrose, 3 mM Mg(Ace)₂, 1 mM DTT, 10 mM Tris-HCl). After centrifugation at 107,000 g for 1 h at 4 °C in a SW41Ti Swinging-Bucket Rotor (Beckman Coulter), the supernatant was discarded, and nuclei pellets were resuspended in 800 μL of a solution containing 0.5% bovine serum albumin (BSA) in PBS and supplemented with DAPI at 1 μg/mL to allow for nuclei detection. Nuclei were sorted on a BD FACS Aria II three-laser device with a 100 μm nozzle and using BD FACSDiva Software v8.0.2. Gating strategies from a representative sort are visualized in Supplementary Fig. 7. Briefly, debris and doublets were excluded using FSC and SSC filters, nuclei were then selected as DAPI-positive (Violet1-A laser) events, and finally GFP-positive 610 nuclei (Blue1-A laser) were sorted directly into BSA-coated low-binding tubes. 15–18,000 nuclei were recovered for GFP+ samples and 100,000 for GFP- samples.

## Single-nuclei RNA-sequencing (snRNAseq)

Following FANS, nuclei were quantified (Countess II, Life Technologies) and ±5000 per GFP+ sample and ±20,000 per GFP- sample were loaded on a single 10X flow cell GEM well using Chromium Single Cell 3' Library Construction Kit (10X Genomics). cDNA libraries were prepared according to the manufacturer's protocol (10X Genomics). Libraries were sequenced at Azenta on a single NovaSeq lane (Illumina) at a depth of ± 350 million reads per sample. Sequential analysis steps are described in detail below, and all custom code and scripts are available upon request.

**snRNAseq preprocessing.** A Cell Ranger (v7.0.0) reference package was generated from the mm10 pre-mRNA mouse genome (GRCm38_v5) that ensured alignment to unspliced pre-mRNAs and mature RNAs. Cell Ranger filtered outputs were analyzed with Seurat v4.3.0 in R v4.2.2. Nuclei containing <900 reads, or <200 or >5000 features (i.e., genes for which at least one read was detected), or >1% of reads mapping to the mitochondrial genome were removed, leaving altogether 12,092 nuclei with a median 2836 reads per nuclei for further analysis similarly to other previously published datasets[56,57]. Analysis was performed following Seurat v4.3.0 vignette with minor modifications. First, raw UMI counts were normalized using *SCTransform*, at which time the percent of mitochondrial contamination was additionally regressed out. SCT-normalized counts were used for all subsequent integration, differential expression and visualization. After *PrepSCTIntegration*, nuclei from all samples then underwent integration using 3000 features for *FindIntegrationAnchors*.

**snRNAseq clustering.** Clustering was performed using 16 principal components and 20 nearest neighbors for *FindNeighbors* and a 0.1 resolution value for *FindClusters*. These values were determined to recapitulate previously defined cell types[56,57]. UMAP dimensionality reduction was finally run with *RunUMAP* on the *integrated_snn* graph calling the *r-reticulate* Python v3.6.10 install of *umap-learn v0.4.6* for visualization purposes. Libraries were processed using *PrepSCTFindMarkers*, and marker genes for each cluster were computed with *FindAllMarkers* regressing out sample identity using logistic regression. Individual clusters were then further manually annotated by comparing enriched marker genes for each cluster (Supplementary Fig. 8) with publicly available single-cell RNAseq databases of NAc tissue[56,57]. Respectively 3 clusters of D1-MSNs and 2 clusters of D2-MSNs were manually combined together, and marker genes for each cell type cluster were computed with *FindAllMarkers* again (full marker gene lists and statistics are shown in Supplementary Data 2). In a parallel analysis, the 3 individual subclusters of D1-MSNs were compared to a recently published dataset focusing on D1-MSNs subclusters[57] (Supplementary Fig. 10). One cluster of 553 glutamatergic nuclei was assumed to result from dissection contamination from the piriform cortex just anterior to NAc (which itself does not contain glutamatergic neurons), and corresponding nuclei were thus removed from further analysis. Libraries from the remaining 11,539 nuclei with a median 2782 reads per nuclei were re-normalized using *PrepSCTFindMarkers* again.

**snRNAseq differential expression.** Cluster-specific pairwise differential expression analysis (a schematic of all comparisons is shown in Fig. 5b) was performed using *FindMarkers* on SCT normalized counts without the use of pseudobulking because biological replicates (16–20 mice per group) were pooled into one library per group. Fold change was computed for all genes, but statistical testing using logistic regression for differentially expressed genes (DEGs) was further restricted to genes detected in >30% of nuclei in at least one of the two groups in the corresponding pairwise comparison, and with >15% expression change between groups. *p*-values were adjusted for false discovery rate (FDR) at a 0.1 significance level for such exploratory purposes (full DEG lists and statistics are provided in Supplementary Data 3–5). For comparative purposes, we also ran differential expression using other cell-level statistical frameworks commonly used for mouse brain single-nuclei/single-cell transcriptomics like Wilcoxon Rank-Sum testing[63–68], negative binomial testing[69], and MAST[63,70,71], as well as random pseudo-sampling (*n* = 4 pseudo-samples with gene expression data summarized using *Seurat::AggregateExpression*) followed by multifactorial DESeq2 (design: ~ ArcGFP+.vs.ArcGFP- * Repeated.5 P.vs.Acute.1 P) similar to recently published work[72]. Comparisons between analysis methods are summarized in Supplementary Data 6. A given nucleus was considered *Arc*+ if it contained at least one UMI mapping to *Arc* transcript (Supplementary Fig. 8i), similar to published methods for ensemble definition in single-cell RNAseq[78]. *Fos* + and *Npas4*+ nuclei were defined the same way. For comparisons between *Arc*+ and *Arc*- nuclei (Fig. 7), the FANS status of individual nuclei (GFP+/-) was regressed out as a latent variable in logistic regression to diminish its influence on differential expression. Within *Arc*+ nuclei, correlations between individual *Arc* counts and those of other transcripts was computed using linear regression (*stats::lm*), regressing out FANS status again, and signed ß coefficient estimates were extracted to determine the strength of the association between *Arc* and other genes' expression. Preferential reactivation (Fig. 8) was assessed via $\chi^2$ testing of group-wise FANS-GFP status x *Arc* status contingency tables using *stats::chisq.test*. Enrichment analyses of gene expression patterns (significant DEG lists) was then performed using the Ingenuity Pathway Analysis (IPA, Qiagen) knowledgebase and classification system, extracting both enriched canonical signaling pathways and predicted upstream regulators significant by Fisher's enrichment test statistical testing and FDR correction. All IPA enrichment analyses were performed against cell-type-specific "background/ universe" gene lists which comprised all genes detected in either D1-MSNs (16,265 genes) or D2-MSNs (15,854) to reduce false-positive calls.

**snRNAseq classifiers.** Supervised, non-probabilistic binary linear classification using support vector machines (SVM) was run on gene expression matrices with individual nuclei as observations, select DEG panels as features and group membership (e.g., acute vs. repeated, or reactivated vs. non-reactivated) as binary outcomes. For controls ("Shuffle"), a list of genes of the same size (same number of gene as in the corresponding DEG list) was randomly extracted from the entire list of genes that were not significant DEGs in the respective comparison. Datasets (e.g., GFP+ nuclei from either acute or repeated ensembles, or GFP+ ARC+ vs. ARC- nuclei) were first randomly split into training (75%) and testing (25%) sets. SVM model training was achieved on the training set using C-type classification, a linear kernel and 5-fold cross validation with cost tuning in R v4.2.2 using the *e1071 v1.7::tune.svm* function. The trained model was then presented with testing set data, and SVM-predicted outcomes were compared to testing set observed outcomes to compute accuracy and $F_1$-score using the formulas detailed in Supplementary Fig. 11. $F_1$-scores were favored when outcome frequency was heavily biased. The operation was repeated 10 times with a new random split of training/testing sets to perform statistical comparisons against "shuffle" controls, which represent the classifier's performance if using random, non-DEG gene expression data.

## Optogenetics

Mice were anesthetized with an intraperitoneal bolus of ketamine (100 mg/kg) and xylazine (10 mg/kg), then head-fixed in a stereotaxic apparatus (Kopf Instruments). Then, 200-μm-wide optical fibers (Doric, MFC_200/240-0.22_4.5mm_MF1.25_FLT) were bilaterally implanted above NAc at AP + 1.4 mm, ML + 1.5 mm, DV −4.0 mm, at a 10° angle. Optical fibers were secured in place using dental cement (3 M) without the use of screws to the skull and covered with a layer of black dental cement (C&B Metabond). Mice were allowed >3 weeks to recover before experimentation. For optogenetic stimulation, mice were connected to a dual optical patch-cord (Doric) connected to a 473 nm blue laser (OEM Laser System). Stimulation was executed as trains of 10 box pulses (15 ms, 8–10 mW) emitted at 20 Hz every 5 s during the entire 20 min CPP test session. Individual animals were tested both with and without light illumination in a within-subject design.

## Ex vivo whole-cell patch-clamp recordings

Mice were anesthetized using isoflurane. Brains were rapidly extracted, and coronal sections (250 μm) were prepared using a Compresstome (Precisionary Instruments Inc.) in cold (0–4 °C) sucrose-based artificial cerebrospinal fluid (SB-aCSF) containing 87 mM NaCl, 2.5 mM KCl, 1.25 mM $NaH_2PO_4$, 4 mM $MgCl_2$, 23 mM $NaHCO_3$, 75 mM sucrose, 25 mM glucose. After recovery for 60 min at 32 °C in oxygenated (95% $CO_2$/ 5% $O_2$) aCSF containing 130 mM NaCl, 2.5 mM KCl, 1.2 mM $NaH_2PO_4$, 2.4 mM $CaCl_2$, 1.2 mM $MgCl_2$, 23 mM $NaHCO_3$, 11 mM glucose, slices were kept in the same medium at room temperature for the rest of the day and individually transferred to a recording chamber continuously perfused at 2–3 mL/min with oxygenated aCSF. Patch pipettes (4–6 MΩ) were pulled from thin wall borosilicate glass using a micropipette puller (Sutter Instruments) and filled with a K-gluconate (KGlu)-based intra-pipette solution containing 116 mM KGlu, 20 mM HEPES, 0.5 mM EGTA, 6 mM KCl, 2 mM NaCl, 4 mM ATP, 0.3 mM GTP (pH 7.2). Cells were visualized using an upright microscope with an IR-DIC lens and illuminated with a white light source (Olympus for Scientifica), and fluorescence visualized through a mCherry bandpass filter upon LED illumination through the objective (p3000ULTRA, CoolLed) using MicroManager v2.0 (https://micromanager.org/). All recordings were made on anterior NAc core MSNs. Excitability was

measured in current-clamp mode by injecting incremental steps of current (0–300 pA, +20 pA at each step). For recording of spontaneous Excitatory Post-Synaptic Currents (sEPSCs), neurons were recorded in voltage-clamp mode at -70 mV and sEPSCs detected with a 8 pA threshold. Whole-cell recordings were performed using a patch-clamp amplifier (Axoclamp 200B, Molecular Devices) connected to a Digidata 1550 LowNoise acquisition system (Molecular Devices). Signals were low pass filtered (Bessel, 2 kHz) and collected at 10 kHz using Axon pCLAMP 11 Software Suite (Molecular Devices). Electrophysiological recordings were extracted using Clampfit (Molecular Devices). All groups were counterbalanced by days of recording and all recordings were performed blind to experimental conditions. Control recordings used naive D1-tdTomato mice, where the selection of patched neurons followed a strict 70% D1+ / 30% D1- ratio to mimic the cell type composition of the Tom+ ensembles (Supplementary Fig. 9).

Statistical analysis required linear mixed models (LMM) to account for nested design (non-independent neurons nested within independent animals) and with repeated measures for excitability curves. These mixed effects models were computed in R v4.2.2 using *lmerTest::lmer* function similar to published work[79]. Random effects (repeated measures and/or nested observations) were modeled as random intercept factors. Subsequent analysis of variance was performed using type III sums of squares with Kenward-Roger's approximation of degrees of freedom. *Post hoc* testing was performed using the *emmeans* package and significance was adjusted using Sidak's correction.

## Statistics

No statistical power estimation analyses were used to predetermine sample sizes, which instead were chosen to match numerous previous publications[42,56,59,79]. Unless specified otherwise above (multinomial logistic regression, electrophysiology and snRNAseq), all statistics were performed in GraphPad Prism v9. In summary, pairwise comparisons were performed with Welch's *t*-tests, correlations using Pearson's *r* and multifactorial designs were analyzed with ANOVAs with repeated measures when appropriate. Pairwise post-hoc testing was adjusted using Sidak's correction. Bar and line graphs represent mean ± sem. Correlation graphs represent regression line with its 95% confidence interval. All *p*-value calculations were two-sided. Significance was set at $p < 0.05$, except for snRNAseq where cut-off was set at FDR < 0.1 as described above. For all post hoc visualization, $*p < 0.05$, $**p < 0.01$, $***p < 0.001$.

## Data availability

All raw and processed snRNAseq data reported in this study are deposited in the Gene Expression Omnibus under accession number GSE289612. Source data are provided with this paper.

## Code availability

Custom R scripts and code utilized in this study, including for statistical analysis, are available upon request.

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

## Acknowledgements

The authors would like to thank Stephen Pirpinas, Katherine Beach, Catherine McManus, Kyra Schmidt, Nathalia Pulido, and Ezekiell Mouzon for transgenics breeding and genotyping, and Dr. Edgardo Aritzia from the Dean's Flow Cytometry CoRE at the Icahn School of Medicine at Mount Sinai for his assistance with nuclei sorting. The authors would like to thank as well Dr. Nikos Tzavaras for his assistance with confocal image acquisitions that were performed at the Microscopy and Advanced Bioimaging CoRE at the Icahn School of Medicine at Mount Sinai. This work was supported by grants from the National Institutes of Health (R01DA014133 and P01DA047233 to EJN and U01-MH116442, R01-MH109677, R01-AG050986, R01-AG067025, R01-AG065582, and R01-AG082185 to PR).

## Author contributions

M.S. and E.J.N. conceived the project. M.S. developed all methodology with the help of A.G. M.S. performed all behavioral experiments, tissue preparation, confocal imaging, and nuclei sorting with the help of A.G., Y.X., A.R.L and L.M.H. M.S. and A.G. performed images analysis. A.G. performed snRNAseq analysis. R.D.C. performed electrophysiological recording under the supervision of S.J.R. J.F.F. performed snRNAseq library preparation. P.R. contributed resources for snRNAseq and software for analysis. M.S. and A.G. performed all formal analysis and visualization. M.S. wrote the manuscript which was reviewed and edited by A.G., R.D.C., L.M.H., S.J.R. and E.J.N.

## Competing interests

The authors declare no competing interests.
