## [Transparent Peer Review file · Nature Communications]

Cocaine-context memories are transcriptionally encoded in nucleus accumbens Arc ensembles.

Corresponding Author: Dr Eric Nestler

Version 0:

Reviewer comments:

Reviewer #1

(Remarks to the Author)

Understanding how drug-context learning is processed in the brain is an important topic. The present manuscript uses activity-dependent labeling to tag neurons in the nucleus accumbens during cocaine-context pairing (either with 1 or multiple pairings). First, they find that the ensemble of engram neurons active during training are reactivated when mice are placed back in the context drug-free. This is in line with multiple engram ensemble studies examining this and other types of memory. However, the current manuscript goes on to show differences between acute or more chronic pairings and then examined the transcriptional profiles of these neurons.

This manuscript comes from a leading group exploring substance abuse and describes the results of quite a bit of work, particularly with the transcriptional profiling, with very intriguing data.

I am impressed by this manuscript overall, and find the data foundational. I believe that others will be able to build off this dataset for further studies. I have only a few comments that the authors might chose to address in the discussion section. I frankly do not see the need for further experiments or analysis as i believe the results represent a unique and intriguing dataset and story.

1. Compared to the acute condition, repeated cocaine exposure resulted in a smaller engram (fewer Arc+ cells), with lower reactivation of cells labeled during the initial injection, lower correlation with cocaine-induced locomotion, and robust memory retrieval in a cocaine place preference paradigm. At the same time, optogenetic activation of the “repeated pairing” engram does not induce memory retrieval, unlike the acute pairing condition. The authors propose that the repeated ensemble serves as a homeostatic mechanism, particularly as reactivation appears to be correlated with a lower CPP score. This is an intriguing hypothesis and, as a reader, I would have loved a more thorough discussion of this idea.

2. Another possibility to explain these data stems from the field’s realization that neuronal activity tagging techniques can sometimes “overtag” true engram neurons. The present manuscript finds that acute cocaine activates a large number of NAcc neurons; many of these neurons could regulate cocaine-induced locomotion and a minority could be considered true fundamental “engram neurons”. It is only with repeated injections that a discrete cocaine-context engram forms and “locomotion” neurons desensitize. This might explain the differences in CPP retrieval, the differences in biophysical properties and why there is relatively little overlap between the two conditions. I imagine there could be a clever way to test this experimentally, but I believe that such an experiment would be beyond the scope of the present manuscript.

3. The analysis of transcripts in Arc+ and Arc- neurons is important data and suggests heterogeneity in plasticity-related transcript changes that are related to the function of specific ensembles. Ideally, the authors would examine the relationships between specific transcripts (or combinations thereof) and ensemble function that is linked to specific aspects of memory acquisition or retrieval. Again, this experiment is beyond the scope of the present manuscript, but the authors could note it as a limitation in the discussion section.

Reviewer #2

(Remarks to the Author)

Salay and colleagues examined the engagement and reactivation of neuronal ensembles in the nucleus accumbens (NAc)

using cocaine-conditioned place preference (CPP)—a task with all the features of a Pavlovian associative learning task. The approach involved taking advantage of the Arc immediate early gene as a marker for neuronal activation, which has been used in the learning and memory field to mark recently activated neuronal ensembles in the hippocampus. Here, the authors used ArcCreER mice to tag and examine the activity of the neural populations engaged during acquisition and during context induced recall, and report that populations identified during acquisition are also active during context induced recall. They further report that different ensembles in early vs late acquisition are distinct, with distinct impacts on memory retrieval. Ensembles were isolated to study using single nucleus RNA sequencing, which revealed plasticity-related transcriptional programs that segregate cocaine-recruited NAc engram cells unique to distinct stages of memory. In addition, optogenetic-mediated neuronal activation of the acute, but not the repeated, ensemble elicited a higher CPP score during the test session. Overall, these findings are highly significant because they further our understanding of cocaine-associated memory formation, which is important as these memories contribute to pathological relapse events.

Main Comments:

The foundation of this paper is based on the immediate early gene Arc. Arc is an unusual immediate early gene—usually referred to as a super-immediate early gene because of transcriptional pausing leaving Arc already partially transcribed and ready for a stimulus to release the pause (Saha et al., Nat Neuro 2011). Thus, would similar results be observed with a different marker of neuronal activation? Or does the potential unique nature of Arc transcription make it an exceptional tool for this study? Could the authors please comment on this in the discussion?

What would the authors expect to see in terms of ensembles if, in the acute model, the second cocaine injection were given on day 2 instead of day 5? This question relates to Arc transcription and cycling, and data represented in Figure 1. Are the differing rates of reactivated ensembles a product of time rather than number of cocaine exposures or phase of acquisition? This is better accounted for with the experimental design described in Figure 2, but in the case where overall Arc expression is decreased after repeated injections (Fig 1b) leading to smaller ensembles (Fig 2c), it is perhaps not surprising that the overlap is smaller during context-induced reactivation (Fig 2d).

The paradoxical finding that the recall activation more closely resembles the acute ensemble than the repeated ensemble is not entirely addressed by the optogenetic experiment. For example, after several days of cocaine conditioning, the first test day is also the first day of extinction. At this time point, the repeated conditioning mice are arguably engaged in extinction learning as well as recall, involving disparate ensembles.

Conversely, the acute ensemble may more resemble the recall activation ensemble because of novelty, and total distance moved induced by novelty. Is it possible that these acute ensembles are also more associated with locomotor movement? The distance moved for the CPP experiments should be reported in order to understand whether this is a component worth analyzing as well.

The process for selecting regions of interest to quantify Arc expression and subsequent ensembles is described in the methods: “Signal was quantified in NAc core and medial shell along the rostro-caudal axis for total of 12-15 different images (363.64 x 363.64 μm) per animal (i.e., 1 image per structure, per side, on three adjacent sections).” Please clarify this process, visually (a schematic) or verbally (more detailed methods). With the relatively sparse Arc signal and the anatomically vague representative images included in Figures 1 and 2, it is difficult to understand how bias was removed from the ROI selection process, image selection process, image analysis process, etc. For example, it seems as if only part of the NAc has been analyzed, and if that is the case – is it the same part for each brain, if so why was this section selected? Is the anatomy of cocaine ensembles conserved across different mice?

Similarly, it seems like the shell and core of the accumbens were considered collectively for most of this paper. Given their differing roles in acquisition of cocaine associated behaviors, and the technical ability to separate these areas can the authors please comment on this combined approach?

Line 294 states: “This dataset also dissects the plasticity-related, activity- and dopamine-dependent transcriptional signature unique to cocaine-activated NAc cells. Beyond these shared features, we show as well that this molecular signature can discriminate between ensemble cells according to the stage of encoding at which they were recruited (early/acute ensemble vs. late/repeated ensemble).” Was this demonstrated using a clustering approach? How was the discrimination performed?

Minor comments:

Did experimenters observe any aversive response to tamoxifen/DMSO? How might this affect the formation of a conditioned place preference (especially for acute groups) and expression of subsequent ensemble?

The size and quantity of the representative IHC figures (1d,i; 2c) needs to be improved.

The graphical representation for laser off/on in figure 2h is confusing and should be clarified.

A visual representation of the nuclei groups described in lines 234-237 would be a welcome addition to figure 3.

Figure 3j suggests that glutamate receptor genes and signal transducing proteins are more likely to occur in the D1 MSNs of repeated ensembles vs acute ensembles, which had increased expression that was more balanced between D1 and D2. Please comment on this difference.

Were there any changes in dopamine receptor genes for neurons in any of the ensemble classes?

Line 314 should refer to Fig. 4c and 4d.

Please specify whether mice were single or group housed. If group housed, were acute/repeated mice housed together or separately?

Please specify the sex of the mice used in these experiments. Were both sexes included in this study? Why or why not?

Reviewer #3

(Remarks to the Author)

Reviewer #4

(Remarks to the Author)

This study investigates Arc expression in the nucleus accumbens after acute or repeated cocaine exposure in mice. It is an important area of research to understand how drug associations develop in reward related regions like the accumbens. The authors combine Arc-TRAP mice with techniques like immuno, optogenetics, and snRNA to understand how acute vs repeated cocaine (and its pairing in a cpp chamber) exposures are related from an ensemble standpoint.

I commend the authors on nicely designed graphics, figures, and clear-to-read text. However, there are major concerns in the data collection, interpretation, and novelty of the findings. Therefore, I do not recommend publication in Nature Communications.

Additionally, enthusiasm for the novelty of this manuscript is overall dampened by the recent publication of a similar manuscript (PMID: 38901723) in Biological Psychiatry that demonstrates that repeated cocaine Arc ensembles are a small but functionally relevant population over acute cocaine Arc ensembles through multiple lines of experimentation/manipulation. This manuscript includes more complex cocaine-related behaviors, including cocaine self-administration. The authors do not comment on this paper although this review request was sent after its publication.

The current study under review has snRNA as a distinguishing line of work, but there are issues with the collection and interpretation of that experiment, as well as the other experiments in the study, that I outline below:

1. The overall message of the manuscript is ununified and contradictory. The abstract emphasizes the idea of reactivation but in the next sentence states that 'early vs late' cells are largely distinct. The end of the abstract states again the importance of the idea of reactivation of initial cocaine paired cells contributing to molecular mechanisms. This is continued through the manuscript.

2. Behavior measures are shown in arbitrary or transformed measures -'arbitrary units' for locomotion in Figure 1, a non-traditional CPP score in Figure 2- that deviate from typical presentation of these types of data. I recommend plotting by a standard measure that is commonly used by the field: locomotion: distance traveled or beam breaks; cpp: t(cocaine) – t(saline) separately for pre- and post-tests. Include both plots even if some is in supplemental.

3. There is a difference in the number of Tom+ and Arc cells between the groups (Extended data figure 3). Precision and recall associated with an F-score is more appropriate for the overlap data as it considers the number of neurons in both experiences.

i. Precision = Number of reactivated neurons (tagged and stained)/Number of neurons stained during second experience
ii. Recall = Number of reactivated neurons (tagged and stained)/Number of neurons tagged during first experience
iii. F-Score = $2 \times (\text{Precision} \times \text{Recall}) / (\text{Precision} + \text{Recall})$

4. The results of F2 are puzzling.

a. There is 'reactivation' of the weak 1 pairing group, but the authors claim there is no cpp acquisition. The data for the 1 pairing almost seems bi-model- 4 mice appear to have a positive cpp score, while 3 mice have a negative score.
b. The cpp score does not seem to be correlated with the reactivation ratio for the repeated group. My guess is it related to the number of cells.
c. The optogenetics data are also puzzling to the narrative, but I attribute it to the increased number of cells tagged during the weak conditioning (extended data figure 3) and a larger 'activation' of the region rather than a specificity of the experience. There is no supplemental figure with fiber placements for this experiment.

5. The sequencing depths are vastly different between the GFP+ and GFP- libraries but are used for differential expression. 350 M reads per sample, but 5 k or 20 k nuclei as input. This means for GFP+ (5k input) the sequencing depth is 70 k / nucleus, and for GFP-, the sequencing depth is 17.5 k / nucleus. This is huge confound, not to mention 16-20 mice were combined per sample with relatively few nuclei recovered from such numbers. DEGs could be due to the increase in

detection in the GFP+ groups. Repeated home cage GFP+ had the most UMIs and the most reported DEGs in F3. QC measures for snRNA data are typically shown as a violin plot instead of listing the median of the group (Extended data fig 6d).

6. There are many issues with the snRNA analysis, some are outlined below.

- a. The use of so many pairwise comparisons to determine DEGs is not statistically sound for this multi-factor dataset. The number of cells for each of the snRNA analysis is unclear and should be reported in the figure legends.
- b. I compared the gene names between D1-MSN markers, D1-MSNs_Acute_GFP-POSvNEG, and D1-MSNs_Repeated_GFP-POSvNEG and found that 26 genes for acute and 37 genes are in the D1 markers list, indicating that the differences in sequencing depth are likely impacting results. For Arc+/-, 38 markers overlapped with acute degs, and 33 markers overlapped with repeated degs. 25% of representative DEGs in F3j are markers of the cell type.
- c. Logistical regression assumes that the distribution of expression levels is similar across groups, but the sequencing depth is very different. The chi squared test is sensitive to sample size, and small sample sizes (Arc/GFP population combos) can lead to unreliable p-values.
- d. 1 read minimum to consider a nucleus Arc+ - is not appropriate. The distribution of Arc expression between homecage and test should be examined and a threshold determined just as a threshold is determined for classifying a cell as Arc+ with orthogonal techniques like immuno.
- e. It is unclear if the pathway analysis used a background list of genes expressed by cell type for enrichment analysis. This is critical to reduce false positives due to tissue-specific gene expression. The IPA results are essentially the same 4-5 terms, and it's likely the cell type markers would give the same results.

7. A recent paper (PMID: 36965548) indicates that a subset of D1-MSNs exhibits the transcriptional response to cocaine. It appears that the D1-MSNs are 2 'clusters' in this dataset, but both have acute/repeated cells within them. The authors should consider how their results relate to this manuscript. The authors might consider plotting the clusters by QC features like nCount or nFeature to ensure that the D1 subclusters (or other subclusters) are not due to quality. It isn't a novel finding that D1 MSNs primarily respond to cocaine at the transcriptional level.

8. There is no naïve control (or even a Tom- control) for Figure 4 to properly contrast the acute and repeated results. Are the patched cells D1 or D2? Neither the figure caption nor methods indicate the number of mice for these experiments. It seems that the data plotted is from all cells recorded not corrected for the number of mice.

9. Semantics comments: 1) Figure 1 refers to acute vs repeated as the number of injections, while figure 2 refers to acute vs repeated as the 'conditioning strength'. While I see a conceptual relationship, they are distinct. Early vs late and acute vs repeated are also interchangeably used. 2) 4OHT and tamoxifen are used presumed interchangeably? They are distinct drugs with very different labeling time courses. 3) CPP memory retrieval is not context-induced relapse. 4) if overlap is low, is this an engram/ensemble? Is cocaine exposure an ensemble or a stimulus response? How did this paper address allocation (mentioned in the discussion)?

10. I could not evaluate any of the code used to generate the figures as it was not available for review, and the sequencing data was not available via a GEO reviewer access code. I requested access via the editor, but the authors were not responsive.

Version 1:

Reviewer comments:

Reviewer #1

(Remarks to the Author)

I was very positive on the earlier version of this manuscript and now the authors have made it even better. Recommend accepting.

Reviewer #2

(Remarks to the Author)

The authors have thoughtfully addressed the reviewer's concerns and provided thorough responses, which we greatly appreciate. The additional clarifications regarding experimental design and methodology, as well as the incorporation of the SVM analysis to identify discrete ensemble groups, significantly enhance the manuscript. The inclusion of Arc in the title highlights an intriguing aspect of the findings, and we believe that adding Fig. R1 in the revised submission would further increase the precision and impact of this work.

Finally, if the core versus shell distinction falls beyond the intended scope, acknowledging this as a limitation in the discussion would provide valuable context for readers.

Reviewer #3

(Remarks to the Author)

Reviewer #4

(Remarks to the Author)

My primary concern remains with the snRNA-seq data collection. While the authors addressed some issues, such as defining the distribution of Arc expression between homecage and test groups, they have overlooked a critical flaw in their data acquisition that severely limits their analysis. The authors' claim that "The use of multiple pairwise comparisons has always been and still remains a complex and controversial issue in RNA-seq analysis" is incorrect. Statistical packages like limma, EdgeR, and DESeq2 have long been validated for RNA-seq (and snRNA-seq) analysis, including complex multifactorial designs (e.g., factor1*factor2) that appropriately account for multiple comparisons.

Although single-cell studies have lagged behind bulk RNA-seq in adopting such methods—partly due to the practice of pooling samples, as done in this study—this limitation was entirely avoidable. For example, a plate-based method like Smart-seq could have been used to 'sequence small, discrete ensembles defined by activity-dependent tagging' while retaining sample-level resolution. This makes the decision to pool samples surprising, particularly in a study otherwise so rigorous in its statistical frameworks and data analysis. As pooling limits the ability to model biological variability, it represents a critical confound in data acquisition that undermines the authors' claims and the study's overall impact.

For the optogenetics data, I appreciate the control experiment that the authors provided in Figure 2i in the updated manuscript. It is necessary for the authors to confirm that this 'home cage tagging with cocaine' is indeed a similarly-sized ensemble as the 'weak pairing of cocaine and cpp box' ensemble. A large body of work from A Badiani and T Robinson show that the context in which a drug is administered in part dictates the size of the 'ensemble'.

On the topic of F-score vs. 'novel analysis framework' (logit-based model?), I appreciate the in-depth discussion of the analysis method. Logit-based models also have limitations, such as suffering from over-fitting with low biological replicates and are less intuitive to interpret, but the results from both methods do agree with one another, so I will leave it alone. If the authors do want this method to catch on in the engram field, I suggest creating a vignette for tagged/stained cells specifically.

Version 2:

Reviewer comments:

Reviewer #4

(Remarks to the Author)

I appreciate the authors thorough review of recent Nature group publications. It helped paint an image of Nature group's current standards, although the variability in workflows/statistical standards among recently published studies is surprising. Brunner et al stuck out for the 'pseudo-sample' classification when sample-level information is not available, as in the case of the present study's experimental design.

The editor may decide if they want to see this prior to publication, but I recommend the authors rerun their analysis with a pseudo-sampling/multifactor analysis and see if their results are reproducible between methods. The main critique in using cell-level analyses (apart from preventing multifactor analysis) is it can overinflate statistical power since cells represent more of a technical replicate over a biological replicate. In the paragraph addressing this limitation, please discuss 'pseudo-sampling' and cite Brunner et al. I looked at the preprint referenced in the discussion paragraph, and it appears to be using multiplexing instead of a plate-based sorting. Multiplexing has been previously described (cite PMID: 31266958), although not widely adopted (as evident in your state of the field table) or successfully used for scarce cells like the Arc GFP tagged cells.

I also appreciate their LR/MAST/Wilcoxon justification and FDR correction discussion. I see inconsistencies in the FDR threshold (Figure 5k and 6f uses < 0.1 when the rest of the paper uses at least < 0.05). I recommend FDR < 0.05 , which is consistent across analyses and in line with the current publication standards of studies applying FDR correction in the table provided.

The distribution plot of Arc counts included in the last rebuttal letter between home cage and test groups should be included in supplemental.

Version 3:

Reviewer comments:

Reviewer #4

(Remarks to the Author)

I appreciate the author's willingness to run additional analysis, and looks like all of my comments are now addressed and discussed.

POINT BY POINT RESPONSE

Reviewer #1 (Remarks to the Author):

Understanding how drug-context learning is processed in the brain is an important topic. The present manuscript uses activity-dependent labeling to tag neurons in the nucleus accumbens during cocaine-context pairing (either with 1 or multiple pairings). First, they find that the ensemble of engram neurons active during training are reactivated when mice are placed back in the context drug-free. This is in line with multiple engram ensemble studies examining this and other types of memory. However, the current manuscript goes on to show differences between acute or more chronic pairings and then examined the transcriptional profiles of these neurons.

This manuscript comes from a leading group exploring substance abuse and describes the results of quite a bit of work, particularly with the transcriptional profiling, with very intriguing data.

I am impressed by this manuscript overall, and find the data foundational. I believe that others will be able to build off this dataset for further studies. I have only a few comments that the authors might chose to address in the discussion section. I frankly do not see the need for further experiments or analysis as i believe the results represent a unique and intriguing dataset and story.

We are grateful for the extremely positive Reviewer's comments and for describing our study as "impressive" and "foundational". We also agree that this dataset represents a unique opportunity to build upon for further studies in our lab and others. We have addressed all the comments below, largely by tackling the raised questions throughout the main text.

1. Compared to the acute condition, repeated cocaine exposure resulted in a smaller engram (fewer Arc+ cells), with lower reactivation of cells labeled during the initial injection, lower correlation with cocaine-induced locomotion, and robust memory retrieval in a cocaine place preference paradigm. At the same time, optogenetic activation of the "repeated pairing" engram does not induce memory retrieval, unlike the acute pairing condition. The authors propose that the repeated ensemble serves as a homeostatic mechanism, particularly as reactivation appears to be correlated with a lower CPP score. This is an intriguing hypothesis and, as a reader, I would have loved a more thorough discussion of this idea.

This is indeed an exciting hypothesis, and we thank the Reviewer for the opportunity to expand. We have now added a few sentences in our Discussion integrating our work with published literature from leading groups studying the cellular/synaptic role of Arc induction (e.g., Drs. Bito, Worley, and Finkbeiner) that put forward the idea that Arc induction is a paradoxical process, as Arc is induced by activity but can promote negative plasticity responses. We see our work as a putative extension of these concepts to the ensemble level, where Arc induction in the repeated ensemble could tag cells that become negatively associated with memory strength. We have also taken this opportunity to highlight how Arc can in that sense be seen as a "special" IEG by comparison to other activity-dependent transcripts (related to Reviewer 2 comment #1).

2. Another possibility to explain these data stems from the field's realization that neuronal activity tagging techniques can sometimes "overtag" true engram neurons. The present manuscript finds that acute cocaine activates a large number of NAcc neurons; many of these neurons could regulate cocaine-induced locomotion and a minority could be considered true fundamental "engram neurons". It is only with repeated injections that a discrete cocaine-context engram forms and "locomotion" neurons desensitize. This might explain the differences in CPP retrieval, the differences in biophysical properties and why there is relatively little overlap between the two conditions. I imagine there could be a clever way to test this experimentally, but I believe that such an experiment would be beyond the scope of the present manuscript.

This is indeed an interesting conceptual point, upon which we have further extended in the Discussion. We think that the definition of engram cells initially described by Tonegawa and colleagues (PMID: 26335640, 29970909, 31896692) is a valid, clear-cut, and a set of experimentally useful criteria, to which we have referred when designing our study to identify memory-related, putative "engram" cells in NAc. To be considered as such, cells are required to: 1) be activated during memory encoding, 2) display persistent plasticity-related changes, 3) be reactivated during memory expression, and 4) be sufficient for memory retrieval. In that sense, we argue that the true "engram"-like cells – which we have shown meet those requirements – in NAc in cocaine-context learning are the activated and reactivated cells (the Tom+/Arc+ "yellow" cells in tagging experiments). Yet, the Reviewer is right to note that tagged ensembles are larger, and most likely comprise additional cells that are not part of the "true engram". However, whether these result from technical "overtagging" (i.e., non-specific cells) or whether these map other behavioral dimensions associated with the encoding/retrieval conditions but not directly related

to memory (locomotion, attention, motivation, behavioral state, etc.), remains unresolved. We added this point to our Discussion. Related to this, we also highlight in the Discussion that the reactivation rates we observed in our CPP experiments (25-40%) is comparable – if not slightly above – to that observed in hippocampus and amygdala in canonical learning and memory studies, which we see as a qualitative indication that these ensembles likely contain similar fractions of “true engram” cells. As noted by the Reviewer, experimentally testing the engram/ensemble substrates of distinct yet highly related behavioral sub-parameters (such as locomotor vs memory responses) is a very a complex endeavor, for us as well as for colleagues in similar fields. We do however underscore that, in our work, we do not directly refer to our ensembles as engrams, but rather as ensembles with engram-like properties – the existence of true memory engrams in striatal regions is a fascinating perspective, and we hope that our work opens the door for such investigations.

3. The analysis of transcripts in Arc+ and Arc- neurons is important data and suggests heterogeneity in plasticity-related transcript changes that are related to the function of specific ensembles. Ideally, the authors would examine the relationships between specific transcripts (or combinations thereof) and ensemble function that is linked to specific aspects of memory acquisition or retrieval. Again, this experiment is beyond the scope of the present manuscript, but the authors could note it as a limitation in the discussion section.

Identifying how specific transcripts (alone or more likely in combination, as suggested by the Reviewer) orchestrate neuronal recruitment in memory engrams is a critical question for the field, and a goal of the follow-up work we have started in the lab. Similar work aiming to dissect the molecular mechanisms of engram allocation (which cells are incorporated?) and reactivation (which cells are reactivated?) is starting to emerge in other memory-related paradigms, and has highlighted critical molecular players, such as CREB (PMID: 25319707, 27187069) or histone acetylation (PMID: 39052786). Our snRNAseq represents a foundational dataset to guide further exploration of such questions – for the first time in striatum. We also hope to combine our ensemble work with CRISPR-mediated multiplexed gene manipulation strategies to selectively manipulate key transcripts within Arc+ encoding cells to test how they support Arc reactivation and memory expression. Our strongest candidates are the genes highlighted in Fig. 6., especially with the newly added classifier analysis (see Reviewer #2 comments), and we appreciate the Reviewer’s note that such experiments are beyond the scope of this present manuscript – we have however included a sentence to that point in the Discussion.

Reviewer #2 (Remarks to the Author):

Salery and colleagues examined the engagement and reactivation of neuronal ensembles in the nucleus accumbens (NAc) using cocaine-conditioned place preference (CPP)—a task with all the features of a Pavlovian associative learning task. The approach involved taking advantage of the Arc immediate early gene as a marker for neuronal activation, which has been used in the learning and memory field to mark recently activated neuronal ensembles in the hippocampus. Here, the authors used ArcCreER mice to tag and examine the activity of the neural populations engaged during acquisition and during context induced recall, and report that populations identified during acquisition are also active during context induced recall. They further report that different ensembles in early vs late acquisition are distinct, with distinct impacts on memory retrieval. Ensembles were isolated to study using single nucleus RNA sequencing, which revealed plasticity-related transcriptional programs that segregate cocaine-recruited NAc engram cells unique to distinct stages of memory. In addition, optogenetic-mediated neuronal activation of the acute, but not the repeated, ensemble elicited a higher CPP score during the test session. Overall, these findings are highly significant because they further our understanding of cocaine-associated memory formation, which is important as these memories contribute to pathological relapse events.

We are thankful for the Reviewer's positive comments on our study and for highlighting the "significant" and "important" contribution of our work to "further our understanding of cocaine-associated memory formation". We also strongly agree that pursuing this line of work is critical to better understand the basis of pathological drug-related conditions. We have addressed all the comments below by adding extensive supplementary data to the manuscript and expanding our Discussion section in light of the key points raised by the Reviewer.

Main Comments:

The foundation of this paper is based on the immediate early gene Arc. Arc is an unusual immediate early gene—usually referred to as a super-immediate early gene because of transcriptional pausing leaving Arc already partially transcribed and ready for a stimulus to release the pause (Saha et al., Nat Neuro 2011). Thus, would similar results be observed with a different marker of neuronal activation? Or does the potential unique nature of Arc transcription make it an exceptional tool for this study? Could the authors please comment on this in the discussion?

We are greatly appreciative of the Reviewer for pointing out this intriguing feature of Arc and for highlighting its unique nature amongst other immediate early genes. Indeed, the unique nature of Arc transcription and function makes it a particularly powerful tool to assess learning and plasticity processes (see also Reviewer #1 comment #1), an argument that we have now covered in more detail in our Introduction and Discussion. Accordingly, we have also added "Arc" to the manuscript title.

With regards to comparing Arc with other IEGs more directly, we have added in a new Extended Data Fig. 1 RT-qPCR data looking at expression patterns of other well-characterized immediate early genes (Fos, Fosb, Zif268/Egr1, and Npas4) after acute and repeated cocaine treatment. This analysis indicates that these other IEGs do exhibit some level of desensitization with chronic exposure, although of varying amplitude – overall consistent with literature on this topic.

However, similar bulk mRNA levels do not imply that these IEGs are induced in the same ensembles. It appears that in response to acute cocaine, Arc and Fos are both induced in very similar subsets of neurons (PMID: 27567310, 38901723). This does not exclude that the few non-overlapping cells might have an important role in segregating the Arc and Fos ensembles.

In our dataset, we show that recall-induced Arc expression generally correlates with that of multiple other IEGs (Fig. 5g), which we used as an argument to confirm that Arc expression does "tag" transcriptionally activated neurons. However, a finer analysis (Fig. R1) revealed that, although significantly overlapping, there is a significant population of neurons that express only Arc (e.g., only 30% of Arc+ cells are Fos+, and 28% are Npas4+). This analysis suggests the possibility that distinct IEG-defined ensembles exert different roles in specific memory-related processes, with the putative homeostatic role of Arc a likely key difference (see also Reviewer #1 comment #1). This represents an interesting avenue for follow-up work, which we now acknowledge in our Discussion. We would prefer not to include this new figure in the revised manuscript, because we think it requires considerable additional work that is beyond the scope of this study, but would of course include it if the Reviewer and Editor would like us to do so.

Fig. R1

What would the authors expect to see in terms of ensembles if, in the acute model, the second cocaine injection were given on day 2 instead of day 5? This question relates to Arc transcription and cycling, and data represented in Figure 1. Are the differing rates of reactivated ensembles a product of time rather than number of cocaine exposures or phase of acquisition?

This is an interesting question. To test this, one can either 1) vary the length of time between the same number of cocaine injections, as suggested by the Reviewer or 2) vary the amount of cocaine injections with the time being the same. We have performed experiment 2 (Extended Data Fig. 2), and seen that a *second* injection of cocaine given 5 days after the first does not lead to a reduced ensemble size unlike a *fifth* injection given 5 days after the first which does. This finding supports our conclusion that the differing rates of ensemble reactivation are most likely not a product of time but rather of the number of cocaine exposure. We have clarified this point in the Results.

Performing the reciprocal experiment (1) would be an excellent control, however, this is not possible experimentally as 4-5 days is the minimum time required to observe Tomato expression in tagged cells. In theory, we would expect a similarly sized ensemble in such an experiment as we see between a 2-injection protocol separated by 5 days (same number of injections, only time in between varies).

This is better accounted for with the experimental design described in Figure 2, but in the case where overall Arc expression is decreased after repeated injections (Fig 1b) leading to smaller ensembles (Fig 2c), it is perhaps not surprising that the overlap is smaller during context-induced reactivation (Fig 2d).

Thank you for the opportunity to clarify the statistical framework that we used to tackle this issue (see also Reviewer #3 comment #3). It is true that the repeated Arc ensemble is smaller than the acute, which makes it critical to correct for that fact when quantifying overlap/reactivation. We have thus assessed the probability of overlap/reactivation over individual “chance” reactivation levels by using logit models, which have been designed to that end. Differences in overlap/reactivation between groups were still very significant after such corrections, which makes us confident that this represents a true biological difference rather than a random sampling artefact. We have detailed the justification for this approach in the main text, and hope to convince this reviewer as well as the field of the statistical validity and power of such analyses.

The paradoxical finding that the recall activation more closely resembles the acute ensemble than the repeated ensemble is not entirely addressed by the optogenetic experiment. For example, after several days of cocaine conditioning, the first test day is also the first day of extinction. At this time point, the repeated conditioning mice are arguably engaged in extinction learning as well as recall, involving disparate ensembles.

This is true. We now acknowledge this experimental limitation in the Discussion. We see this related to Reviewer #1 comment #2), with the question of the recall-activated but non-“engram” cells. Some of those could very well encode extinction behavior, especially in light of the theory that extinction might recruit new “extinction learning” circuits and ensembles rather than affect the original memory trace *per se*. (PMID: 26447572; 17882236, 28329757). And again, such a relationship between Arc ensembles and extinction vs recall behavior might very well be a specificity of Arc compared to other IEGs because of the above-mentioned link between Arc and homeostatic processes. Thank you for this thought-provoking suggestion, which will undoubtedly guide further investigation and conceptual reflection.

Conversely, the acute ensemble may more resemble the recall activation ensemble because of novelty, and total distance moved induced by novelty. Is it possible that these acute ensembles are also more associated with locomotor movement? The distance moved for the CPP experiments should be reported in order to understand whether this is a component worth analyzing as well.

This is an important consideration, and locomotion is an important control metric to take into account especially in striatal regions. We have plotted locomotor activity within the CPP box during test sessions before and after cocaine conditioning (Extended Data Fig. 5a). While locomotor activity was sensitized (increased) by cocaine conditioning – likely either a conditioned response or denoting active drug-seeking – this effect was similar between acute and repeated pairing groups, demonstrating that locomotor activity is not an underlying factor for the differences in ensemble sizes and reactivation rates. We also noted this important observation in the main text.

The process for selecting regions of interest to quantify Arc expression and subsequent ensembles is described in the methods: “Signal was quantified in NAc core and medial shell along the rostral-caudal axis for total of 12-15 different images (363.64 x 363.64 μm) per animal (i.e., 1 image per structure, per side, on three adjacent

sections).” Please clarify this process, visually (a schematic) or verbally (more detailed methods). With the relatively sparse Arc signal and the anatomically vague representative images included in Figures 1 and 2, it is difficult to understand how bias was removed from the ROI selection process, image selection process, image analysis process, etc. For example, it seems as if only part of the NAc has been analyzed, and if that is the case – is it the same part for each brain, if so why was this section selected? Is the anatomy of cocaine ensembles conserved across different mice?

We are thankful for the opportunity to clarify this important point, and have revised the corresponding methods section, which we acknowledge was not clear. For each animal, we imaged 3 sections (at the same AP for all animals), on both sides with 3 square ROIs on each side almost entirely covering the medial section of the NAc (from the tip of the lateral ventricle [dorsal limit] to the medial forebrain bundle and ventral pallidum [ventral limit], and from the lateral/medial septum [medial limit] to just lateral to the anterior commissure [lateral limit]). This gave us ~2 mm² of tissue for each animal, which is a robust and representative amount. Images were acquired by an experimenter blind to the group, removing any bias from the ROI selection. Imaging was performed with individual ROI instead of whole-hemisphere tiles for maximizing both the quality (images taken at 40X) and consistency (no decreases in signal at tile borders, or in case of uneven slices or antibody penetration) of IHC signals, which was critical to remove unwanted bias/differences in overall cell counting. We focused on the medial NAc (core + medial shell) because it is the region that has received by far the most attention in studies of drug responses.

Similarly, it seems like the shell and core of the accumbens were considered collectively for most of this paper. Given their differing roles in acquisition of cocaine associated behaviors, and the technical ability to separate these areas can the authors please comment on this combined approach?

It is true that NAc core and shell are part of somewhat distinct, parallel sub-circuits. However, for this study, we wish to keep referring to “NAc” in general for multiple reasons:

1) We have checked whether Arc ensemble dynamics were similar between core and shell, and this was indeed the case. Arc was induced in both regions by acute cocaine, and Arc ensemble size desensitized in both regions with repeated exposure (Fig. R2).

2) While IHC and electrophysiology allowed us to potentially disentangle NAc core and shell, this was not possible with optogenetics and snRNAseq (which used whole NAc dissections). For consistency, we therefore combined IHC data from core and shell.

3) Our manuscript is already conceptually complex, and we believe core vs shell distinctions, while important in general, would not represent the most appropriate lens through which to discuss and interpret our data here, especially given that we do not detect major differences between the two subregions.

Altogether, we hope to have convinced this Reviewer that our manuscript does not need to dive into more detailed dissection of core vs shell effects, however, we would be happy to do so if the Reviewer and Editor find this important.

Line 294 states: “This dataset also dissects the plasticity-related, activity- and dopamine-dependent transcriptional signature unique to cocaine-activated NAc cells. Beyond these shared features, we show as well that this molecular signature can discriminate between ensemble cells according to the stage of encoding at which they were recruited (early/acute ensemble vs. late/repeated ensemble).” Was this demonstrated using a clustering approach? How was the discrimination performed?

We are particularly grateful for this comment, as it prompted us to perform the additional analyses described below, which we see as robust validation of some of our central findings and interpretations. In the initial manuscript, we did not perform any clustering or discrimination analysis – we simply justified our statement (“discriminate”) by the DEG analysis. However, the Reviewer’s suggestion was an intriguing lead: *Is it possible to truly “discriminate” between nuclei of key groups using only their transcriptomic signature?* To test this, we capitalized on support vector machines (SVM), a fairly commonly utilized supervised classification technique. We trained and asked such decoders to predict nuclei identity in 2 experiments which we see as central to our manuscript (1- acute vs repeated GFP+ ensembles in Fig. 3i, and 2- reactivated vs non-reactivated “engram” cells in Fig. 6d) using only the expression of the previously identified corresponding DEGs (i.e., 20-80 genes in

Fig. R2

1-, 10-20 “reactivation” genes in 2-). Decoders performed extremely well (Extended Data Fig. 11), which we interpret as a strong finding: precise, subtle differences in gene expression are “enough” to properly and robustly differentiate between those groups of cells. The implications of this new analysis are important and two-fold. First, this analysis validates that the identified DEGs are truly DEGs through reciprocal testing. Second, the analysis demonstrates that relatively limited genetic information can be used by mathematically fairly simple algorithms to dissociate between distinct neuronal ensembles, which leads to the hypothesis that brain circuits might similarly use this transcriptomic make-up (and not only circuit identity) to properly route behaviorally-relevant information within relevant neuronal ensembles. We agree that this is a strong interpretation, which we have not mentioned to the same extent in the manuscript, but we are extremely thankful for this comment which triggered this investigation and these hypotheses.

Minor comments:

Did experimenters observe any aversive response to tamoxifen/DMSO? How might this affect the formation of a conditioned place preference (especially for acute groups) and expression of subsequent ensemble?

This is an important consideration. As detailed in Methods, we used an aqueous solution for tamoxifen in 5% DMSO, to which aversive reactions are minimal. We did not qualitatively observe aversive reaction upon injection. In any case, all mice (including controls), received tamoxifen, so any potential confounding factor related to tamoxifen itself or its vehicle are accounted for by experimental design.

The size and quantity of the representative IHC figures (1d,i; 2c) needs to be improved.

We increased the size of all representative IHC images by 50%.

The graphical representation for laser off/on in figure 2h is confusing and should be clarified.

We tried our best to clarify with minor modifications, and remain open to more specific/precise recommendations if needed.

A visual representation of the nuclei groups described in lines 234-237 would be a welcome addition to figure 3. Fig. 3b schematically represents all group comparisons performed throughout the manuscript, including the ones described at those lines.

Figure 3j suggests that glutamate receptor genes and signal transducing proteins are more likely to occur in the D1 MSNs of repeated ensembles vs acute ensembles, which had increased expression that was more balanced between D1 and D2. Please comment on this difference.

We have added a comment in the Results – thank you for the suggestion.

Were there any changes in dopamine receptor genes for neurons in any of the ensemble classes?

We had not looked at dopamine receptors in particular – this is an interesting suggestion. First, we confirmed *Drd1* and *Drd3* expression in D1-MSNs and reciprocally *Drd2* expression in D2-MSNs (Extended Data Fig. 8b) – a good “sanity check”. *Drd4* and *Drd5* expression levels were too low to be detected. In terms of DEG analysis, we examined all dopamine receptor genes (*Drd1*, *Drd2*, *Drd3*) in all of our comparisons. The only differences that passed our stringent significance thresholds (see Methods), were the following:

- Increased *Drd2* expression in cocaine-activated (GFP+) compared to non-activated (GFP-) D2-MSNs
- Decreased *Drd2* expression in recall-activated (Arc+) compared to non-activated (Arc-) D2-MSNs

As we wish not to focus on select candidate genes in this initial study, we have not mentioned this fact in detail in the manuscript – it is nevertheless possible to find that information in the supplemental tables (3,4,5) that lists all DEGs for all comparisons. And we would be happy to add direct mention should the Reviewer and Editor prefer us to do so.

Line 314 should refer to Fig. 4c and 4d.

This has been addressed.

Please specify whether mice were single or group housed. If group housed, were acute/repeated mice housed together or separately?

Mice were group housed, but one litter always only comprised mice from one experimental condition. We have specified this in Methods.

Please specify the sex of the mice used in these experiments. Were both sexes included in this study? Why or why not?

Mouse sex is now specified in each figure legend – thank you for catching this oversight. qPCR, IHC, optogenetics and electrophysiology used males, while snRNAseq used both males and females. Follow-up, confirmatory work focusing on select candidate genes will include equal numbers of males and females.

Reviewer #3 (Remarks to the Author):

We thank this Early Career researcher for the time and attention paid to our manuscript.

Reviewer #4 (Remarks to the Author):

This study investigates Arc expression in the nucleus accumbens after acute or repeated cocaine exposure in mice. It is an important area of research to understand how drug associations develop in reward related regions like the accumbens. The authors combine Arc-TRAP mice with techniques like immuno, optogenetics, and snRNA to understand how acute vs repeated cocaine (and its pairing in a cpp chamber) exposures are related from an ensemble standpoint.

I commend the authors on nicely designed graphics, figures, and clear-to-read text. However, there are major concerns in the data collection, interpretation, and novelty of the findings. Therefore, I do not recommend publication in Nature Communications.

We thank this Reviewer for their positive comments on our manuscript's design and clarity. We also understand the Reviewer's other concerns – however, these mostly stem from analytical and interpretation details rather than fundamental flaws. We have fully addressed all of these concerns below with much extended analysis, different methods of quantification, methodological clarification, and additional experimental data. We appreciate the prompt to better justify some of our experimental design and analysis choices and to include novel important analyses (e.g., F-score, IPA background lists, etc.), which we agree were either missing or misleading in the original submission. The revised manuscript is therefore much improved, and we believe even more convincing thanks to this Reviewer's input.

Additionally, enthusiasm for the novelty of this manuscript is overall dampened by the recent publication of a similar manuscript (PMID: 38901723) in Biological Psychiatry that demonstrates that repeated cocaine Arc ensembles are a small but functionally relevant population over acute cocaine Arc ensembles through multiple lines of experimentation/manipulation. This manuscript includes more complex cocaine-related behaviors, including cocaine self-administration. The authors do not comment on this paper although this review request was sent after its publication.

The current study under review has snRNA as a distinguishing line of work, but there are issues with the collection and interpretation of that experiment, as well as the other experiments in the study, that I outline below: We thank the Reviewer for the opportunity to discuss our findings in light of the above-mentioned study from a former collaborator. Before anything else, we explain that this reference was not included in the original manuscript because it had not been published at the time of our manuscript's first submission in January 2024 to another *Nature Publishing Group* journal, which was then transferred to *Nature Communications*. We have of course referenced the Thibeault et al study in our revised manuscript.

However, we see our study as fundamentally different in terms of the scientific questions investigated, the overall experimental design, and the robustness of the underlying data, which in our opinion makes our work a major conceptual and technical advance for the multiple reasons listed below.

First, the main common finding is that Arc induction desensitizes with chronic exposure to cocaine, which, for our study, is only a first descriptive datapoint upon which the rest of our study is based. Such a finding does not require the use of Arc-Cre tagging strategies but can simply be found with Arc qPCR and/or IHC, as we did, before confirming that Arc-Cre tagging recapitulates those endogenous patterns. We also want to point out that the Thibeault study starts with the very surprising, and unexplained in their manuscript, finding that acute cocaine does **NOT** induce Arc expression (their Fig. 1C), which contradicts years of data and raises concerns about the sensitivity of their tagging approach to endogenous Arc induction.

Second, the Thibeault et al study does not at all investigate endogenous reactivation of their “tagged” ensembles. *This is a major conceptual difference between ours and their study.* Tracking reactivation is at the basis of the entire memory/engram field, and the main rationale behind the development of activity-dependent tagging strategies (PMID: 28439228), as they allow experimenters to track the fate of the same neurons activated by experience A / memory encoding during later experience B / memory retrieval. This is the conceptual framework that guided our experimental design, and which makes our study a good fit and very important and novel contribution to this field. Our study remains the first and only work to date demonstrating neuronal ensembles with engram-like reactivation properties in striatum during cocaine-context learning, which is a highly impactful conclusion that is not touched upon by the Thibeault manuscript.

Third, and related to the previous point, we are surprised that the Reviewer finds the Thibeault study to include “more complex behaviors”, as their tagging strategy was in fact much simpler than ours (simple homecage tagging, no learning paradigm like our CPP protocol). They do go on to test the activation/inactivation of these homecage ensembles in supporting operant reinforcement. This is an interesting finding, but one that misses the mark on a critical concept in the field of neuronal ensemble tagging: ensembles are by definition supposed

to specifically encode a given, precise experience (in their case a bolus ip cocaine experience in the homepage), but tested for their role in an entirely different task (self-administration). How can these ensembles then be argued to selectively encode a cocaine experience? This is also even more confusing as they show that their control saline ensembles, which are of the same size as their cocaine ensembles, have the same effects on self-administration. To us, these effects are actually in line with the one interpretation proposed by the Reviewer below (comment #4c) that the effect might simply rely on a general activation of the region (as in, yes, activating NAc cells supports reinforcement, inactivating NAc cells blocks it, both well-established in the field). By contrast, our study uniquely compares the memory-related properties with the required specificity, and with the necessary controls: What are the precise roles of the ensembles activated at the first, initial vs later stages of memory encoding in relation to memory expression in the specific context in which that memory was encoded? This is why we have used a canonical Pavlovian associative learning paradigm like CPP, which has very well-defined learning/memory components, and fits the theoretical framework of the engram research field (largely focused on similar Pavlovian tasks like fear conditioning in other regions such as hippocampus and amygdala).

Fourth, our electrophysiological findings are entirely consistent with those of Thibeault (increased excitability of the repeated ensemble). There is undoubtedly value in replication. Also, we want to point out that this is not a central finding in our manuscript but rather a functional validation that the transcriptional differences detected in our snRNAseq do correlate with physiological differences.

Fifth, we argue that our snRNAseq in itself represents more than a “distinguishing line of work”. The investigation of the molecular determinants of ensemble allocation and ensemble-specific transcriptional responses is a fascinating enterprise, which is burgeoning, technically challenging due to the necessary combination of several state-of-the-art techniques, and has proved extremely impactful. Indeed, our study is far more in line with those in traditional memory investigations (e.g. PMID: 31837649, 25319707, 33177708, 39052786), but where we for the first time establish principles for striatum and in response to drugs of abuse.

Please find an in-depth, point-by-point response to detailed individual comments below.

1. The overall message of the manuscript is ununified and contradictory. The abstract emphasizes the idea of reactivation but in the next sentence states that ‘early vs late’ cells are largely distinct. The end of the abstract states again the importance of the idea of reactivation of initial cocaine paired cells contributing to molecular mechanisms. This is continued through the manuscript.

We are happy to clarify our statements. The confusion might be that we are not talking about the same exact meaning for “reactivation” between Fig. 1 and the rest of the manuscript. This is purely semantics, as the same word (“reactivation”) is used by us and others in the field of ensemble tagging and memory to describe 2 quite distinct scenarios.

On the one hand, in Fig. 1, “reactivation” refers to the overlap between the ensembles associated with a first vs a last injection of cocaine – conceptually, this is akin to testing the similarity (quantified by % ensemble overlap) in the response to experience A vs experience B, which is one well-accepted use of the diverse tagging strategies. We use this result to conclude that the ensemble response to experience B (last injection) is different from the response to experience A (first injection), which is quite unexpected because experience A and B are in practice very similar (an ip injection of cocaine in the same context). Thus our distinction between ‘early vs late’ ensembles, which we hypothesize encode distinct yet co-existing aspects of the cocaine treatment experience. It is actually this counter-intuitive, initial observation that justified the rest of our study.

On the other hand, we then wished to look at reactivation *per se* of the early vs late encoding ensembles (again, distinct but co-existing) during recall of an associated encoded memory – this is why we pivoted to a learning paradigm like CPP: to evaluate how experience/memory encoding ensembles (2 distinct ones, early vs late) reactivation respectively relate to memory recall.

Overall, the key point, which we discuss in our manuscript, is that these two distinct ensembles (early vs late) do not overlap (in terms of “type 1 reactivation”), and when associated with the encoding of a context memory, are differently reactivated (in terms of “type 2 reactivation”).

We hope to have convinced this Reviewer that, in that sense, the message of the manuscript is not “ununified and contradictory”. For clarity in the revised manuscript, we now do not refer to “type 1 reactivation” as “reactivation” but rather as either “overlap” or “re-recruitment”, which is less confusing, and keep the “reactivation” nomenclature to describe memory recall-associated reactivation. We thank the Reviewer for the opportunity to improve the clarity of our presentation.

2. Behavior measures are shown in arbitrary or transformed measures -‘arbitrary units’ for locomotion in Figure 1, a non-traditional CPP score in Figure 2- that deviate from typical presentation of these types of data. I recommend plotting by a standard measure that is commonly used by the field: locomotion: distance traveled or beam breaks; cpp: t(cocaine) – t(saline) separately for pre- and post-tests. Include both plots even if some is in supplemental.

We have used those metrics for clarity and homogeneity, and understand that readers might want to see less-transformed metrics. We have thus adjusted and/or added new graphs using meters traveled per 30 min session for locomotion (Fig. 1) and t(cocaine) – t(saline) values for CPP (Fig. 2).

This also led us to notice that the Methods section erroneously stated that mouse locomotion was recorded for 60 min – this was an inadvertent mistake, and is now corrected to state 30 min instead. Thank you for bringing this to our attention.

While we now additionally show CPP data with separated pre- and post-test differences in time spent in each chamber as suggested, please note that our CPP Score, calculated as $[t(\text{cocaine}) - t(\text{saline})]_{\text{post}} - [t(\text{cocaine}) - t(\text{saline})]_{\text{pre}}$ is as much of a common, sometimes even recommended, standard measure used in the field (PMID: 37693282) as the Difference Score suggested by the Reviewer, and mathematically equivalent. This is why we have kept this simpler, unified metric (one value per animal instead of two) for later figures and correlation analyses.

3. There is a difference in the number of Tom+ and Arc cells between the groups (Extended data figure 3). Precision and recall associated with an F-score is more appropriate for the overlap data as it considers the number of neurons in both experiences.

i. Precision = Number of reactivated neurons (tagged and stained)/Number of neurons stained during second experience

ii. Recall = Number of reactivated neurons (tagged and stained)/Number of neurons tagged during first experience

iii. F-Score = $2 \times (\text{Precision} \times \text{Recall}) / (\text{Precision} + \text{Recall})$

This is an excellent suggestion. We had already calculated our “Reactivation Ratio” using the formula (ii) for Recall. We have now added “Recall” as a title for those graphs, and calculated “Precision” and “F₁-score” using formulas (i) and (iii), now in supplementary information. All of these metrics confirmed our former conclusions.

We strongly agree with the Reviewer that it is critical to account for individual differences in the number of Arc+ and Tom+ cells for each animal, since those individual-wise “chance” levels are often overlooked in other studies. Calculating F₁-scores is one way to somewhat tackle this issue, although it has definitely not become prevalent in the engram literature and to the best of our knowledge remains mostly used in machine learning and classification efforts. In addition, F-scores suffer critical caveats especially when applied to this type of dataset: they perform poorly and/or can be misleading when classes are imbalanced (which is the case here), have a symmetrical precision/recall trade-off (which is not particularly desirable here), and ignore true negatives (which are in this case important at the systems level).

However, we appreciate the opportunity to expand more than we did in the original manuscript on the novel analytical method we developed and put forward in this study (Fig. 1k and Fig. 2e) to – we believe – more efficiently and more appropriately tackle this critical issue of different total number of Arc+ and Tom+ cells for each individual animal replicate, which lead to individual “chance” reactivation/overlap levels that need to be accounted for, as the Reviewer is – rightly – requesting. In our study, we consider such datasets as categorical count data with 4 multinomial outcomes (Arc+/Tom+, Arc+/Tom-, Arc-/Tom+, Arc-/Tom-) possible for each cell with a nested design where individual cells are nested within individual animals, which allows to compute animal-wise chance levels. To model and analyze such datasets, we referred to the standard-setting “Bible” textbook on analyzing categorical data (Categorical Data Analysis, Agresti A., 2002, John Wiley & Sons, Inc., doi: 10.1002/0471249688) and accordingly applied baseline-category logit models for multinomial counts, which allow for random effect factors to account for nested design. We used the *mclogit::mblogit* function in R and refer to the corresponding vignette for details. These models perform in the log-odds space and estimate log-odds ratios for each individual animal as well as corrected group-wise log-odds for each outcome for each combination of group membership (here in a 2x2 design). From those log-odds estimates, we back-calculated estimated probabilities for each outcome, and simulated dispersion metrics (95% CI) through bootstrapping (see Methods for more details). This gave us the chance-adjusted (i.e., corrected for differing number of Arc+ and Tom+, as well as differing total number of cells for each animal) probability of a Arc+/Tom+ cell, which we interpret as the corrected probability of reactivation for each group – we argue that this is the best metric that can be derived from such datasets to measure reactivation (superior to simple precision and recall ratios, and superior to F-

scores.) We hope to convince this Reviewer of the strength of such approaches, and that this will get picked up in the field for future analyses.

We have slightly expanded on this novel analysis framework in the main text of the revised manuscript, and have nevertheless included (as mentioned above) the F_1 -score suggested by the Reviewer in supplemental information (Extended Data Fig. 4,5), which we see as a nice complementary quantification method.

4. The results of F2 are puzzling.

a. There is 'reactivation' of the weak 1 pairing group, but the authors claim there is no cpp acquisition.

This is indeed what we observe (high group-wise reactivation, low (i.e., weak) group-wise memory strength/CPP score). It is unexpected, and this is why this represents one of the central, major findings of our study.

The data for the 1 pairing almost seems bi-modal- 4 mice appear to have a positive cpp score, while 3 mice have a negative score.

CPP score data are widely distributed, as it always is (e.g., PMID 36889314 for a large panel of CPP experiments). The new graph suggested by the Reviewer with separate pre- and post-test data illustrates perhaps even more clearly the absence of preference in acute/weak pairing animals. Place preference is thus always a group-wise measure, where "no preference" (as a group) derives from some animals having a positive score and some having a negative one. In fact, we here use this inter-individual variability to analyze preference as a continuous variable for correlations with reactivation. This is where we conclude that the higher the reactivation of the acute ensemble is, the stronger the memory, which led to the hypothesis that reactivation of the acute/1P ensemble is "positive" for memory expression – which we then causally test with artificial, forced reactivation with optogenetics.

b. The cpp score does not seem to be correlated with the reactivation ratio for the repeated group. My guess is it related to the number of cells.

We are very surprised by this comment. There is a strong, significant negative correlation between CPP score and reactivation of the repeated ensemble (Fig. 2f, right panel, $r = -.89^{**}$), which we argue supports our hypothesis that the repeated ensemble might play a "negative" role for memory strength and/or expression (therefore distinct from the "positive" role for the acute ensemble).

In other words, we interpret Fig. 2 d-f findings as "the more a mouse reactivates the acute ensemble and the less it reactivates the repeated ensemble, the stronger the memory recall". We hope this clarification is helpful.

c. The optogenetics data are also puzzling to the narrative, but I attribute it to the increased number of cells tagged during the weak conditioning (extended data figure 3) and a larger 'activation' of the region rather than a specificity of the experience.

We gladly clarify. Based on the correlations seen above, we hypothesize that reactivation of the acute ensemble enhances memory expression while reactivation of the repeated ensemble decreases memory expression. We test this by forcing either acute or repeated ensemble reactivation via optogenetics. We demonstrate that activating the acute but not the repeated ensemble increases memory expression. This observation is consistent with our hypothesis, not "puzzling" to the narrative.

It is true that the acute ensemble is larger in number of cells than the repeated one (Extended Data Fig. 5), which is expected and consistent with Fig. 1 data and endogenous Arc activation and desensitization patterns by acute vs repeated cocaine. To test whether the increase in CPP induced by optogenetic reactivation of the acute ensemble was selective to the exact cells recruited by cocaine-context memory encoding-activated cells/ensemble and not due to a large activation of the region as suggested by the Reviewer, we performed the control experiment in Fig. 2i. Here, we created a similarly-sized cocaine ensemble which consisted of the cells activated by a similar injection of cocaine but in the homecage, not in the CPP box (i.e., a different contextual environment). This ensemble is thus the same size and activated by cocaine but in a distinct context – a good "control" ensemble. Optogenetic activation of this control ensemble did not increase preference, indicating that simply activating a portion of NAc cells is not sufficient for CPP memory, but rather that this effect depends on the specific cells recruited (i.e., the ones encoding a specific cocaine-context association). As discussed above, we see this as a critical control – which parenthetically was not included in the Thibeault study (see above) – to justify that it is truly this acute ensemble (and not other, non-context-selective cells in that region) that encode this specific CPP memory: demonstrating the selectivity of that ensemble was crucial to us to illustrate engraving-like properties for these ensembles. This observation provides a strong argument in support of our conclusions.

There is no supplemental figure with fiber placements for this experiment.

We have now added a supplemental figure for fiber placements – thank you for noting this oversight.

5. The sequencing depths are vastly different between the GFP+ and GFP- libraries but are used for differential expression. 350 M reads per sample, but 5 k or 20 k nuclei as input. This means for GFP+ (5k input) the sequencing depth is 70 k / nucleus, and for GFP-, the sequencing depth is 17.5 k / nucleus. This is huge confound, not to mention 16-20 mice were combined per sample with relatively few nuclei recovered from such numbers. DEGs could be due to the increase in detection in the GFP+ groups. Repeated home cage GFP+ had the most UMIs and the most reported DEGs in F3.

We thank the Reviewer for the opportunity to clarify this key methodological detail. The Reviewer is right to note that there is a 3-4-fold difference in read sequencing depth between GFP+ and GFP- nuclei. However, we argue that this is not a “huge confound”, as differences in sequencing depth across samples is fairly common in any RNAseq experiment, and are efficiently accounted and corrected for in all standard analysis pipelines, including standard Seurat preprocessing like we used here.

First, differences in *read* sequencing depth do not directly translate to differences in *UMI* detection. With UMIs being the actual number of detected RNA molecules in a given nucleus (which is a finite, biologically defined number), increasing read depth might just increase the number of sequenced duplicates, which are by definition removed when converted to UMIs. In our dataset, while the difference in read depth is 3-4-fold, the difference in UMI depth is only 1-2-fold, as evidenced in our table of median UMIs per nucleus (Extended Data Fig. 8).

Second, the first step of any RNAseq analysis is to normalize libraries for sequencing depth (here, number of UMIs per nucleus). As part of the standard Seurat pipeline, we did so using the *Seurat::SCTransform* function, which calls the *sctransform::vst* function under the hood. This step also regressed out (i.e., corrected for) mitochondrial RNA contamination, which is another unwanted source of variation. We note that we did not specify this step in the corresponding Methods section of the original manuscript, and have of course corrected it in the revised version – we apologize for this inadvertent omission. All further analysis (sample integration, differential expression) and visualization used these SCT corrected counts via calls on *PrepSCTIntegration* and *PrepSCTFindMarkers* functions. Newly added violin plots (Extended Data Fig. 8) visually confirmed the absence of major differences in UMI counts between groups.

Third, if DEGs were “due to the increase in detection in the GFP+ groups” as suggested by the Reviewer, we would only detect upregulated genes (more expression in GFP+ than GFP- nuclei). This is absolutely not the case (Fig. 3, ~40/60% split between up- and down-regulated genes), which is a good “sanity check” that the UMI-depth normalization was effective.

Fourth, it is true that “relatively few nuclei [were] recovered”. This is in fact the main technical difficulty that the entire ensemble field has been facing when aiming to sequence small, discrete ensembles defined by activity-dependent tagging. This is also why our study is one of the very first to do so, and the first in non-hippocampal regions, and we see overcoming this challenge as a major strength of our study.

QC measures for snRNA data are typically shown as a violin plot instead of listing the median of the group (Extended data fig 6d).

We initially presented only the median non-normalized UMI per group for clarity. We have now added full violin plots for normalized UMIs per nucleus per group – thank you for this suggestion.

6. There are many issues with the snRNA analysis, some are outlined below.

a. The use of so many pairwise comparisons to determine DEGs is not statistically sound for this multi-factor dataset.

The use of multiple pairwise comparisons has always been and still remains a complex and controversial issue in RNAseq analysis. While imperfect, there is currently no better alternative for snRNAseq than pairwise comparisons: there is for example no accepted statistical framework to analyze multi-factorial designs in snRNAseq (of the form expression ~ factor1 * factor2) as they are only starting to emerge and gain use in bulk RNAseq studies. For clarity, we have described in detail the comparisons we performed in Fig. 3b. Our reliance on FDR-adjusted p-values, as well as very stringent expression and log fold change thresholds, provides a conservative estimate of “true” differences, with an appropriate balance of false negative vs false positive calls. Of note, our newly-included decoding analyses (see Reviewer 2 comments) confirm that the genes we call DEGs efficiently discriminate between compared groups.

The number of cells for each of the snRNA analysis is unclear and should be reported in the figure legends.

We have now done so – thank you for noting this oversight. For clarity, all numbers are reported as tables in Extended Figure Data 8 instead of in individual figure legends.

b. I compared the gene names between D1-MSN markers, D1-MSNs_Acute_GFP-POSvNEG, and D1-MSNs_Repeated_GFP-POSvNEG and found that 26 genes for acute and 37 genes are in the D1 markers list, indicating that the differences in sequencing depth are likely impacting results. For Arc+/-, 38 markers overlapped with acute degs, and 33 markers overlapped with repeated degs. 25% of representative DEGs in F3j are markers of the cell type.

We do not see how this is a problem, or how this can relate to sequencing depth (also see above comment #5, as all differential expression was performed using library size-normalized UMI counts). It is expected that some of the genes that are enriched in D1 cells (“markers”) are also regulated in D1 cells. What would be concerning would be if D2 marker genes for example would show up as DEGs in D1 cells, but this is not at all the case here.

c. Logistical regression assumes that the distribution of expression levels is similar across groups, but the sequencing depth is very different.

Our understanding is that, on the contrary, one of the advantages of logistic regression is that it does *not* assume similar expression levels across groups, as it models the probability of a binary outcome (group membership) as a function of the predictor (gene expression levels), and thus performs well when expression of a given gene is on/off (i.e., either detected or not in each nucleus). And again, differential expression testing using logistic regression was performed on UMI counts corrected for sequencing depth (SCT counts), which in any case efficiently eliminates potential biases related to sequencing depth (see above comment #5). Further, we did qualitatively compare the results obtained by logistic regression to other options such as Wilcoxon Sum Rank and MAST tests, and the results were overwhelmingly similar, which supports our confidence in the output. We chose logistic regression because it allowed us to regress out unwanted sources of variation (e.g., sample identity when testing for cluster marker genes), as recommended in Seurat vignette.

The chi squared test is sensitive to sample size, and small sample sizes (Arc/GFP population combos) can lead to unreliable p-values.

We agree that chi-square testing can become unreliable for either large or small sample sizes. However, small sample size usually leads to false negatives, i.e., the inability to detect a true significant association, rather than the opposite. We are thus confident in the validity of our significant results. Moreover, the rule of thumb is that expected counts for each cell in the contingency table are to be >5, which is the case here. Finally, for the D1 enrichment in GFP+ cells, this was confirmed by orthogonal RNA FISH experiments (Extended Data Fig. 9). However, should the Reviewer and Editor feel strongly about the unreliability of chi-square testing in this case, we are open to either replacing it with another suggested test, or to remove it entirely as the corresponding interpretations and conclusions are not central to our manuscript.

d. 1 read minimum to consider a nucleus Arc+ - is not appropriate. The distribution of Arc expression between homecage and test should be examined and a threshold determined just as a threshold is determined for classifying a cell as Arc+ with orthogonal techniques like immuno.

We are thankful for the opportunity to better justify our threshold of ≥ 1 UMI to call a nucleus Arc+. We also corrected “read” to “UMI” in the corresponding Methods section – thank you for noting this inadvertent error.

First, this is what has canonically been done to binarize (YES/NO) expression levels of single transcripts in published studies (e.g., PMID 33177708). Second, a threshold of ≥ 1 UMI per nucleus is also justified when plotting the distribution of Arc expression between test and homecage control samples as suggested (Fig. R3): nuclei with <1 Arc UMI are depleted, while nuclei with ≥ 1 Arc UMI are enriched in CPP test samples compared to homecage controls. This is also how one would define positivity thresholds in IHC for instance. Moreover, we show that Arc+ nuclei (defined using the ≥ 1 UMI threshold) also express higher levels of other IEGs (Fig. 5g), which confirms that using this threshold efficiently discriminates cells that induce activity-dependent transcription – we see this as a nice cross-validation of both the ability of snRNAseq to capture recently activated nuclei and the presence of Arc transcripts as a marker for such nuclei. Finally, we show in Fig. 5a that using this threshold allows for the detection of a ~4-fold increase in the number of Arc+ cells after CPP testing compared to homecage controls,

consistent with orthogonal measurements using IHC. In sum, we have robustly validated the detection threshold used our study.

e. It is unclear if the pathway analysis used a background list of genes expressed by cell type for enrichment analysis. This is critical to reduce false positives due to tissue-specific gene expression. The IPA results are essentially the same 4-5 terms, and it's likely the cell type markers would give the same results.

This is indeed a critical consideration, which we had overlooked in former analyses. As suggested, we have now re-run all enrichment analyses in IPA using a background list of expressed genes specific for each cell type. In practice, this reduced the background/reference "universe" of genes against which enrichment is tested to only 16,265 genes in D1-MSNs, and 15,854 genes in D2-MSNs – significantly reduced numbers from the >35,000 genes (whole transcriptome) used previously as a default. We have updated all corresponding figures. The new enrichment analysis, while affecting both pathways (especially for GFP+ vs GFP- neurons) and upstream regulator predictions confirmed and even refined all former conclusions regarding the enrichment for genes and pathways involved in synaptic physiology and signaling in tagged cells, Arc+ cells, etc. This more correct enrichment analysis – which reduced the significance of likely formerly false-positive GO term calls – even further reinforces our argument that repeated ensemble cells exhibit more strongly enriched neuronal plasticity-related transcriptional regulation than acute ensemble cells, for both D1 and D2-MSNs (Fig. 3e,f). Additionally, the list of predicted upstream regulators for the observed transcriptional changes now makes even more sense, with the addition of key, known drug-activated transcriptional regulators like MECP2, HDAC4, and FOS. This makes for even more solid ground to argue on key differences between the acute vs repeated ensembles, and which might underlie the electrophysiological differences seen in Fig. 4. We are thankful for the prompt to re-run these analyses, which we have also discussed in more detail in the Results section, as we appreciate how it has strengthens our manuscript.

7. A recent paper (PMID: 36965548) indicates that a subset of D1-MSNs exhibits the transcriptional response to cocaine. It appears that the D1-MSNs are 2 'clusters' in this dataset, but both have acute/repeated cells within them. The authors should consider how their results relate to this manuscript.

We are grateful for this excellent suggestion, which led to both strong, cross-species validation of both datasets, and novel findings, which we now discuss in the main text with the corresponding data as a stand-alone supplemental figure (see below and new Extended Data Fig. 10).

We are in frequent contact with the Day lab at UAB, who regularly publishes reference snRNAseq of NAc tissue such as the one mentioned by the Reviewer. We mention in our manuscript that our cell-type cluster annotation relies on those studies (PMID: 32637607). We now have added the PMID: 36965548 reference as well.

The most recent study referenced by the Reviewer (Phillips et al, 2023) does indeed dive deeper into subtypes of D1-MSNs: their analysis parameters distinguish 3 clusters of D1-MSNs (so-called Grm8-MSNs also express *Drd1*). Sub-clustering of our D1-MSNs reproduces these 3 clusters with good fidelity (Extended Data Fig. 10) and similar proportions in each cluster (*Drd1*-MSN-1 ~60%, *Drd1*-MSN-2 ~20%, Grm8-MSN ~20%). Further, we plotted the expression of the gene panel used in Phillips et al, Fig. 2d in our dataset to contrast *Drd1*-MSN-1 and *Drd1*-MSN-2, and recapitulated extremely similar patterns (highlighted in blue boxes in Extended Data Fig. 10b-c) with enriched Ca²⁺-related genes in *Drd1*-MSN-1 and enriched K⁺-related genes in *Drd1*-MSN-2. Perhaps even more interestingly, Phillips et al. reported that the *Drd1*-MSN-1 cluster was preferentially activated by recent cocaine exposure (reproduced in Extended Data Fig. 10d), and so did we: GFP+ cells (i.e., cocaine-activated cells) were significantly enriched in that *Drd1*-MSN-1 cluster but depleted in others, especially in the acute/weak pairing condition (~90% of GFP+ cells in that cluster). Together, this stands as a solid validation of our snRNAseq dataset and a remarkable cross-species finding given that our dataset is in mice while the Phillips et al. study is in rats – it overall confirmed that only a subcluster of D1-MSNs is activated by recent cocaine experience.

However, when looking at recall-induced transcriptional activation, we find that this activation is spread among all subtypes of D1-MSNs (Extended Data Fig. 10f), in stark contrast to cocaine-activated cells – although we did detect trends towards stronger activation in *Drd1*-MSN-1 cluster for some IEGs. This is a truly unique and novel finding, which supports the idea that both the molecular and ensemble/circuit-level mechanisms leading to transcriptional activation are different during memory recall and memory encoding. We thank the Reviewer again for the prompt to perform such analyses.

Of note, we think – although we do not have strong enough data to make us confident enough to argue or publish such a hypothesis – that the Grm8-MSNs might correspond to the Tac2+ subpopulation of D1-MSNs that has recently been shown to counter-intuitively oppose cocaine reinforcement (PMID: 35960793). In light of this, it is intriguing to interpret the fact that only the acute, but not repeated, ensemble appears depleted in that

Grm8/Tac2-cluster (Extended Data Fig. 10e): this would be very much consistent with our overall hypothesis that the repeated ensemble is more “negatively” regulating cocaine-associated learning.

Despite these differences between D1-MSN subclusters, and also largely because the GFP+ tagged ensemble is composed overwhelmingly of Drd1-MSN-1 cells (80 to 90%), we wish to keep the remaining analyses focused on combined D1-MSNs, both for overall clarity – we acknowledge that our study is already complex without including those additional notions – and for statistical power as further subclustering will even further reduce the number of cells in each cluster for cluster-specific differential expression analysis. Because Drd1-MSN-1 cells comprises such a large majority of our combined D1-MSNs clusters, it is very likely that the DEGs we observe are driven by that cluster.

The authors might consider plotting the clusters by QC features like nCount or nFeature to ensure that the D1 subclusters (or other subclusters) are not due to quality.

We added the suggested graphs in Extended Data Fig. 8. Neither overall clustering nor D1 subclustering are driven by differences in sequencing quality – which is additionally regressed out during SCTransform normalization (see comment #5 above). Note that glial cells have overall lower UMI detection rates, which is always observed in brain snRNAseq and thought to reflect lower transcriptional complexity.

It isn't a novel finding that D1 MSNs primarily respond to cocaine at the transcriptional level.

This is true, but we never argue that this is the main conclusion of our study. We rather see it as a confirmatory result for the robustness and solidity of our tagging and sequencing approach. The goal, and as a result the added value, of our work resides in fact in being able to more finely access the activated D1-MSNs and D2-MSNs vs their non-activated neighbors and selectively track their fate during either later stages of memory encoding or memory recall. A major lesson of our study is that the field needs to move beyond the simple distinction between D1- and D2 MSNs to consider the distinct mixtures of the two cell types that comprise functionally-relevant ensembles engaged during specific tasks.

8. There is no naïve control (or even a Tom- control) for Figure 4 to properly contrast the acute and repeated results. Are the patched cells D1 or D2?

Thank you for this suggestion – including a naive group is indeed an important control to contrast whether it is the acute or repeated ensemble cells that deviate from a baseline state. We presume that the recorded Tom+ ensemble cells being patched randomly are ~70% D1-MSNs and ~30% D2-MSNs (Fig. 4 and Extended Data Fig. 9). Because we know that D1- and D2-MSNs have distinct electrophysiological properties, including excitability (PMID: 18687967, 31831522), it is important for that naive control group to comprise similar fractions of D1- and D2-MSNs – as other splits could bias the average metrics towards either D1-like or D2-like phenotypes. To circumvent that issue, we recorded MSNs in D1-tomato mice while making sure to maintain a 70/30 ratio. This also had the benefit to record from cells also expressing Tomato.

Fig. 4 has been updated with these new Results. The results are striking: repeated Tom+ cells were more excitable than naive (and acute Tom+ cells) at high current inputs, while there was a trend for acute Tom+ cells to be less excitable than controls (and repeated Tom+ cells) at lower input currents. This is intriguing because it suggests that *both* acute and repeated ensemble cells might have opposite baseline electrophysiological adaptations. This further reinforces the need to study the transcriptional correlates of such differences, which our snRNAseq dataset aims to tackle, and will be the focus of follow-up studies.

Neither the figure caption nor methods indicate the number of mice for these experiments. It seems that the data plotted is from all cells recorded not corrected for the number of mice.

The figure legend indicates the total number of mice and total number of cells for each group. Data are indeed plotted with $n = \text{neuron}$, with all statistics appropriately corrected for such nested design (non-independent individual neurons nested within independent animals) according to current best practices using linear mixed models (LMM) that include random intercept factors to model both nesting (for all graphs) and repeated measures (for input/output curves). This was listed in the figure legend (as LMM-ANOVA), but we did forget to include the corresponding description in the Methods – this has been corrected in the revised manuscript. Thank you for noting this oversight.

9. Semantics comments: 1) Figure 1 refers to acute vs repeated as the number of injections, while figure 2 refers to acute vs repeated as the ‘conditioning strength’. While I see a conceptual relationship, they are distinct. Early vs late and acute vs repeated are also interchangeably used.

These are indeed semantic points that we debated at great length, with the objective of maximal clarity. Our choice was to use acute vs repeated when describing experimental protocols (either for locomotor sensitization or CPP), early vs late when describing stages of memory encoding and plasticity, and weak vs strong when describing memory strength. We used acute vs repeated as much as possible, as we think these are the most practical and straight-forward definition, while early vs late or weak vs strong carry some level of interpretation. Thank you for the opportunity to clarify.

2) 4OHT and tamoxifen are used presumed interchangeably? They are distinct drugs with very different labeling time courses.

We have exclusively used 4OHT. We have modified every occurrence of “tamoxifen” to “4OHT”. Thank you for catching this inaccuracy.

3) CPP memory retrieval is not context-induced relapse.

We agree. We do not refer to CPP retrieval as context-induced relapse at any point in the manuscript, but rather as a model for drug-context Pavlovian associative learning.

4) if overlap is low, is this an engram/ensemble? Is cocaine exposure an ensemble or a stimulus response?

In our view, there are key conceptual differences between an “ensemble” and an “engram”, which we clarify below.

An ensemble is any group of cells that share some common denominating feature, either sometimes phenotypical (e.g., ensembles consisting of pyramidal neurons vs interneurons ensembles or of D1-MSNs vs D2-MSNs) or most commonly functional (co-activated cells, observed whether by IEG expression – as we do here with Arc induction – or concomitant time-locked spiking with *in vivo* electrophysiology, or concomitant time-locked calcium responses with *in vivo* 2-photon calcium imaging, PMID: 38155864). There is not any direct relationship to overlap or reactivation in the definition of “ensembles”.

Defining an engram is much more complex. As now commonly accepted in the field, we abide by the definition initially proposed by Tonegawa and colleagues (see Reviewer 1 comment #2 for detailed discussion). Here, the notion of overlap/reactivation becomes critical as one of the criteria for engram properties is that neurons that are activated during memory encoding are reactivated during memory recall.

In that sense, in our study, cocaine exposure leads to Arc+ ensembles (as in groups of neurons that concomitantly induce Arc). We then go on to show that a portion – different between acute vs repeated ensembles – of the cells in those ensembles are likely to be part of an engram for cocaine-context memories: these cells (i) are reactivated, (ii) show plasticity, and (iii) are behaviorally sufficient for memory recall. Our study is the first to establish such engram-like properties for Arc ensembles in NAc. We try to be as clear as possible with this distinction throughout the text, as some of our analyses focus on ensembles (e.g., comparing GFP+ to GFP- neurons), while others tackle engram properties more directly (e.g., reactivation rates, reactivated cells, etc.).

Moreover, overlap/reactivation is not particularly “low” as stated by the Reviewer: reactivation rates of 20-40% are in the exact same range, if not higher, than what is observed for hippocampal or amygdalar engrams (see Discussion).

How did this paper address allocation (mentioned in the discussion)?

Ensemble or engram allocation is a fascinating question, which is just now starting to be investigated at the molecular level with the necessary precision (e.g., PMID: 31837649, 25319707, 33177708, 39052786). Our study touches upon these concepts but do not test them *per se*, as our observations are correlational: because we sequence nuclei after activation and/or learning but not before, we cannot distinguish whether the features that segregate ensemble vs non-ensemble cells were pre-existing (and thus guided allocation) or are acquired through plasticity as a result such activation. In the last figure, we designed an analysis strategy that borders on this distinction in the context of engram reactivation by removing activity-dependent transcripts from the list of transcripts that differ between reactivated and non-reactivated ensemble cells (Fig. 6c), which was confirmed by the newly-added classifier analysis (see Reviewer 2). However, this is only a first step which will guide further in-depth, causal studies in follow-up work, and the reason why we only discuss allocation in the Discussion. We are happy to remove or modify this point if the Reviewer and Editor would like us to do so.

10. I could not evaluate any of the code used to generate the figures as it was not available for review, and the sequencing data was not available via a GEO reviewer access code. I requested access via the editor, but the authors were not responsive.

We apologize for the miscommunication. R code and count matrices (.h5 cellranger outputs) are deposited in a public repo on Github (github.com/arthurgodino/Salery_snRNAseq) for code review. All raw (.fastq), processed (.h5), and metadata will be deposited with open access on GEO upon manuscript publication. We ask the Editor to contact us should the Reviewer have any difficulty in viewing the raw data: we are most happy to share any and all of it.

REVIEWER COMMENTS

We thank all 3 Reviewers, as well as the Early Career Reviewer, for their time evaluating our revised manuscript, and their very positive comments on our study's findings, quality, and impact. Please find below a point-by-point response to the few remaining minor comments.

Reviewer #1 (Remarks to the Author):

I was very positive on the earlier version of this manuscript and now the authors have made it even better. Recommend accepting.

Thank for your kind words, and for your positive recommendation.

Reviewer #2 (Remarks to the Author):

The authors have thoughtfully addressed the reviewer's concerns and provided thorough responses, which we greatly appreciate. The additional clarifications regarding experimental design and methodology, as well as the incorporation of the SVM analysis to identify discrete ensemble groups, significantly enhance the manuscript. The inclusion of Arc in the title highlights an intriguing aspect of the findings, and we believe that adding Fig. R1 in the revised submission would further increase the precision and impact of this work.

Thank you for the positive comments on our recent additions to the manuscript. As suggested, we have included former Fig. R1 in the main Fig. 5 as a new panel – we agree that the overlap with activity-dependent ensembles defined by other IEGs is a very useful resource for the field.

Finally, if the core versus shell distinction falls beyond the intended scope, acknowledging this as a limitation in the discussion would provide valuable context for readers.

Thank you for the suggestion. We have added a short sentence on core vs shell differences and our study's limitations in that regard in the Discussion, and have included relevant references.

Reviewer #3 (Remarks to the Author):

We are thankful for your work in helping with the review of our manuscript.

Reviewer #4 (Remarks to the Author):

My primary concern remains with the snRNA-seq data collection. While the authors addressed some issues, such as defining the distribution of Arc expression between homecage and test groups, they have overlooked a critical flaw in their data acquisition that severely limits their analysis. The authors' claim that "The use of multiple pairwise comparisons has always been and still remains a complex and controversial issue in RNA-seq analysis" is incorrect. Statistical packages like limma, EdgeR, and DESeq2 have long been validated for RNA-seq (and snRNA-seq) analysis, including complex multifactorial designs (e.g., factor1*factor2) that appropriately account for multiple comparisons. Although single-cell studies have lagged behind bulk RNA-seq in adopting such methods—partly due to the practice of pooling samples, as done in this study

We are grateful for the opportunity to discuss in more detail how the analytical framework we used for our snRNAseq dataset fits within the current literature and the current standards on the use of this technology in neuroscience. The Reviewer correctly notes that for bulk RNAseq analysis of multifactorial datasets, the widely-used and well-validated packages like voomlimma, edgeR, and DESeq2 allow for fairly easy-to-use and easy-to-interpret methods to appropriately model variance partitioning in such multi-factor experimental designs: we, like many others, have consistently used those factor1*factor2 designs in our bulk RNAseq studies (as just one example please see PMID: 36889314). The Reviewer is also right to note that single-cell studies have lagged in adopting such strategies. We point out that using voomlimma/edgeR/DESeq2 with multifactorial designs is only possible when pseudo-bulking snRNAseq/scRNAseq data into sample-level count matrices – using those packages at the cell-level would violate many of the underlying assumptions (largely related to zero-inflation and drop-out rates) and lead to gross overfitting. No published study to date has used either approach at the cell-level, reassuringly. Accordingly, **we simply cannot use those packages on our dataset** which technically required unavoidable sample pooling (see below for detailed discussion).

However, we want to further justify and illustrate for the Reviewer and Editor how our study adheres to current analysis standards. We manually curated manuscripts published in the last 6 months (Jul-Nov 2024) in *Nature*, *Nature Neuroscience*, or *Nature Communications* that performed cluster-specific differential expression analysis (DEA) on sn/scRNAseq datasets. The resulting list (Table below) is not meant to be exhaustive, but rather representative of where the field stands. As stated above, voomlima/edgeR/DESeq2 have only been used in studies where sample collection allows for sample-level analyses,

Reference	PMID	Species	Exp. Design	Technology	Technical replicates	Analysis level	Pseudobulk	DEA	p adjustment?
Wälchli et al, Nature, 2024	38987604	Human	Multifactorial	10X Genomics (microfluidics)	Y	Sample-level	Y	DESeq2	NO
Fröhlich et al, Nat Neuro, 2024	39227716	Human	Multifactorial	10X Genomics (microfluidics)	Y	Sample-level	Y	dreamlet LMM	FDR<.1
Gruber et al, Nat Comms, 2024	39578444	Mouse	Multifactorial	10X Genomics (microfluidics)	Y	Sample-level	Y	DESeq2	NO
Brunner et al, Nat Comms, 2024	39098920	Mouse	1-factor, >2 levels	10X Genomics (microfluidics)	Y	Pseudo-sample	Y	DESeq2 & Wilcoxon	FDR<.05
Rodrigues-Amorim et al, Nat Comms, 2024	39112447	Mouse	Multifactorial	10X Genomics (microfluidics)	Y	Sample-level	Y	voom-limma	FDR<.05
Yen et al, Nat Comms, 2024	39604385	Mouse	Multifactorial	ScaleBio (plate)	Y (but pooled)	Sample-level	Y	DESeq2	FDR<.05
Garma et al, Nat Comms, 2024	39039043	Human	Pairwise	10X Genomics (microfluidics)	Y	Sample- & Cell-level	Y/N	edgeR-LRT & Wilcoxon	FDR<.05
Wang et al, Nat Comms, 2024	38987616	Human	Multifactorial	10X Genomics (microfluidics)	Y	Cell-level	N	MAST	Bonferroni
Galimberti et al, Nat Comms, 2024	39095390	Human (organoids)	Multifactorial	10X Genomics (microfluidics)	Y	Cell-level	N	Wilcoxon	NO
Slota et al, Nat Comms, 2024	39580485	Mouse	Pairwise	10X Genomics (microfluidics)	Y	Cell-level	N	MAST	FDR<.05
Hughes et al, Nat Comms, 2024	39117647	Mouse	Multifactorial	10X Genomics (microfluidics)	Y	Cell-level	N	MAST & Wilcoxon	FDR<.05
Perez et al, Nat Comms, 2024	39013876	Mouse	Multifactorial	10X Genomics (microfluidics)	Y	Cell-level	N	Negative binomial	Bonferroni
Tsurutani et al, Nat Comms, 2024	39370447	Mouse	Pairwise	10X Genomics (microfluidics)	N (pooled)	Cell-level	N	Wilcoxon	NO
Mantas et al, Nat Comms, 2024	39289358	Mouse	Pairwise	10X Genomics Visium (spatial)	Y	Cell-level	N	Wilcoxon	FDR<.1
Bormann et al, Nat Comms, 2024	39043661	Rat	Multifactorial	10X Genomics (microfluidics)	Y	Cell-level	N	MAST	Bonferroni
Schartz et al, Nat Comms, 2024	39147742	Mouse	Multifactorial	Parse Bio Evercode (plate)	Y	Cell-level	N	Wilcoxon	FDR<.05
Liu et al, Nature, 2024	38961294	Mouse	Multifactorial	SmartSeq2 (plate)	Y	Cell-level	N	Mann-Whitney U	FDR<.01
Babey et al, Nature, 2024	38987585	Mouse	Pairwise	SmartSeq2 (plate)	N (pooled)	Cell-level	N	Wilcoxon	NO

i.e., where replicates are (mostly, not in Yen et al. or Brunner et al.) individual biological replicates. Moreover, these largely comprise human studies, where samples and cell populations are large and maintaining individual variability is critical. Of note, the few mouse studies using pseudo-bulking also use large chunks of tissue and do not require pre-sorting of rare cell populations as is required in our study. A majority of recent studies reviewed and published in *Nature Group* journals thus performed cell-level, pairwise-comparisons analyses like we do, even if in some case they theoretically could use sample-level analyses. Moreover, focusing on studies with multifactorial designs like ours, it also stands out that the only listed study that sequenced activity-dependent tagged and sorted populations (like we do), published in *Nature* by Stephen Quake and Thomas Südhof (Liu et al.) used cell-level analyses, even as they used plate-based sorting (see also below).

We also note variability in the statistical framework used to call DEGs. We have thus benchmarked the method we used (logistic regression, LR) against the two other most commonly used methods (Wilcoxon Rank Sum test and MAST; PMID: 26653891) in our own dataset. We found extremely high convergence of DEG calls, as demonstrated by very strong and very significant correlations between respective FDR-adjusted p-values (Pearson's $r = 0.97$, $p < 10^{-16}$ vs Wilcoxon; $r = 0.84$, $p < 10^{-16}$ vs MAST), and very large and very significant overlap of respective DEG lists (84%, Fisher's exact test $p < 10^{-16}$ vs Wilcoxon; 69%, Fisher's exact test $p < 10^{-16}$ vs MAST). LR testing was also more conservative than Wilcoxon (+9% DEG calls with Wilcoxon vs LR) and only slightly more liberal than MAST (-3% DEG calls with MAST vs LR), altogether indicating that LR is a good, middle-of-the-road, robust (as in, consistent with others) method for DEG analyses especially in an exploratory setting like in our study. We chose LR because, unlike Wilcoxon, it allows for regressing out unwanted sources of variations as we specify in Methods (like a factor2 if looking for a factor1 effect in a factor1*factor2 design), which we see as the next best possible strategy outside of voomlimma/edgeR/DESeq2. We also point out that we use FDR adjusted p-values, a more conservative approach as compared to other studies that do not.

In conclusion, we hope to have convinced the Reviewer and Editor that ***our study entirely adheres to current publication standards in snRNAseq analyses***, especially in the face of the additional technical challenges we had to overcome to sequence tagged ensembles (e.g., pooling required by the small percentage of ensemble cells in the tissue) described below. *Our pooling strategy, the absence of pseudo-bulking, and the algorithms used for DEG analyses are all clearly stated in our manuscript's Methods.*

—this limitation was entirely avoidable. For example, a plate-based method like Smart-seq could have been used to 'sequence small, discrete ensembles defined by activity-dependent tagging' while retaining sample-level resolution. This makes the decision to pool samples surprising, particularly in a study otherwise so rigorous in its statistical frameworks and data analysis. As pooling limits the ability to model biological variability, it represents a critical confound in data acquisition that undermines the authors' claims and the study's overall impact.

This comment hits particularly hard, especially the words "entirely avoidable". In fact, sorting and sequencing our Arc ensembles using plate-based strategies like SmartSeq via collaboration with experts around us at Mount Sinai was the original plan, with the goal of keeping individual sample resolution and being able to correlate gene expression with behavior, as well as to use sample-level analyses (see above). We spent significant time, energy, and resources trying to pilot such a sorting and sequencing strategy, only to come up short: unsatisfactory yield for both nuclei and RNA, RNA quality, contamination, etc. This is why we pivoted back to 10X microfluidics, with the unsurmountable caveat that we had to pool samples. Sequencing activated or ensemble cells has been and remains a critical goal for the field, and many others have reported similar obstacles. This is why our study, although with the limitations that samples are pooled and analyses use pairwise comparisons, *remains the first to sequence memory-relevant single cells in the striatum, and one of the very first in the brain (again, others, like Liu et al. discussed above, also use cell-level analyses).*

We have added a paragraph acknowledging these limitations in our Discussion. In particular, we highlight the recent, unpublished work of our colleagues Katherine Savell and Bruce Hope at NIDA, who developed a new plate-based sorting and sequencing technique named XPoSE-seq to capture and sequence such activity-labeled small populations with convincing yield, efficiency, and selectivity (bioRxiv: doi: <https://doi.org/10.1101/2023.09.27.559834>). This technique, which they have applied to mPFC ensembles so far, stands as an exciting platform for the field going forward – including future studies of ours. *We, however, simply cannot reproduce our entire dataset using this new, unpublished technique as part of this revision – this would be a huge, unreasonable ask for revision work which will delay by over 2 years minimum a manuscript that, in its current form, has otherwise convinced the two other Reviewers.*

Overall, our work fits well (and even errs on the conservative side) within current technical and analytical limitations for brain snRNAseq datasets, as we did the best possible to date given the technical challenges we had to overcome. It would be unfair to hold us to higher standards than those currently required for publication in Nature Group journals, as illustrated above.

For the optogenetics data, I appreciate the control experiment that the authors provided in Figure 2i in the updated manuscript. It is necessary for the authors to confirm that this 'home cage tagging with cocaine' is indeed a similarly-sized ensemble as the 'weak pairing of cocaine and cpp box' ensemble. A large body of work from A Badiani and T Robinson show that the context in which a drug is administered in part dictates the size of the 'ensemble'.

We have now added quantification for this statement in Extended Data Fig. 6 – thank you for noting this oversight. Of note, the work from Badiani and Robinson is exclusively done in rats. It is unknown how much this difference translates to mice, as per their own admission.

On the topic of F-score vs. 'novel analysis framework' (logit-based model?), I appreciate the in-depth discussion of the analysis method. Logit-based models also have limitations, such as suffering from over-fitting with low biological replicates and are less intuitive to interpret, but the results from both methods do agree with one another, so I will leave it alone. If the authors do want this method to catch on in the engram field, I suggest creating a vignette for tagged/stained cells specifically. We agree on the fact that all analyses have their own limitations – we see the convergence of the conclusions of multiple parallel approaches as a sign of strength and robustness. Creating a method vignette is an interesting suggestion – thank you.

POINT-BY-POINT RESPONSE TO REVIEWER COMMENTS – 3rd REVISION

Reviewer #4 (Remarks to the Author):

I appreciate the authors thorough review of recent Nature group publications. It helped paint an image of Nature group's current standards, although the variability in workflows/statistical standards among recently published studies is surprising. Brunner et al stuck out for the 'pseudo-sample' classification when sample-level information is not available, as in the case of the present study's experimental design.

Thank you for noting the thoroughness of our literature review. We agree that the level of variability is surprising.

The editor may decide if they want to see this prior to publication, but I recommend the authors rerun their analysis with a pseudo-sampling/multifactor analysis and see if their results are reproducible between methods. The main critique in using cell-level analyses (apart from preventing multifactor analysis) is it can overinflate statistical power since cells represent more of a technical replicate over a biological replicate.

This is a good idea. We have now used the exact same approach as in Brunner et al using random pseudo-sampling and multifactorial pseudo-sample-level DESeq2. In each cluster (D1-MSNs and D2-MSNs), the same number of nuclei were randomly selected for each experimental group and these subsets randomly split into 4 equal pseudo-samples. Gene expression data were summarized for each pseudo-sample using Seurat's *AggregateExpression* function and standard multifactorial DESeq2 (design: ~ ArcGFP+.vs.ArcGFP- * Repeated.5P.vs.Acute.1P) was run on the resulting count matrices. Similar to our former comparisons with other cell-level methods (Wilcoxon, MAST), DEGs called by pseudo-sampled, multifactorial DESeq2 were very consistent with those called by logistic regression.

We see the convergence of results at the gene expression pattern level between these different methods as a very strong evidence of robustness. *This is why we have now added a new Supplemental Table S6 quantifying the similarity of our DEG analyses between distinct methods (cell-level Wilcoxon, MAST, and Negative Binomial) and the Brunner et al pseudo-sampled analysis suggested by the Reviewer. We also briefly mention these comparisons in the Results, Discussion and Methods section.*

We have now performed most currently available analysis frameworks on our dataset, which – in terms of overall gene expression patterns – all resulted in the same novel conclusions: 1. Transcriptional plasticity is induced primarily in Arc-activated cells; 2. Cells activated at early vs late stages of cocaine exposure are transcriptionally different; 3. Arc induction denotes larger patterns of activity-dependent transcription; and 4. There are transcriptional substrates of ensemble reactivation.

Clearly, individual gene significance varies between analyses, as would be expected and is the case with all such datasets in the literature. Overall, our study does not focus on select candidates, but rather surveys larger transcriptomic landscapes, which are extremely similar across analyses (for instance, enriched in plasticity-related genes, etc.). We also want to point out that we publicly provide all raw and processed data, should others (and ourselves) be interested in such select candidate genes.

In the paragraph addressing this limitation, please discuss 'pseudo-sampling' and cite Brunner et al. I looked at the preprint referenced in the discussion paragraph, and it appears to be using multiplexing instead of a plate-based sorting. Multiplexing has been previously described (cite PMID: 31266958), although not widely adopted (as evident in your state of the field table) or successfully used for scarce cells like the Arc GFP tagged cells.

We have added these points, and relevant references, to the Results, Methods and Discussion. The Reviewer is right to note that the Savell et al new method relies on multiplexing and microwells, not plate-sorting *per se*. We have corrected this statement.

I also appreciate their LR/MAST/Wilcoxon justification and FDR correction discussion. I see inconsistencies in the FDR threshold (Figure 5k and 6f uses < 0.1 when the rest of the paper uses at least < 0.05). I recommend FDR < 0.05 , which is consistent across analyses and in line with the current publication standards of studies applying FDR correction in the table provided.

Thank you for appreciating this discussion. As stated above, we are now convinced that congruence across analytical methods is altogether a critical point, which is why we have added those additional methods in Supplemental Information. Thank you for the prompt to perform such additional analyses.

We have used FDR < 0.1 cutoff across all snRNAseq analyses, as stated in Methods. We wish to keep this threshold, which has been used before (e.g., PMID: 39289358), and is still more conservative than no correction at all (e.g., PMID: 39370447, 38987585). Using FDR < 0.05 would trim down our DEG lists even further, which is not the point of such an initial exploratory analysis, especially as we remain focused on overall gene expression patterns. Again, using FDR < 0.05 would not change any of the overall conclusions listed above. We find that using this significance cut-off (in addition to an expression fold change cut-off) yields the best compromise in terms of DEG lists to illustrate patterns of gene regulation. Please also note

that, in figures, we transparently report $FDR < 0.1$ using \sim , a symbol distinct from $*$, which is used exclusively when $FDR < 0.05$; and that the full lists of p-values and FDR-adjusted p-values are provided for all comparisons in Supplemental tables should others be interested in specific genes for further study.

Lastly, we argue that our SVM cross-validation strategy (Extended Data Fig. 10) confirms that DEGs called at $FDR < 0.1$ truly, efficiently and convincingly discriminate between ensembles and conditions, and thus are very likely “true” DEGs. $FDR < 0.1$, in our own dataset, thus has biological meaning and relevance, which is the underlying reason behind significance cut-offs – perhaps even more so than adhering to arbitrary standards, and especially in exploratory settings as is the case here.

The distribution plot of Arc counts included in the last rebuttal letter between home cage and test groups should be included in supplemental.

Thank you for this prompt. This graph is now included in Extended Data Fig. 8.